# Reassessment of shortwave surface cloud radiative forcing in the Arctic: Consideration of surface albedo – cloud interactions

Johannes Stapf[1], André Ehrlich[1], Evelyn Jäkel[1], Christof Lüpkes[2], and Manfred Wendisch[1]

[1]Leipzig Institute for Meteorology (LIM), University of Leipzig, Germany
[2]Alfred Wegener Institute, Helmholtz Centre for Polar and Marine Research, Bremerhaven, Germany

**Correspondence:** Johannes Stapf (johannes.stapf@uni-leipzig.de)

**Abstract.** The concept of cloud radiative forcing (CRF) is commonly applied to quantify the impact of clouds on the surface radiative energy budget (REB). In the Arctic, specific radiative interactions between microphysical and macrophysical properties of clouds and the surface strongly modify the warming or cooling effect of clouds, complicating the estimate of CRF obtained from observations or models. Clouds tend to increase the broadband surface albedo over snow or sea ice surfaces, compared to cloud-free conditions. However, this effect is not adequately considered in the derivation of CRF in the Arctic so far. Therefore, we have quantified the effects caused by surface albedo-cloud interactions over highly reflective snow or sea-ice surfaces on the CRF using radiative transfer simulations and below-cloud airborne observations above the heterogeneous springtime marginal sea ice zone (MIZ) during the Arctic CLoud Observations Using airborne measurements during polar Day (ACLOUD) campaign. The impact of a modified surface albedo in the presence of clouds, as compared to cloud-free conditions, and its dependence on cloud optical thickness is found to be relevant for the estimation of the shortwave CRF. A method to consider this surface albedo effect on CRF estimates by continuously retrieving the cloud-free surface albedo from observations under cloudy conditions is proposed, using an available snow and ice albedo parameterization. Using ACLOUD data reveals that the estimated average shortwave cooling by clouds almost doubles over snow and ice covered surfaces (-62 $\mathrm{W\,m^{-2}}$ instead of -32 $\mathrm{W\,m^{-2}}$), if surface albedo-cloud interactions are considered. As a result, the observed total (shortwave plus longwave) CRF shifted from a warming effect to an almost neutral one. Concerning the seasonal cycle of the surface albedo it is demonstrated that this effect enhances shortwave cooling in periods where snow dominates the surface, and potentially weakens the cooling by optical thin clouds during the summertime melting season. These findings suggest that the surface albedo-cloud interaction should be considered in global climate models and in long-term studies to obtain a realistic estimate of the shortwave CRF to quantify the role of clouds in Arctic amplification.

## 1 Introduction

Interdisciplinary research conducted within the last decades has led to a broader, but not yet complete understanding of the rapid and, compared to mid-latitudes, enhanced warming in the Arctic (so-called Arctic amplification) (Gillett et al., 2008; Overland et al., 2011; Serreze and Barry, 2011; Stroeve et al., 2012; Jeffries et al., 2013; Cohen et al., 2014; Wendisch et al., 2017). Since the numerous interactions of physical processes, responsible for Arctic amplification, are intertwined and difficult

to observe, climate models are needed to quantify the individual contributions of feedback processes to Arctic climate change (Screens and Simmonds, 2010; Pithan and Mauritsen, 2014). However, the model results show a large spread in representing the feedback mechanisms. One prominent example is the cloud radiative feedback, which includes the effects of an increasing cloud amount in the Arctic, balancing between the potential increase of both longwave downward radiation (positive) and

cloud top reflectivity (negative). To enable reliable projections of future climate changes in the Arctic, the understanding of the individual physical processes and feedback mechanisms causing Arctic amplification is required (Pithan and Mauritsen, 2014; Goosse et al., 2018), as well as observations of how clouds influence the Arctic surface radiative energy budget (REB).

To quantify the radiative effect of clouds on the REB, the concept of cloud radiative forcing (CRF, expressed as $\Delta F$) is defined as the difference between the net total (shortwave plus longwave) radiative energy flux densities,

$$F_{\text{net}} = F^{\downarrow} - F^{\uparrow}, \tag{1}$$

also called irradiances, in all-sky ($F_{\text{net,all}}$) and cloud-free ($F_{\text{net,cf}}$) conditions (Ramanathan et al., 1989):

$$\Delta F = F_{\text{net,all}} - F_{\text{net,cf}}. \tag{2}$$

A warming effect at the surface will be caused by clouds if the net radiative flux densities in a cloudy atmosphere are larger than in corresponding cloud-free conditions.

Long-term ground-based observations of CRF in the Arctic (Walsh and Chapman, 1998; Shupe and Intrieri, 2004; Dong et al., 2010; Miller et al., 2015) showed that in the longwave wavelength range ($4 - 100\,\mu\text{m}$) clouds tend to warm the surface. The magnitude of the warming is influenced by macrophysical and microphysical cloud properties (e.g., Shupe and Intrieri, 2004) and by regional characteristics (Miller et al., 2015) and climate change (Cox et al., 2015). In the shortwave spectral range ($0.2 - 4\,\mu\text{m}$), clouds rather cool, whereby the strength and timing over the year is determined, besides cloud microphysical

properties, by the solar zenith angle (SZA) and the seasonal cycle of surface albedo (e.g., Intrieri et al., 2002; Fitzpatrick and Warren, 2007; Dong et al., 2010; Miller et al., 2015). However, the required cloud-free reference ($F_{\text{net,cf}}$) poses a serious problem to all observations in the cloudy Arctic (Shupe et al., 2011), as the unknown thermodynamic and surface albedo conditions in cloud-free environments are modified by the presence of clouds itself.

Low-level clouds in the Arctic boundary layer cause elevated temperature inversions, modified thermodynamic profiles, and

changed turbulent energy and momentum fluxes, as compared to a cloud-free atmosphere. In addition, the clouds modify the surface energy budget by the two competing effects of longwave warming and shortwave cooling. This results in two typical states of thermodynamic profiles (Tjernström and Graversen, 2009) and longwave radiative irradiances (Stramler et al., 2011; Wendisch et al., 2019) observed in the Arctic winter. As demonstrated by Walsh and Chapman (1998), the surface temperature change accompanied by the transitions from cloudy to clear skies is not an instantaneous effect; it rather occurs in the range of

hours to days and potentially only advanced boundary layer models might predict the transition between the two states after a given time.

Besides temperature and humidity changes, clouds modify the illumination and reflection of the surface. For highly reflecting snow surfaces, radiative transfer simulations show that two processes are crucial: (i) A cloud-induced weighting of

the transmitted downward irradiance to smaller wavelengths, causing an increase of shortwave surface albedo, and (ii) a shift from mainly direct to rather diffuse irradiance in cloudy conditions, which decreases the shortwave albedo (Warren, 1982). Observations have shown that there is a tendency that the surface albedo is larger in cloudy, compared to cloud-free conditions (e.g., Grenfell and Perovich, 2008), which was demonstrated for a seasonal cycle by Walsh and Chapman (1998) for highly

reflective surface types. Radiative transfer simulations enable to evaluate in detail the processes involved in the cloud-related surface albedo changes. Both processes (i and ii), have been parameterized for snow and ice, for example by Gardner and Sharp (2010) based on simulations. However, their impact on estimates of CRF in the Arctic have not yet been assessed.

In this study, available approaches to derive the CRF are reviewed, focusing on processes involved in the transition between cloudy and cloud-free state (section 2). After an introduction of the airborne observations, the instrumentation and the radiative

transfer simulations (section 3), we combine a snow surface albedo model with an atmospheric radiative transfer model to show how the surface albedo of different snow and ice types is modified by the presence of clouds (section 4). The potential impact of this surface albedo-cloud interaction on the estimate of shortwave CRF is analyzed depending on the surface type and seasonality. The application of an areal-averaged surface albedo for the radiative transfer simulations of the shortwave cloud-free irradiances (section 5.1) enables us to derive the CRF in the heterogeneous albedo environment of the marginal sea ice zone

(MIZ) using low-level (below cloud) airborne observations of the REB during the Arctic CLoud Observations Using airborne measurements during polar Day (ACLOUD) campaign (Wendisch et al., 2019). A method to retrieve the shortwave surface albedo in the hypothetical cloud-free atmosphere from measurements under cloudy conditions, by using the above mentioned snow and ice albedo parameterization from Gardner and Sharp (2010) and a shortwave transmissivity-based retrieval of cloud liquid water path (Appendix A) is introduced (section 5.2). This allows us, in combination with the derived longwave CRF, to

analyze the general concept of a warming or cooling effect of clouds on the sea ice during the campaign and to illustrate the impact of the surface albedo-cloud interaction for the total balancing effect of clouds in the springtime MIZ (section 6).

## 2   The concept of cloud radiative forcing (CRF)

### 2.1   Review of approaches to derive the CRF

To derive the CRF from ground-based observations, simultaneous measurements of net irradiances in all-sky, i.e. cloudy

($F_{\mathrm{net,all}}$) and cloud-free ($F_{\mathrm{net,cf}}$) conditions would be needed. From a practical point of view it is impossible to simultaneously measure in cloudy and in cloud-free conditions at the same location and time. Therefore, the common approach is to measure net irradiances in cloudy conditions, and estimate the respective net irradiances in the hypothetical cloud-free atmosphere. For ground-based observations, two general approaches have been applied in the past to estimate the $F_{\mathrm{net,cf}}$.

### 2.1.1   Estimating net irradiance in cloud-free conditions

Firstly, a radiative transfer-based approach is used, which aims to estimate the instantaneous CRF by discarding the cloud in the simulations from the observed atmosphere, neglecting changes of the thermodynamic atmospheric properties over time

and differences of the cloudy and cloud-free surface albedo as described by Intrieri et al. (2002); Shupe and Intrieri (2004); Sedlar et al. (2011); Cox et al. (2015); Wang et al. (2018) and partly by Miller et al. (2015) using cloud-free surface albedo observations. A second, rather climatological approach (Walsh and Chapman, 1998; Dong et al., 2010; Cox et al., 2016) uses observations in cloud-free conditions to extrapolate the cloud-free state during cloudy periods. In this technique, either fitting algorithms for the estimate of cloud-free downward irradiance from Long and Ackerman (2000) and Long and Turner (2008) are applied, or the average cloud-free irradiances are used to represent monthly reference values as implemented by Walsh and Chapman (1998) and partly by Dong et al. (2010) where the upward longwave irradiance is averaged. The cloud-free, shortwave upward irradiance can be obtained from methods described in Long (2005).

In these approaches, the physical processes involved in the estimate of $F_{\mathrm{net,cf}}$ are represented differently, which leads to systematic differences in the resulting CRF. From autumn to spring, the longwave CRF derived from the radiative transfer-based approach should tend to simulate a more positive (warming) longwave CRF compared to the climatological approach. This is potentially due to the colder surface temperatures and frequent presence of surface-based temperature inversions in cloud-free conditions, causing a less negative longwave net irradiance compared to the cloud-free simulated cloudy atmosphere, which is in general less stable and exhibits a warmer surface temperature. In late spring and summer, the surface temperature difference between cloud-free and cloudy state is smaller (Walsh and Chapman, 1998), and smaller differences between the CRF estimates of the two approaches should be expected. Therefore, the bias in the longwave CRF estimate between the climatological and radiative transfer-based approach is controlled by the prevailing conditions and the CRF itself.

Similar issues with the cloud-free reference net irradiances are reported from satellite-based approaches (Allan and Ringer, 2003), where a subsampling of cloud-free regions is used for the estimate of CRF similar to the climatological approach. The satellite-observed cloud-free conditions are in general more stable and drier compared to the cloudy regimes assumed to be cloud-free, which affects the obtained longwave CRF values and results in inconsistencies when compared to climate model longwave CRF estimates (Allan and Ringer, 2003), where the cloud-free irradiances are calculated by neglecting clouds in the radiation scheme.

### 2.1.2 Handling of surface albedo

The shortwave CRF is strongly affected by the assumed surface albedo. The potentially lower values of surface albedo in the cloud-free state (Walsh and Chapman, 1998) would result in a more positive estimate of shortwave $F_{\mathrm{net,cf}}$ by the climatological approach compared to the instantaneous one using higher values of surface albedo under cloudy conditions. Thus, an increase of the cooling effect of clouds retrieved from the climatological approach relative to the instantaneous radiative transfer-based CRF has to be expected, whereby a percentage deviation of albedo can be related to the deviation of shortwave $F_{\mathrm{net,cf}}$.

For the instantaneous radiative transfer-based CRF, changes in surface albedo between the cloudy and cloud-free state have been neglected by the use of the prevailing (cloudy) albedo in the radiative transfer simulations (Intrieri et al., 2002; Shupe and Intrieri, 2004; Sedlar et al., 2011; Wang et al., 2018). An exception is the study from Miller et al. (2015), where cloud-free observations of surface albedo are fitted linearly as a function of SZA to obtain cloud-free albedo values during cloudy periods. This approach neglects the non-linear dependence of the albedo with SZA (Gardner and Sharp, 2010), the impact of snow

grain size, and potential seasonal changes of cloud-free surface albedo indicated by the observed albedo shown in Miller et al. (2018), and thus, induces large uncertainties in the estimate of cloud-free shortwave net irradiance and may even distort the obtained seasonal cycle of CRF. The climatological approach from Long (2005) estimates the cloud-free surface albedo during cloudy periods based on observations during cloud-free conditions, taking the prevailing SZA into account. However, it should be noted that for longer cloudy periods the cloudy-sky (observed) surface albedo is used in combination with downward cloud-free irradiance to represent the upward shortwave irradiance because an extrapolation in changing albedo conditions caused by precipitation and melting events (changes in snow microphysical properties) is not possible. An application of the climatological approach is primarily limited by the high cloud fraction commonly observed in the Arctic (Shupe et al., 2011). It causes large uncertainties in the estimated cloud-free irradiance, as reported by Intrieri et al. (2002), preventing an application to long-term observations with reported high cloud fractions (e.g., Sedlar et al., 2011). Although the climatological approach will produce a more realistic estimate of CRF (especially longwave) with reduced uncertainties and representation of humidity changes (Dong et al., 2006), it remains unclear how representative a monthly average of cloud-free irradiance with a monthly averaged cloud fractions of often well above 90 % can be.

The potential systematic differences in shortwave CRF estimates associated with the respective assumed surface albedo motivate to provide a quantitative measure of these differences as well as an improved solution for the albedo reference to enable a more harmonized understanding of CRF.

## 2.2 Definitions and process handling

Below, we derive the radiative transfer-based instantaneous CRF. Our approach is unique in that we use a continuous estimate of the cloud-free surface albedo of snow and ice obtained from concurrent observations in cloudy conditions, and in that we account for horizontal photon transport through using an areal-averaged surface albedo to compute the downward shortwave irradiances, as discussed below.

### 2.2.1 Longwave CRF

To distinguish between components of the surface CRF, Eq. 2 is separated in longwave and shortwave terms. The longwave component reads:

$$\Delta F_{\mathrm{lw}} = \left( F_{\mathrm{lw,all}}^{\downarrow} - F_{\mathrm{lw,all}}^{\uparrow} \right) - \left( F_{\mathrm{lw,cf}}^{\downarrow} - F_{\mathrm{lw,cf}}^{\uparrow} \right). \tag{3}$$

As was stated by Cox et al. (2015), the CRF definition refers to net irradiances, while the cloud radiative effect (CRE) quantifies changes in the downward irradiance. By splitting the upward terms in Eq. 3 in a component emitted by the surface with a temperature $T_{\mathrm{s}}$ and broadband surface emissivity $\epsilon_{\mathrm{s}}$ of 0.99 (Warren, 1982) as well as a reflected residual of $F_{\mathrm{lw}}^{\downarrow}$:

$$-F_{\mathrm{lw,all}}^{\uparrow} + F_{\mathrm{lw,cf}}^{\uparrow} = -\epsilon_{\mathrm{s}} \cdot \sigma \cdot T_{\mathrm{s}}^4 - (1 - \epsilon_{\mathrm{s}}) \cdot F_{\mathrm{lw,all}}^{\downarrow} + \epsilon_{\mathrm{s}} \cdot \sigma \cdot T_{\mathrm{s}}^4 + (1 - \epsilon_{\mathrm{s}}) \cdot F_{\mathrm{lw,cf}}^{\downarrow}, \tag{4}$$

the upward term reduces to:

$$-F_{\mathrm{lw,all}}^{\uparrow} + F_{\mathrm{lw,cf}}^{\uparrow} = (1 - \epsilon_{\mathrm{s}}) \cdot (F_{\mathrm{lw,cf}}^{\downarrow} - F_{\mathrm{lw,all}}^{\downarrow}). \tag{5}$$

This approach assumes the same constant surface temperature in the cloudy and cloud-free state, and thus, represents the commonly defined instantaneous longwave CRF similar to Shupe and Intrieri (2004), Sedlar et al. (2011), and Miller et al. (2015) and should be considered in the interpretation of the CRF values as discussed in the previous section. The essential input for radiative transfer simulations in the longwave wavelength range is the atmospheric temperature profile, the absorber gas profile and aerosol. Hence, the longwave instantaneous CRF becomes independent of the upward irradiance and reduces to:

$$\Delta F_{\mathrm{lw}} = F^{\downarrow}_{\mathrm{lw,all}} - F^{\downarrow}_{\mathrm{lw,cf}} + (1 - \epsilon_{\mathrm{s}}) \cdot (F^{\downarrow}_{\mathrm{lw,cf}} - F^{\downarrow}_{\mathrm{lw,all}}). \tag{6}$$

### 2.2.2 Shortwave CRF

The shortwave component of the CRF is given by:

$$\Delta F_{\mathrm{sw}} = \left( F^{\downarrow}_{\mathrm{sw,all}} - F^{\uparrow}_{\mathrm{sw,all}} \right) - \left( F^{\downarrow}_{\mathrm{sw,cf}} - F^{\uparrow}_{\mathrm{sw,cf}} \right). \tag{7}$$

The surface albedo $\alpha$ is defined as the ratio of $F^{\uparrow}_{\mathrm{sw}}$ and $F^{\downarrow}_{\mathrm{sw}}$. To account for surface albedo changes due to different illumination conditions (cloudy, cloud-free) and cloud optical thickness (Warren, 1982), the surface albedo is decomposed into an albedo observed in cloudy conditions ($\alpha_{\mathrm{all}}$) and an albedo, which continuously represents the cloud-free state ($\alpha_{\mathrm{cf}}$). Thus, the instantaneous shortwave CRF definition reads:

$$\Delta F_{\mathrm{sw}} = \left( F^{\downarrow}_{\mathrm{sw,all}} - \alpha_{\mathrm{all}} \cdot F^{\downarrow}_{\mathrm{sw,all}} \right) - \left( F^{\downarrow}_{\mathrm{sw,cf}} - \alpha_{\mathrm{cf}} \cdot F^{\downarrow}_{\mathrm{sw,cf}} \right). \tag{8}$$

Another relevant parameter is the radiative transfer simulated downward shortwave irradiance at the surface in cloud-free conditions ($F^{\downarrow}_{\mathrm{sw,cf}}$), which is modulated by atmospheric parameters, but also by the surface albedo. For highly reflective surface types such as snow, the upward irradiance is significantly higher compared to values obtained over mostly absorbing surfaces like ocean water. Partly the upward irradiance is scattered back towards the surface contributing to the downward irradiance. Consequently, the multiple scattering between surface and atmosphere causes an increase of downward irradiance over snow and ice compared to open ocean. Photons reflected from a bright surface might be scattered back to the surface increasing the downward radiation over dark areas in surrounding ocean water. This might typically happen in the vicinity of the MIZ or in case of leads. For airborne observations in the MIZ as well as ground based measurements in heterogeneous terrain, this effect is not negligible for $F^{\downarrow}_{\mathrm{sw}}$ (Ricchiazzi and Gautier, 1998; Kreuter et al., 2014).

To address this problem, the downward irradiance for the cloud-free conditions in regions with heterogeneous surface albedo needs to be simulated with an areal-averaged albedo $\alpha_{\mathrm{ar}}$, also called effective albedo (Weihs et al., 2001; Wendisch et al., 2004). For example, a local surface albedo over a small lead embedded in homogeneous sea ice is not representative for the areal-averaged surface albedo. To complete the formulation of the shortwave CRF used in this study we modify Eq. 8 to:

$$\Delta F_{\mathrm{sw}} = \left( F^{\downarrow}_{\mathrm{sw,all}} - \alpha_{\mathrm{all}} \cdot F^{\downarrow}_{\mathrm{sw,all}} \right) - \left( F^{\downarrow}_{\mathrm{sw,cf}}\Big|_{\alpha_{\mathrm{ar}}} - \alpha_{\mathrm{cf}} \cdot F^{\downarrow}_{\mathrm{sw,cf}}\Big|_{\alpha_{\mathrm{ar}}} \right). \tag{9}$$

where $F^{\downarrow}_{\mathrm{sw,cf}}\Big|_{\alpha_{\mathrm{ar}}}$ represents the downward shortwave irradiance at the surface simulated with the areal-averaged albedo in cloud-free conditions.

In section 4, the impact of clouds on $\alpha_{\mathrm{cf}}$ and its influence on the estimate of CRF is analyzed using radiative transfer simulations. In section 5.1 the impact of horizontal photon transport on $F^{\downarrow}_{\mathrm{sw,cf}}\big|_{\alpha_{\mathrm{ar}}}$ and the CRF is quantified by illustrative ACLOUD observations.

## 3 Observations and modeling

### 3.1 Airborne measurements and instrumentation

The cloudy atmospheric boundary layer in the MIZ northwest of Svalbard was studied using the research aircraft Polar 5 and Polar 6 from the Alfred Wegener Institute (AWI) during the ACLOUD campaign performed in spring between 23 May and 26 June 2017 (Wendisch et al., 2019). Parts of the flights were dedicated to characterize the near-surface radiative energy budget below ABL clouds. 16 hours of data measured below clouds (if present) at an altitude less than $250\,\mathrm{m}$ (average $80\,\mathrm{m}$) covering

a distance of 3700 km were collected. The sea ice concentration observed along the low-level flights by instruments mounted on the aircraft is displayed in Fig. 1, together with a Moderate Resolution Imaging Spectroradiometer (MODIS) satellite image. During the ACLOUD campaign, the location of the MIZ, indicated by the contour lines of average sea ice fraction ($I_{\mathrm{f}}$), was almost stationary (Knudsen et al., 2018). The sea ice was more compact (higher concentration) north of $81°\,\mathrm{N}$ geographic latitude, and rather heterogeneous towards the west and the open ocean. The majority of flights was conducted over the MIZ

with $66\,\%$ over areas with high ice concentration ($I_{\mathrm{f}} > 80\,\%$) and $17\,\%$ over the region with moderate ice concentration ($80\,\% > I_{\mathrm{f}} > 15\,\%$) and $17\,\%$ over the more or less open ocean ($I_{\mathrm{f}} < 15\,\%$). As the dataset is merged from different flights covering about six weeks, it comprises various sea ice characteristics and synoptic situations (Knudsen et al., 2018). The data should be considered as a snapshot of the late spring conditions in this region.

The instrumentation of Polar 5 and Polar 6 during the ACLOUD campaign is described by Wendisch et al. (2019) and by

Ehrlich et al. (2019b). In this paper, shortwave and longwave, upward and downward broadband irradiance measurements are analyzed. The data were collected with a frequency of $20\,\mathrm{Hz}$ using two sets of pyranometers ($0.2$-$3.6\,\mathrm{\mu m}$) and pyrgeometers ($4.5$-$42\,\mathrm{\mu m}$) (Stapf et al., 2019a). From these irradiance data the net irradiance and surface albedo were derived. The processing of the pyranometer and pyrgeometer data is detailed in Ehrlich et al. (2019b). The surface brightness temperature was determined by a Kelvin Infrared Radiation Thermometer (KT–19) (Stapf et al., 2019a). The ice fraction $I_{\mathrm{f}}$ along the flight track was

estimated from measurements of a digital camera equipped with a hemispheric lens. The geometrically calibrated images were obtained with a sampling frequency of $6\,\mathrm{s}$; from the images the cosine-weighted sea ice concentration was calculated (Jäkel et al., 2019). The local atmospheric thermodynamic state, including air temperature and relative humidity, was determined by dropsondes (Ehrlich et al., 2019a) and aircraft in situ observations (Hartmann et al., 2019) during ascents and descents in the vicinity of the low-level flight sections.

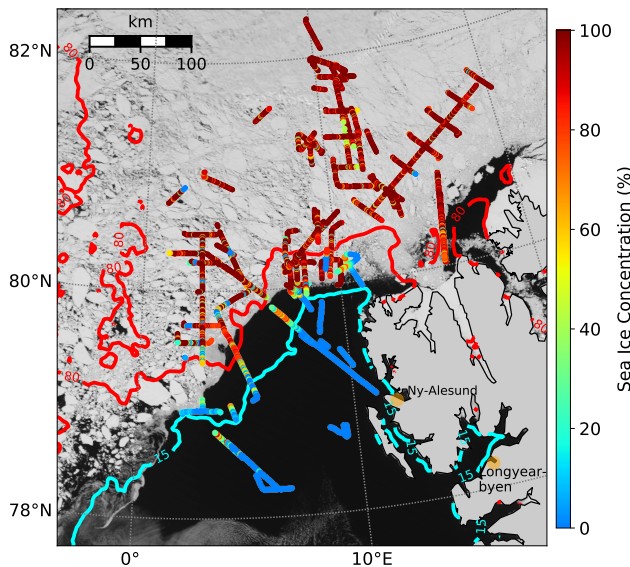

**Figure 1.** MODIS satellite image on 1 June 2017, representing the typical sea ice distribution during the ACLOUD campaign. All low-level flight sections during the ACLOUD campaign are indicated with the sea ice fraction derived from airborne observations. Red (80 %) and light-blue (15 %) contours indicate the campaign average sea ice fraction from daily satellite-based sea ice data (Spreen et al., 2008).

## 3.2 Radiative transfer simulations

The radiative transfer simulations for the cloud-free conditions were performed with the libRadtran package (Emde et al., 2016) using the one-dimensional, plane-parallel discrete ordinate radiative transfer solver DISORT (Stamnes et al., 1988), and the molecular absorption parameterization from Kato et al. (1999) for the shortwave spectral range (0.28–4 µm), and from Gasteiger et al. (2014) for the longwave wavelengths range (4–100 µm). The aerosol particle optical thickness was neglected in the simulations because the full column aerosol information was not available for low-level flights in cloudy conditions. Therefore, the estimated CRF needs to be considered as direct aerosol plus cloud radiative forcing.

The atmospheric state, required as input for the radiative transfer simulations, was based on in situ measurements of temperature and relative humidity on board of both aircraft and, if available, dropsonde measurements from the Polar 5 aircraft. For the thermodynamic state above the flight altitude of the aircraft, the in situ observations were merged with radiosoundings from Ny-Ålesund (Svalbard) (Maturilli, 2017a, b) and onboard Polarstern (Schmithüsen, 2017), which were partly spatially and temporally separated from the airborne observations by several hundred kilometres and up to three hours. The temperature profile below the lowest flight altitude was linearly interpolated to the surface brightness temperature observed by the KT19. For cloudy conditions radiative transfer simulations (assuming a surface emissivity of 0.99, Hori et al., 2006) of the observed atmospheric profiles indicate that neglecting an atmospheric correction for cloudy conditions is justified and the brightness temperature of the KT19 can be related to the surface temperature causing uncertainties below $\pm 0.2\,\mathrm{K}$. During cloud-free

conditions a correction of 0.5 K has been added to the brightness temperature (average value obtained from radiative transfer simulations) to compensate for atmospheric and surface emissivity effects. The impact of flight altitude (average 80 m) can be neglected and potential surface temperature uncertainties hardly affect the downward irradiance simulated in flight altitude. The sub-Arctic summer profile (Anderson et al., 1986) was used to complete the profiles including gas concentra-
tions up to 120 km altitude. Daily ozone concentrations in the flight region of ACLOUD were considered and obtained from http://exp-studies.tor.ec.gc.ca/cgi-bin/selectMap. The high vertical resolution of the in situ observations was reduced for the radiative transfer simulations to 30 m below 1000 m and stepwise increases to 5 km at 120 km altitude. The surface albedo was obtained from upward and downward looking pyranometers and a method described in section 5.2.

Spectral surface albedo values for the sensitivity study in section 4 were simulated using the spectral Two-streAm Radiative
TransfEr in Snow model (TARTES) (Libois et al., 2013). 3D radiative transfer simulations for the albedo smoothing kernels applied in section 5.1 and the Appendix A were performed with the open-source Monte Carlo Atmospheric Radiative Transfer Simulator (MCARaTS) (Iwabuchi, 2006; Iwabuchi and Kobayashi, 2008).

### 3.3   Necessity of a local thermodynamic profile for the estimate of CRF

In the MIZ, the thermodynamic state of the atmosphere changes within short distances due to the influence of the surface on
the airmass (warm air moving north towards cold sea ice, cold air moving south towards warm open ocean) (e.g., Lampert et al., 2012). As shown by Tjernström et al. (2015, 2019), such events significantly impact the local energy budget along the trajectory. As an example of the influence of surface properties and large scale processes on temperature profiles, drop sonde and in situ data measured on 2 June 2017 by instruments installed on Polar 6 are shown in Fig. 2a. The synoptic situation during this flight (west of Svalbard) was characterized by southbound warm air advection with optically thick clouds moving
from the open ocean over the MIZ. The consecutive in situ profiles illustrate the changes in inversion height along the flight leg, which changed from roughly 800 m over the ocean to 250 m over the sea ice within 50 to 100 km. The relative humidity (not shown here) changed accordingly.

Using these profiles, radiative transfer simulations are performed to calculate $F_{\mathrm{sw,cf}}^{\downarrow}$ and $F_{\mathrm{lw,cf}}^{\downarrow}$. The surface albedo and SZA is fixed for this sensitivity study to 0.8 and 60° respectively, similar to the observed conditions over sea ice during that
flight. Therefore, the results only show the impact of changing thermodynamics, but not the effects induced by the observed changes in SZA or surface albedo. Fig. 2b shows the simulated downward irradiance and corresponding values of the shortwave and longwave CRF. While longwave irradiance increases with increasing humidity and temperature (enhanced emission), the shortwave irradiance decreases (enhanced scattering and absorption). The CRF for each atmospheric profile is estimated using the average observed $F_{\mathrm{sw,all}}^{\downarrow}$ and $F_{\mathrm{lw,all}}^{\downarrow}$ during the low-level flight legs observed on 2 June 2017. The results show a strong
variability in $\Delta F$ induced by changes in the thermodynamic structure. The relative deviations range up to 29 % for the longwave and 11 % for the shortwave CRF, which highlights the need to consider changes of the atmospheric thermodynamic state within a few kilometers to derive the CRF. Especially for airmass transformation like warm air intrusions and cold air outbreaks in the Arctic (Pithan et al., 2018), this is a relevant issue.

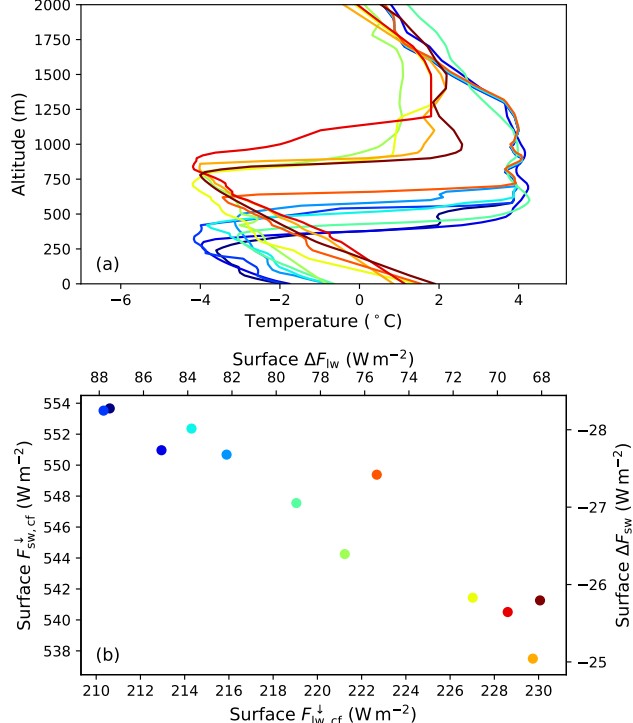

**Figure 2.** (a) Temperature profiles observed during the warm air intrusion on 2 June 2017. The profiles are obtained from dropsonde and in situ measurements (merged with radiosoundings) and are color-coded by the air temperature in the lowest 200 m. (b) Correlation between simulated cloud-free $F_{sw}^{\downarrow}$ and $F_{lw}^{\downarrow}$ assuming the observed atmospheric profiles from (a) (same color code). The second x and y axis shows the expected longwave/shortwave CRF at the surface by assuming a constant $F_{sw,all}^{\downarrow}$ (412 W m$^{-2}$) and $F_{lw,all}^{\downarrow}$ (298 W m$^{-2}$) based on observations. For a better comparability, the surface albedo and the SZA was fixed in the simulations to 0.8 and 60°, respectively.

## 4  Modeling the surface albedo-cloud interaction

The radiative interaction between the spectral surface albedo and the spectral downward irradiance transmitted through clouds and its impact on the wavelength integrated broadband albedo have been analyzed in a wide range of observations (e.g., Grenfell and Perovich, 1984, 2004; Brandt et al., 2005; Grenfell and Perovich, 2008) and modeling studies (e.g., Wiscombe and Warren, 1980; Warren, 1982; Gardner and Sharp, 2010). Observation-based studies provided valuable information on the impact of clouds on the broadband albedo for certain surface types by specifying averaged values of observed surface albedo in cloudy and cloud-free conditions, however, without relating the found differences to a measure of cloud optical thickness. Radiative transfer-based results presented by Shine (1984) illustrate the impact of cloud optical thickness induced spectral weighting effects on the broadband albedo and shortwave net irradiances for common spectral surface albedo types, though, by neglecting the impact of the illumination conditions on the surface albedo which might be misleading for certain surface types. The combination of a surface albedo model, capable of handling this transition from diffuse to direct dominated illumination,

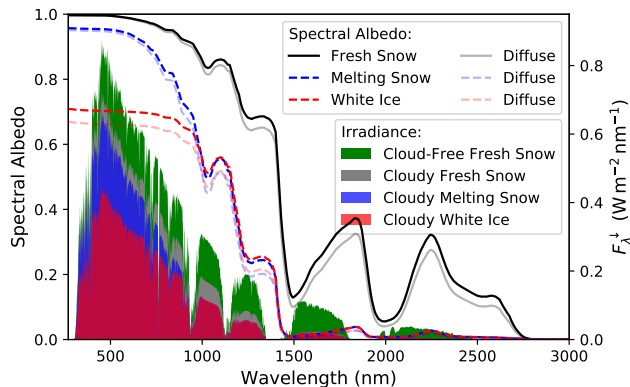

**Figure 3.** Simulated spectral snow albedo of three seasonal sea ice types generated using different SSA and snow thickness values (specified in the main text) above sea ice with spectrally neutral albedo of 0.5. Non-attenuated lines show the albedo of the cloud-free situations (SZA of 65°), attenuated lines (transparent colors) represent the albedo for overcast/diffuse conditions (color-related). The downward irradiance (right y-axis) simulated for these cases are shown by the shaded areas. Green shows the cloud-free spectra over fresh snow, gray, blue and red under cloudy conditions (LWP of $80\,\mathrm{g\,m^{-2}}$) for the surface albedo related by the colors.

with an atmospheric radiative transfer model, enables to study the interplay of processes which shape the broadband surface albedo, as was illustrated by Gardner and Sharp (2010). In the following, such an approach is used to investigate the potential impact of cloud-induced surface albedo changes on the shortwave CRF estimate.

### 4.1 Impact of clouds on surface albedo

5    The effect of clouds on the broadband surface albedo, implemented in Eq. 9, is analysed by a set of albedo spectra of three sea ice types common in the Arctic for different seasons (e.g., Perovich et al., 2002; Grenfell and Perovich, 2004; Zatko and Warren, 2015). Different snow packs with a density of $300\,\mathrm{kg\,m^{-2}}$ and variable snow geometric thickness and specific surface area (SSA, a measure of snow grain size) (Gardner and Sharp, 2010) were defined, and located above a layer representing bare sea ice with a wavelength-constant broadband albedo of 0.5. Fresh cold and dry snow (SSA = $80\,\mathrm{m^2\,kg^{-1}}$, 20 cm thick) represents early to late spring conditions, melting snow (SSA = $5\,\mathrm{m^2\,kg^{-1}}$, 20 cm thick) the melting season in late spring early summer, and thin melting snow/white ice (SSA = $5\,\mathrm{m^2\,kg^{-1}}$, 1 cm thick) summer conditions, before the melt pond formation. The spectral albedo for each type is simulated with the TARTES model for 65° SZA; the respective results are shown in Fig. 3 (lines) together with the corresponding downward irradiances simulated with libRadtran (shaded spectra).

   The impact of snow properties on the spectral surface albedo is characterized by the fact that with decreasing SSA (increasing effective grain size) the absorption at longer wavelengths increases (Warren, 1982; Gardner and Sharp, 2010). It becomes obvious by comparing the albedo of fresh and melting snow in Fig. 3. Thus, a decreasing SSA amplifies the contrast between shorter and longer wavelengths. In contrast, a thinning of the snow layer or impurities in snow enhance the absorption mainly in the shorter visible wavelength range, as illustrated by the albedo of melting snow in comparison to that of white ice.

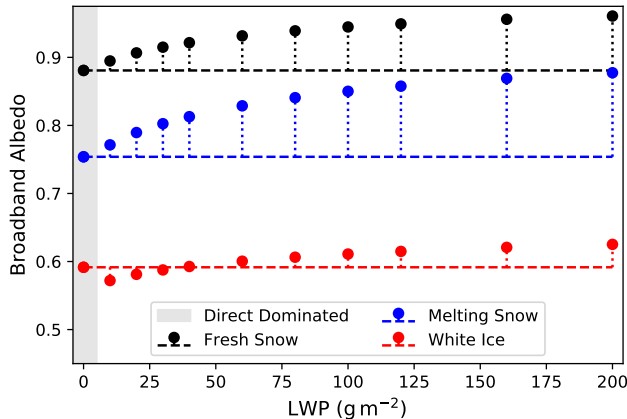

**Figure 4.** Broadband albedo integrated from simulated up- and downward spectral irradiance as a function of cloud LWP ($r_{\mathrm{eff}}$ of $8\,\mu\mathrm{m}$) using the color-related albedo spectra of Fig 3 and a SZA of $65°$. The approximate area of direct dominated radiation is indicated by the gray shading. The horizontal dashed line indicates the cloud-free albedo as a reference.

Two processes influencing the broadband snow albedo are related to the transition from cloud-free to cloudy atmospheric conditions. In an overcast atmosphere with clouds of sufficient optical thickness, mainly diffuse radiation illuminates the surface as compared to cloud-free conditions, when the direct shortwave radiation dominates (Gardner and Sharp, 2010). In the Arctic, large values of SZA ($> 50°$) are common. In overcast conditions, scattering processes in clouds effectively decrease

the averaged incoming (effective) angle of the mainly diffuse irradiance to approximately $50°$ above snow (Warren, 1982). With decreasing so-called effective SZA, the penetration depths of photons into the snow and ice surface increases, enhancing the probability of absorption, and thus, decrease the overall broadband surface albedo (Warren, 1982). In Fig. 3 this effect is illustrated by the attenuated lines (transparent colors) representing the respective diffuse albedo values. Compared to the surface albedo of fresh snow in cloud-free atmospheric conditions (black line) the change of effective SZA (in this example from $65°$

to approximately $50°$ SZA) causes a lower spectral surface albedo (attenuated black line) in the non-visible wavelength range, while the highly reflective visible wavelengths are not affected. Thus, for this surface type only a small impact on the actual broadband albedo can be expected, because for the majority of the related downward shortwave irradiance (e.g. grey shaded area in Fig. 3) the albedo remains high. However, for surface types with a spectral albedo characterized by stronger absorption in the visible wavelength range (albedo of white ice, red dashed line in Fig. 3), also stronger changes between direct-dominated

and diffuse albedo (attenuated dashed red line) and thus also the broadband albedo are expected.

Besides the cloud-induced changes of the effective SZA discussed above, clouds reduce the incident irradiance by attenuating especially in the near-infrared wavelength range (Grenfell and Perovich, 2008). This can be seen in Fig. 3 by comparing the green and gray shaded spectrum representing cloud-free and cloudy conditions with a liquid water path (LWP) of $80\,\mathrm{g\,m^{-2}}$, respectively. In this example, the presence of the cloud reduces the downward irradiance by $18\,\%$ at a wavelength of $500\,\mathrm{nm}$

and $78\,\%$ at $1600\,\mathrm{nm}$. With increasing cloud optical thickness the spectral slope of downward irradiance is imprinted in the

surface spectra. As the spectral albedo of ice and snow is higher for shorter wavelengths (e.g. black line in Fig. 3) and the downward irradiance spectra are shifted to shorter wavelengths in the presence of clouds, the wavelength integrated (broadband) albedo will increase. This effect becomes stronger the more pronounced the slope between visible and near-infrared wavelength becomes, which can be induced by two processes: either stronger absorption by clouds due to a higher LWP, or by the underlying surface albedo with decreasing near-infrared albedo. The latter is controlled by decreasing SSA (transition from fresh to melting snow), resulting in a reduced near-infrared reflection of the surface (compare black and dashed blue lines in Fig. 3), which indirectly affects $F_{\mathrm{sw}}^{\downarrow}$ by a reduced multiple scattering between surface and clouds in this wavelength range (compare grey and blue shaded spectra). However, for the spectral albedo of white ice (dashed red line) the slope in the $F_{\mathrm{sw}}^{\downarrow}$ spectrum (red shaded) is less pronounced than in the other cases, and a weaker increase of broadband albedo is expected for increasing LWP.

For all three surface albedo types shown in Fig. 3, the effect of clouds (as a function of LWP) on the broadband surface albedo is presented in Fig. 4. For the different surface types, a significant change of up to 12 % relative to the individual cloud-free values of surface albedo can be found with increasing cloud optical thickness, which is modulated by the interaction of surface and cloud radiative properties. In general, the lower the ratio of spectral surface albedo between shorter and longer wavelengths is, the stronger is the increase of broadband albedo with increasing LWP. Spectral absorption of the surface in shorter wavelengths decreases the broadband surface albedo, but it will also alter the behaviour with increasing LWP (Fig. 4, red). For low LWP values, the broadband surface albedo is lower compared to cloud-free conditions due to a significant lower spectral diffuse albedo (dashed red and attenuated dashed red line in Fig. 3) at shorter wavelengths. This represents an important difference to the results from Shine (1984) (their Fig. 2), where changes from direct to diffuse radiative transfer were not considered and this feature remained concealed. However, with increasing LWP the weighting effect in transmitted $F_{\mathrm{sw}}^{\downarrow}$ to shorter wavelength compensates/dominates and, as a consequence, it increases the broadband surface albedo compared to cloud-free conditions. This transition depends critically on the cloud optical thickness and SZA and might also be a reason for the lack of observations of a lower cloudy than cloud-free albedo, with the exception of reports by Carroll and Fitch (1981). A rough separation in cloudy and cloud-free conditions might not be sufficient to resolve this feature associated with these more absorbing surface types.

## 4.2 Impact on shortwave CRF

To estimate the significance of the surface albedo-cloud interaction on CRF, radiative transfer simulations are used, either assuming the correct cloud-free surface albedo as a reference, or the prevailing surface albedo in cloudy conditions, as shown in Fig 4. The difference in CRF ($\Delta F_{\mathrm{sw}}(\alpha_{\mathrm{cf}}) - \Delta F_{\mathrm{sw}}(\alpha_{\mathrm{all}})$) between both approaches is shown in Fig. 5 as a function of SZA and LWP.

In case of snow surfaces, influenced by the SSA (Fig. 5a and b), the cooling effect of clouds on the surface is underestimated (bluish colors), if the cloudy albedo ($\alpha_{\mathrm{all}}$) is used to derive the shortwave CRF. In general, the lower the SZA and the higher the LWP are, the stronger the underestimation of the cooling effect becomes. Furthermore, the coarser the snow grains (melting snow) the stronger the underestimation. In contrast, during summer and for thin melting snow or white ice (Fig. 5c), the cooling

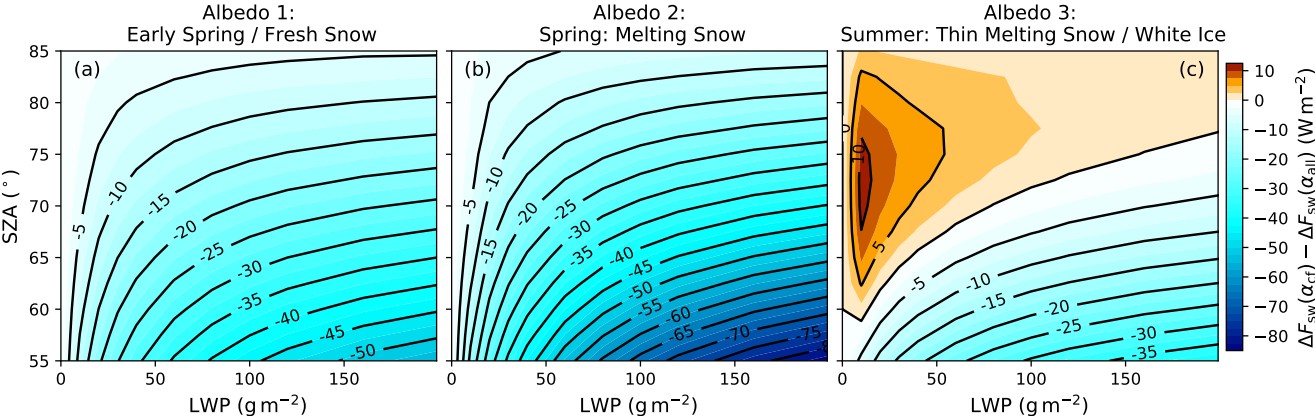

**Figure 5.** Bias of the shortwave CRF ($\Delta F_{\text{sw}}(\alpha_{\text{cf}}) - \Delta F_{\text{sw}}(\alpha_{\text{all}})$) caused by neglecting the change between observed cloudy and cloud-free surface albedo as a function of cloud LWP ($r_{\text{eff}} = 8\,\mu\text{m}$) and SZA. The three albedo types from Fig. 3 have been assumed, (a) fresh snow representative for early spring, (b) melting snow during late spring, and (c) thin melting snow/white ice found in early summer. Negative (bluish) values indicate a stronger shortwave cooling effect for $\Delta F_{\text{sw}}(\alpha_{\text{cf}})$. Changes in direct/diffuse radiation due to SZA are taken into account.

effect is overestimated for low sun and optically thin clouds, if the apparent cloudy albedo is used for $\Delta F_{\text{sw}}$, and shifts towards the underestimation for optically thick clouds and/or lower SZA.

The surface albedo-cloud interaction significantly impacts the estimate of shortwave CRF and the obtained values from the different approaches in the available CRF studies in the Arctic. Especially for clouds over snow, the cooling effect of
clouds is considerably larger when cloud-related changes in surface albedo are considered. Also for climate models with simple albedo parameterizations, for example fixed broadband albedo values for certain surface types, the results from Fig. 5 can be interpreted as a potential bias in the shortwave REB and CRF depending on the cloud optical thickness. Due to the dependence on specific spectral surface albedo types, a seasonal dependence of this surface albedo-cloud interaction, and thus, the shortwave CRF, is indicated.

**4.3  Seasonal cycle of shortwave CRF**

In Fig. 6 a conceptual scheme of the modified seasonal cycle of CRF due to the surface albedo-cloud interaction is proposed. The time series of surface albedo as observed during the Surface Heat Budget of the Arctic Ocean (SHEBA) campaign (Uttal et al., 2002) (data from Roode and Bretherton, 2007) is shown, to illustrate the seasonal transition as reported by Perovich et al. (2002). During spring, early summer, and autumn, surface albedo values related to snow on sea ice are found. The results
from Fig. 5 indicate that the cloud generated shift of transmitted irradiance towards shorter wavelength (process 1 in Fig. 6) is dominant in these situations/seasons and clouds actually induce a stronger cooling effect on the surface, relative to $\Delta F_{\text{sw}}(\alpha_{\text{all}})$. With the beginning of the melting season, the change between diffuse and direct albedo will dominate (process 2) for optically

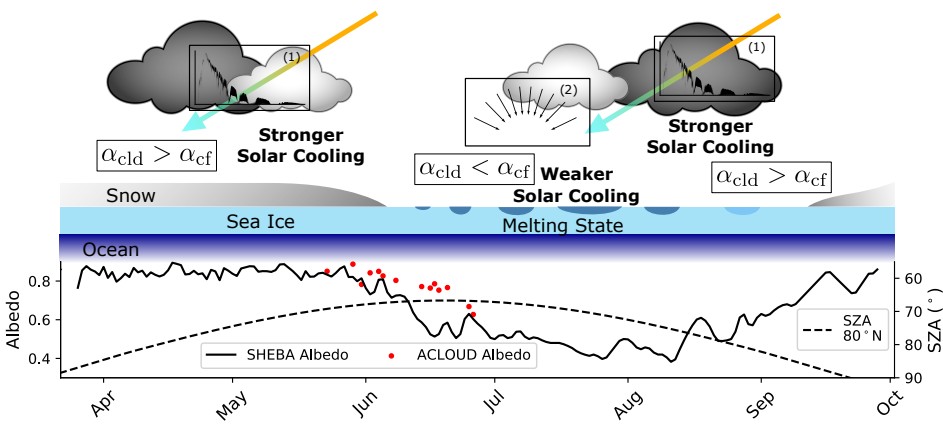

**Figure 6.** Conceptual scheme of the seasonal cycle of surface albedo-cloud interaction related modification of shortwave CRF causing a stronger or weaker cooling relative to $\Delta F_{\text{sw}}(\alpha_{\text{all}})$. Dominant processes influencing the transition from cloudy ($\alpha_{\text{cld}}$) to cloud-free surface albedo ($\alpha_{\text{cf}}$) in the specific season are represented by the icons (1) (weighting of downward irradiance to shorter wavelength with increasing LWP) and (2) (transition from direct to diffuse radiative transfer). The seasonal cycle of surface broadband albedo is shown by SHEBA observations (200 m albedo line). Averaged ACLOUD observations for homogeneous sea ice ($I_{\text{f}} > 95\,\%$) are shown in red scatter points. Computed daily averaged SZA for $80^\circ$ N in dashed black.

thin clouds and high SZA, potentially reducing the clouds' cooling effect on the surface depending on the conditions. In this period the onset of melting (rapidly decreasing albedo), the melt pond fraction, and the SZA (dashed black line in Fig. 6) together with the cloud optical thickness would critically influence the sign of this modification. However, as was reported by Walsh and Chapman (1998), for regions where snow or bare sea ice is found even in summer, a lower albedo in cloud-free

5 conditions, and thus, a stronger cooling effects of clouds can be expected all year long. Though, conclusions about the annually averaged shortwave CRF modified by surface albedo-cloud interactions are not yet possible, as coupled surface-atmosphere radiative transfer models capable of representing surface types like melt ponds are required to study the full seasonal cycle.

For the ACLOUD campaign, snow on sea ice was the dominant surface type (Jäkel et al., 2019), transitioning from cold and fresh snow to melting snow. This explains the slightly delayed decline in surface albedo (Fig. 6, red scatter points) compared

10 to SHEBA data (black), where during this period the melt pond formation can already be identified by the rapidly decreasing surface albedo (Perovich et al., 2002). In exactly this period the transition from a positive (warming) to a negative (cooling) total (shortwave plus longwave) CRF was reported by Intrieri et al. (2002) using all-sky albedo values. Transferred to the results from Fig. 5 already without melt ponds a stronger cooling effect of clouds should be expected by applying $\alpha_{\text{cf}}$, which could also modify the onset of the total cooling effect of clouds during the ACLOUD campaign.

## 5 Refining the derivation of shortwave CRF

To fulfill the requirements of the shortwave CRF definition given in Eq. 9 and thus also to take into account the processes discussed in the previous section 4, the need for a continuous estimate of the cloud-free albedo ($\alpha_{\mathrm{cf}}$) from observations under cloudy conditions becomes obvious. In addition, the application to the airborne observation during ACLOUD with the heterogeneous surface albedo environment in the MIZ requires the estimate of a representative downward shortwave irradiance ($F_{\mathrm{sw,cf}}^{\downarrow}\big|_{\alpha_{\mathrm{ar}}}$). In the following sections both aspects are discussed and the application to ACLOUD observations is demonstrated.

### 5.1 Considering surface albedo heterogeneities and horizontal photon transport

The observed variability of the surface albedo in the MIZ can directly be related to the variability of the observed sea ice fraction $I_{\mathrm{f}}$ shown in Fig. 1. Both will influence the observed field of downward shortwave irradiance, as discussed in section 2.2. For the observations carried out on 23 May 2017, the measured broadband surface albedo along the flight track is shown in Fig. 7a. The low-level section started in the MIZ over large ice floes and small leads with optically thin clouds and ended over the open ocean in the vicinity of the ice edge with occasionally scattered sea ice floe fields and optically thick clouds. Thus, leads and open water areas with the size of a few tens of meters up to a few kilometers caused a highly variable local surface albedo.

In Fig. 7b the simulated $F_{\mathrm{sw,cf}}^{\downarrow}$ using the surface albedo observed at 20 Hz illustrates the problems related to strong albedo fluctuations. The simulated $F_{\mathrm{sw,cf}}^{\downarrow}$ changes on small horizontal scales by up to $35\,\mathrm{W\,m^{-2}}$ (SZA average: $59.2°$). However, due to horizontal photon transport from surrounding ice fields, in reality the changes in $F_{\mathrm{sw,cf}}^{\downarrow}$ are less pronounced. The quantitative impact of multiple scattering on $F_{\mathrm{sw,cf}}^{\downarrow}$ obtained from radiative transfer simulations is indicated by the gray shaded area in Fig. 7b with a maximum contribution of almost $40\,\mathrm{W\,m^{-2}}$ (relative to open ocean). Therefore, the downward irradiance for the cloud-free conditions, required for Eq. 9, needs to be simulated with an appropriate areal-averaged albedo representing the multiple scattering contribution from the surrounding albedo fields.

To estimate a required filter shape and width to obtain an areal-averaged albedo, 3D radiative transfer simulations of a typical scenario are performed (not shown here), where leads of different sizes are embedded in homogeneous sea ice similar to the study from Podgorny et al. (2018). The simulated irradiance of the 3D model output in the vicinity of the leads is reproduced by 1D simulations by applying the filter embedded in Fig. 7b to the 3D modelled albedo (theoretically observed) and by using the thereby obtained areal-averaged albedo for the 1D model simulations to continuously estimate the $F_{\mathrm{sw,cf}}^{\downarrow}\big|_{\alpha_{\mathrm{ar}}}$. The appropriate weighting of near-field and far-field albedo is applied by kernel $k$ defined by a Laplace-distribution:

$$k(x, \mu, \gamma) = \frac{1}{2\gamma}\left(-\frac{|x-\mu|}{\gamma}\right),\tag{10}$$

with $\gamma$ of $5\,\mathrm{km}$, the median $\mu$ and a scale $x$ of $30\,\mathrm{km}$. This rather large filter width indicates that small leads below 1km embedded in homogeneous sea ice show a minor impact on $F_{\mathrm{sw}}^{\downarrow}$ in cloud-free conditions.

The resulting areal-averaged albedo is shown in Fig. 7a, together with the simulated $F_{\mathrm{sw,cf}}^{\downarrow}\big|_{\alpha_{\mathrm{ar}}}$ (Fig. 7b), which follows the large-scale trends of surface albedo, but mitigates small-scale fluctuations. The consequences for the local shortwave CRF

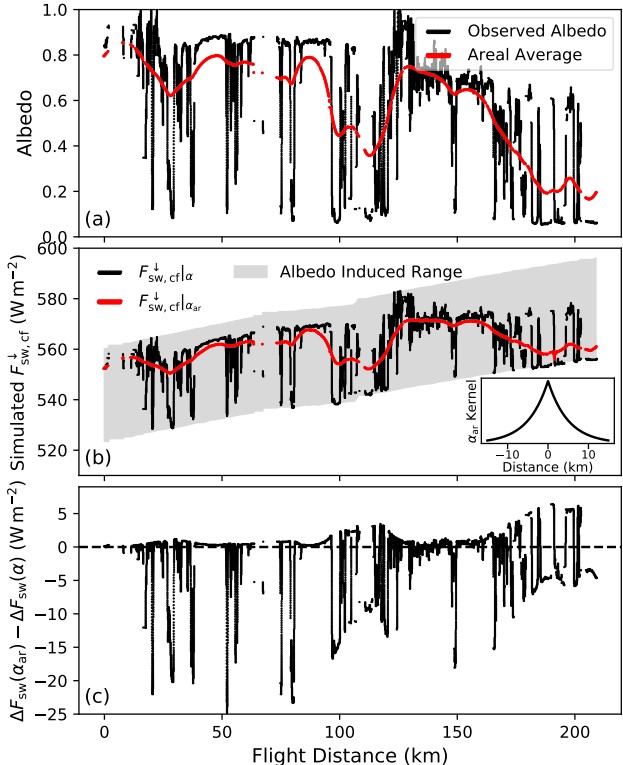

**Figure 7.** Time series (covered distance) of measured broadband surface albedo (black) (a) and simulated $F_{\mathrm{sw,cf}}^{\downarrow}$ (b) along the low-level flight track during 23 May 2017. The red line in (a) shows the areal-averaged albedo using the kernel embedded in (b). (b) The gray area illustrates the potential variability of $F_{\mathrm{sw,cf}}^{\downarrow}$ due to surface albedo changes. The black and red scatter shows the $\left.F_{\mathrm{sw,cf}}^{\downarrow}\right|_{\alpha}$ and $\left.F_{\mathrm{sw,cf}}^{\downarrow}\right|_{\alpha_{\mathrm{ar}}}$ respectively. (c) Difference in shortwave CRF estimate between $\Delta F_{\mathrm{sw}}(\alpha_{\mathrm{ar}})$ and $\Delta F_{\mathrm{sw}}(\alpha)$.

estimate resulting from the neglect of these 3D effects are shown in Fig. 7c as deviation between $\left.F_{\mathrm{sw,cf}}^{\downarrow}\right|_{\alpha_{\mathrm{ar}}}$ and $\left.F_{\mathrm{sw,cf}}^{\downarrow}\right|_{\alpha}$. On average, the effect is of minor importance for the flight section in Fig. 7 (average -1.9 W m$^{-2}$), because under- and overestimation of shortwave CRF cancel in this specific example, similar to results from Benner et al. (2001). On a local scale, however, it should be highlighted that due to horizontal photon transport the $\left.F_{\mathrm{sw,cf}}^{\downarrow}\right|_{\alpha_{\mathrm{ar}}}$ is up to 28 W m$^{-2}$ larger above leads compared

5  to the $\left.F_{\mathrm{sw,cf}}^{\downarrow}\right|_{\alpha}$. The difference in the derived CRF reaches values between -25 W m$^{-2}$ over open water embedded in homogeneous sea ice, where the $F_{\mathrm{sw,cf}}^{\downarrow}$ is underestimated by applying the local albedo, and +6 W m$^{-2}$ above scattered ice floe fields in the ocean with an overestimation of $F_{\mathrm{sw,cf}}^{\downarrow}$. Hence, the uncertainties and artificial fluctuations in CRF are limited by applying the areal-averaged albedo in the $F_{\mathrm{sw,cf}}^{\downarrow}$ simulations. This enables a more reliable estimate of the CRF in the heterogeneous MIZ and over the specific surface types, taking into account that the complexity of surface albedo fields in the MIZ can only

10  be insufficiently represented by this simplified approach to estimate the areal-averaged albedo.

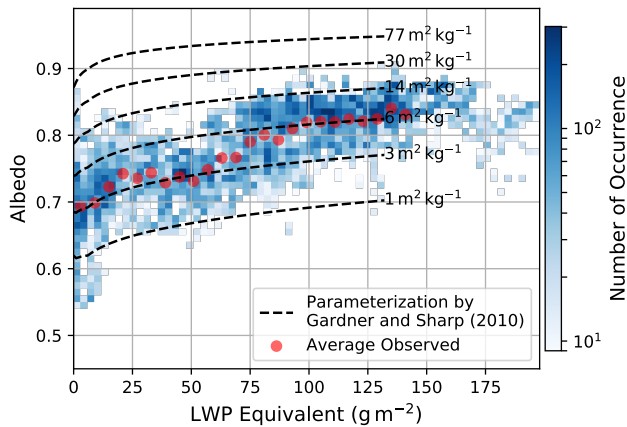

**Figure 8.** Relation between broadband albedo and retrieved LWP equivalent observed on 14 June 2017 ACLOUD flight over homogeneous sea ice ($I_f > 98\%$). The broadband albedo parameterization by Gardner and Sharp (2010) is shown for different SSA and the average SZA by the dashed lines (impurity load of 0.1ppmw). Averaged observations ($6\,\mathrm{g\,m^{-2}}$ bins) are shown in red scattered points.

## 5.2 Retrieval of cloud-free albedo from cloudy-sky observations

To obtain a continuous estimate of the cloud-free albedo ($\alpha_{cf}$) during cloudy conditions as required for Eq. 9, the broadband albedo parameterization developed by Gardner and Sharp (2010) for snow and ice surfaces is applied. Gardner and Sharp (2010) considered the dependence of broadband albedo with respect to SZA, SSA, concentration of light-absorbing carbon as well as the cloud optical thickness. The parameterization is valid for homogeneous snow and ice including a cloud optical thickness below 30 (LWP of 133 $\mathrm{g\,m^{-2}}$ with $r_{eff}$ of 8 μm). During ACLOUD, the observed albedo ranged between 0.9 for homogeneous sea ice covered with cold snow and values below 0.6 during the later stage of the campaign with the onset of melting (Wendisch et al., 2019; Jäkel et al., 2019). To include these data in the analysis and cover this range of albedo values besides the other parameters only as function of the unknown snow grain size (SSA), an impurity load of absorbing carbon of 0.1 ppmw is chosen, which causes a similar spectral behaviour of the albedo as changes in snow thickness (Gardner and Sharp, 2010). Although Jäkel et al. (2019) showed that snow overlaying sea ice was the predominant surface type for closed sea ice conditions during ACLOUD, the potential variability in the spectral surface albedo with respect to absorption in the shortwave wavelength range caused by snow thickness or/and impurity fluctuations during the campaign is only roughly covered by this assumption and needs to be considered in the interpretation of the obtained cloud-free albedo values.

The parameterization is used to generate lookup tables to derive the cloud-free albedo depending on the unobserved SSA and the required variables of cloudy-sky albedo, LWP and local SZA. Isolines of SSA are used to extrapolate the cloud-free albedo (LWP = $0\,\mathrm{g\,m^{-2}}$). To apply the albedo parameterization by Gardner and Sharp (2010) the cloud optical thickness or LWP is required. As the cloud properties change on small horizontal scales, a retrieval of LWP based on the airborne measurements of cloud transmissivity was used, which is described in the Appendix A.

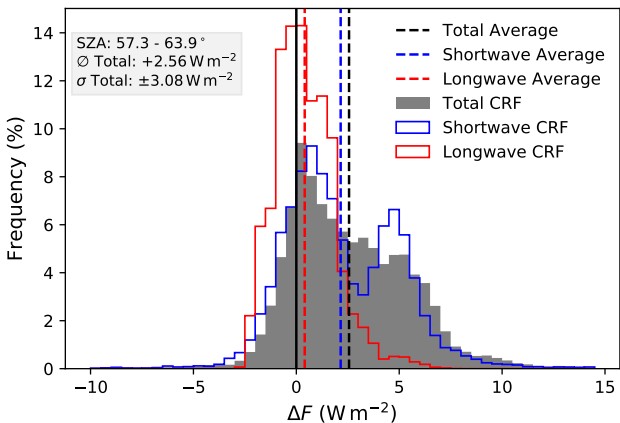

**Figure 9.** Histogram of shortwave, longwave and total $\Delta F$ derived during the cloud-free ACLOUD flight on 25 June 2017. Statistics are given in the gray box (mean $\varnothing$, standard deviation $\sigma$).

The retrieval of LWP allows an investigation of the dependence of the surface albedo on the cloud optical thickness, which is shown in Fig. 8 as an example for measurements over homogeneous sea ice (selected $I_{\mathrm{f}} > 98\,\%$) on 14 June 2017. In addition, the albedo parameterization by Gardner and Sharp (2010) is displayed for different values of SSA (isolines) and of the averaged SZA (63.7°). During 1.7 hours of low-level flights below clouds, a large area was mapped (80.7-81.8° N, 9.8-12.7° E) and a strong variability in cloud optical thickness including occasional openings with direct illumination of the surface and optical thick multilayer clouds was covered. The surface temperatures were close to zero, indicating the beginning melting season (Jäkel et al., 2019). The observed albedo values averaged for $6\,\mathrm{g\,m}^{-2}$ bins (dashed red in Fig. 8) change from 0.7 for low values of LWP to albedo values above 0.8 for a LWP larger than $100\,\mathrm{g\,m}^{-2}$. While the overall trend of increasing albedo with increasing LWP is represented, the slope follows the parameterization for a SSA between $3\,\mathrm{m}^2\,\mathrm{kg}^{-1}$ for lower LWP values and $6\,\mathrm{m}^2\,\mathrm{kg}^{-1}$ for higher LWP. This might be related to different observed cloud and surface areas as the distribution includes data from both aircraft.

Extrapolating the observations (pair of variates) of LWP equivalent and surface albedo along isolines of SSA to a LWP of zero gives an estimate of the cloud-free surface albedo. For the example given here in Fig. 8, for a $\alpha_{\mathrm{all}}$ of 0.82 and LWP of $100\,\mathrm{g\,m}^{-2}$ a cloud-free albedo of 0.74 would be estimated, and thus, 0.06 lower than the observed one in overcast conditions. For LWP values exceeding the limitation of the parameterization the maximum valid LWP was applied. Rarely occurring surface albedo values above/below the range of the parameterization from Gardner and Sharp (2010) have been filtered out. Thus, by combining temporal and spatial appropriate lookup tables (local SZA) and the observations of LWP and broadband surface albedo, a continuous estimate of $\alpha_{\mathrm{cf}}$ is provided, which is suitable for the derivation of the instantaneous CRF that takes the surface albedo-cloud interaction in to account (Eq. 9).

## 5.3 Uncertainties

During ACLOUD, a flight in cloud-free conditions on 25 June 2017 provides the opportunity for a comparison between measured and simulated irradiances in order to estimate the accuracy of this dataset. The difference between observed and simulated $F_{cf}^{\downarrow}$ for the low-level flights of both aircraft (2.1 hours of data) is $5.7\pm7.1\,\mathrm{W\,m^{-2}}$ (1.1 %) in the shortwave and $0.41\pm1.45\,\mathrm{W\,m^{-2}}$ (0.2 %) in the longwave irradiance. The histograms of the CRF for that day are shown in Fig. 9. The mean values of the entire flight section is $2.15\,\mathrm{W\,m^{-2}}$ in the shortwave and $0.41\,\mathrm{W\,m^{-2}}$ in the longwave. The slightly positive CRF might be caused by the upper air sounding approximately 300 km in the south of the flight track or the aerosol conditions (aerosol optical thickness was set to zero in the simulation). In addition to the measurement uncertainties of the used broadband radiometer (<3 %, Ehrlich et al., 2019b), the radiative transfer modeling can induce a bias (<2 %) in the shortwave wavelength ranges (Randles et al., 2013). Due to the absence of cloud-free conditions during other low-level flights of the ACLOUD campaign this comparison can be considered as a rough estimate of potential uncertainties during the whole ACLOUD campaign.

A special aspect of this dataset concerns the measurement strategy itself, whereby the irradiances are observed in the aircraft flight altitude (during low-level sections in average 80 m) and in various atmospheric thermodynamic profiles. Radiative transfer simulations of all available profiles during the campaign indicate that the derived total CRF in flight altitude may have an offset less than $\pm2\,\mathrm{W\,m^{-2}}$ compared to surface-related/surface-based observation, depending on the prevailing thermodynamic profiles. Details are given in the supplement provided to this study.

In this study, the retrieval of $\alpha_{cf}$ is applied only above homogeneous sea ice, conditions frequently observed during ACLOUD. In the MIZ, though, the heterogeneous sea ice and the correspondingly reduced surface albedo prevents an application of the original parametrization by Gardner and Sharp (2010). In the future, however, this might become possible by making use of the cosine weighted sea ice fraction $I_f$ and its linear relation to the albedo, whereby changes in surface albedo caused by the surface albedo-cloud interaction can be scaled to the prevailing $I_f$ and ocean albedo by assuming diffuse radiative transfer (Lambertian albedo) (not shown in this study).

The uncertainties in the retrieval of $\alpha_{cf}$ and the shortwave $F_{net,cf}$ depend mainly on the observed $\alpha_{all}$, as was investigated by applying synthetic albedo and LWP distributions to the lookup tables. Due to the non-linear increase of $\alpha_{all}$ with LWP, the potential error induced by uncertainties in the retrieved LWP is larger for lower LWP and depends on the prevailing surface types. The overall uncertainty in the cloud-free shortwave net irradiances above a homogeneous high surface albedo using the retrieved $\alpha_{cf}$ should be below 20 % and decrease with decreasing surface albedo.

## 6 Impact of surface albedo-cloud interaction on the CRF during ACLOUD

With the application of the methods described in section 5, the CRF during the ACLOUD campaign can be analysed with respect to the surface albedo-cloud interaction. The observed impact of clouds on the surface albedo was illustrated in Fig. 8 for one flight. A comparison of measured (all-sky) $\alpha_{all}$ and retrieved cloud-free albedo $\alpha_{cf}$ calculated for all low-level flights during ACLOUD over homogeneous sea ice ($I_f > 98\,\%$) is shown as frequency distributions in Fig. 10a. The broad distribution

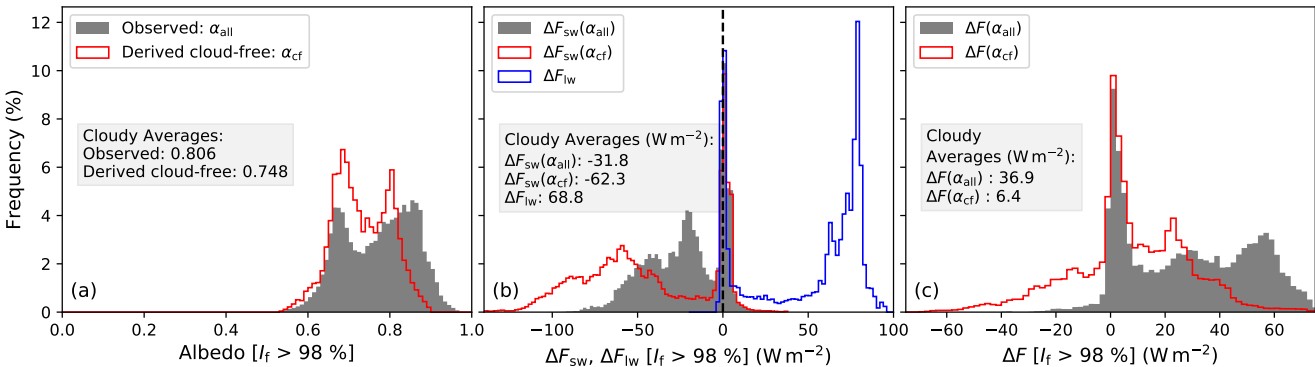

**Figure 10.** (a) Frequency distribution of the observed ($\alpha_{\mathrm{all}}$, gray) and cloud-free estimated ($\alpha_{\mathrm{cf}}$, red) surface albedo for all ACLOUD measurements obtained over homogeneous sea ice ($I_{\mathrm{f}} > 98\,\%$). (b) Terrestrial (blue) and shortwave CRF using the observed albedo ($\Delta F_{\mathrm{sw}}(\alpha_{\mathrm{all}})$, gray) and the shortwave CRF applying the estimated cloud-free albedo ($\Delta F_{\mathrm{sw}}(\alpha_{\mathrm{cf}})$, red). (c) The total (shortwave + longwave) CRF calculated with both albedo parameters is shown in panel (c). Average values for cloudy conditions (LWP $> 1\,\mathrm{g\,m}^{-2}$) are given in the embedded text boxes of each panel.

of observed albedo illustrates the seasonal transition of sea ice properties from a cold period end of May 2017 into the melting season in June 2017 (Wendisch et al., 2019; Jäkel et al., 2019). On average, the $\alpha_{\mathrm{all}}$ observed during cloudy conditions (LWP $> 1\,\mathrm{g\,m}^{-2}$) was about 0.8. The estimated cloud-free albedo for the same conditions gives an average value of 0.74, which is approximately 6 % lower than $\alpha_{\mathrm{all}}$. The distribution of $\alpha_{\mathrm{cf}}$ is slightly narrower than the measured one in all-sky conditions,

because the majority of cloud-free flight sections took place near to the end of the campaign with low values of surface albedo, and thus gives a lower bound to the distribution.

To illustrate the effects of the surface albedo-cloud interaction during the ACLOUD campaign, the CRF is computed using both the measured cloudy albedo ($\alpha_{\mathrm{all}}$) and the estimated cloud-free albedo ($\alpha_{\mathrm{cf}}$). Fig. 10b shows the frequency distribution of the shortwave CRF for both solutions, observed over homogeneous sea ice ($I_{\mathrm{f}} > 98\,\%$). The CRF based on the observed

albedo ($\Delta F_{\mathrm{sw}}(\alpha_{\mathrm{all}})$, gray bars) shows a bimodal distribution. The mode around $0\,\mathrm{W\,m}^{-2}$ represents cloud-free conditions and heterogeneous optically thin clouds, where 3D effects induce occasionally positive shortwave CRF values as reported in Wendisch et al. (2019). The broader mode between -60 $\mathrm{W\,m}^{-2}$ and -20 $\mathrm{W\,m}^{-2}$ characterizes cloudy conditions shaped by the prevailing LWP, SZA and surface albedo.

Applying the estimated cloud-free albedo ($\Delta F_{\mathrm{sw}}(\alpha_{\mathrm{cf}})$, red histogram in Fig. 10a), shifts the cloudy mode in the shortwave

CRF in Fig. 10b to more negative values, indicating a stronger cooling effect, which was already implied by the radiative transfer simulations in section 4. In total, the shortwave CRF using the observed albedo values $\alpha_{\mathrm{all}}$ shows on average a weak cooling effect of -32 $\mathrm{W\,m}^{-2}$ under cloudy conditions (LWP $> 1\,\mathrm{g\,m}^{-2}$). Applying the surface albedo for cloud-free conditions almost doubles the shortwave cooling effect to -62 $\mathrm{W\,m}^{-2}$. The non-linearity in the functional dependence of surface albedo and LWP (Fig. 4 and 8) spreads the frequency distribution of $\Delta F_{\mathrm{sw}}(\alpha_{\mathrm{cf}})$ (interquartile range 36 $\mathrm{W\,m}^{-2}$ instead of 26 $\mathrm{W\,m}^{-2}$

for $\Delta F_{\mathrm{sw}}(\alpha_{\mathrm{all}})$), while the mode for cloud-free conditions is not affected. These values hold for the ACLOUD observations with an average LWP of $58\,\mathrm{g\,m^{-2}}$ and a SZA of $61°$ during cloudy conditions over sea ice.

Under the same conditions the longwave CRF distribution (blue histogram in Fig. 10b) shows an even more distinct cloudy mode with a peak in the frequency distribution around the $78\,\mathrm{W\,m^{-2}}$ bin. During cloudy conditions (LWP $> 1\,\mathrm{g\,m^{-2}}$) $\Delta F_{\mathrm{lw}}$
averages to $69\,\mathrm{W\,m^{-2}}$ indicating a strong warming effect during this late spring early summer conditions.

The impact of surface albedo-cloud interaction becomes evident in the distribution of total (shortwave plus longwave) CRF (Fig. 10c), which shifts for cloudy conditions from a significant total warming effect of $37\,\mathrm{W\,m^{-2}}$ ($\Delta F(\alpha_{\mathrm{all}})$) over sea ice to an in average almost neutral effect ($6\,\mathrm{W\,m^{-2}}$) by applying $\alpha_{\mathrm{cf}}$. Also the distribution of the $\Delta F(\alpha_{\mathrm{cf}})$ indicates that already when the $\alpha_{\mathrm{cf}}$ dropped approximately below 0.75 (mid of June), the cooling effect was dominant, meanwhile the $\Delta F_{\mathrm{sw}}(\alpha_{\mathrm{all}})$)
was positive throughout the campaign. Considering that the predominant surface type of the campaign was still sea ice covered by snow, the transition from a warming to a cooling effect of clouds could already start early in the season, even before the formation of melt ponds and the rapid decline in surface albedo (Fig. 6), which underlines the potential impact of surface albedo-cloud interactions.

## 7    Summary and conclusions

To estimate the warming or cooling effect of clouds on the Arctic surface from observations or models, a precise characterization of the cloud-free state is required, which serves as a reference. Although the radiative cloud-free state constructed from cloudy observations remains an apparently theoretical construct with freedom of interpretation and definition, there are several relevant aspects for the characterization of the CRF in the Arctic, which are listed in the points below.

- In the transition region between open ocean and closed sea ice (the MIZ), the thermodynamic state of the atmosphere
changes on horizontal scales of a few kilometers (Lampert et al., 2012), which influences the cloud-free reference state and the resulting simulated radiative field. To obtain reliable estimates of CRF along meridional air mass transports into and out of the Arctic, such as warm air intrusions or cold air outbreaks (Tjernström et al., 2015; Pithan et al., 2018; Tjernström et al., 2019), a high temporal and spatial resolution of thermodynamic profile measurements along the trajectory close to the MIZ is required. In this paper we could prove the importance of this effect using observations
collected during a warm air advection case in early summer showing relatively weak related thermodynamic changes, but a considerable impact on the estimate of CRF.

- Variability in sea ice concentration is closely linked with fluctuations in surface albedo. The derivation of downward shortwave irradiances under cloud-free conditions in heterogeneous surface albedo conditions requires an estimate of the effective areal average surface albedo, determining the multiple scattering on large spatial scales (e.g., Kreuter et al.,
2014). For the airborne observations collected during ACLOUD, we illustrate that moving average filters with shapes appropriate to reproduce 3D radiative transfer need to be applied to obtain values of shortwave CRF considering horizontal surface albedo inhomogeneities appropriately.

- The transition between cloudy and cloud-free atmospheric states is accompanied by changes in the radiative transfer, affecting the surface albedo, and the CRF. In the available CRF studies in the Arctic, either observations during cloud-free periods have been used to extrapolate the expected cloud-free surface albedo during cloudy periods, or the surface albedo observed in cloudy conditions have been used. However, as the snow and ice albedo depends on parameters like snow grain size, prevailing SZA, and cloud optical thickness, the available approaches only insufficiently represent the cloud-free albedo in the cloudy Arctic.

- Changes in shortwave surface albedo with increasing cloud optical thickness are considerable and directly impact the shortwave net irradiances, and thus, the estimate of shortwave CRF. Combining spectral snow surface albedo models with atmospheric radiative transfer simulations enables us to characterize the two processes involved in spectral surface albedo-cloud interactions and to assess the importance of these effects on shortwave CRF as a function of cloud LWP and SZA for three common surface types in the Arctic (fresh snow, melting snow and sea ice). The spectral weighting effect of downward irradiance appears to be dominant for snow surfaces and enhances the cooling effect of clouds at the surface. For the second process, a change from mainly direct radiation in the cloud-free state to rather diffuse radiation in the cloudy state, the sign of the modification depends on SZA, cloud optical thickness, and the melting state of sea ice.

- For the ACLOUD campaign, characterized by snow on sea ice in the beginning melting season, the averaged shortwave CRF estimate over homogeneous sea ice of -32 $\mathrm{W\,m^{-2}}$ (cooling) almost doubles to -62 $\mathrm{W\,m^{-2}}$, when surface albedo-cloud interactions are taken into account by using the proposed retrieval of cloud-free albedo from cloudy observations. Due to this consideration, the campaign averaged total (shortwave plus longwave) CRF is shifted from a mainly warming effect of clouds over sea ice to an almost neutral effect, for the ACLOUD observations with relatively small SZA.

- The observed surface albedo trend during the ACLOUD campaign (Fig. 6) induces a transition of the CRF from a warming to a cooling already for snow covered surface types, and thus, earlier in the season than reported during SHEBA. In addition, the instantaneous longwave CRF approach might additionally induce an overestimate of the warming effect (section 2) potentially shifting the total CRF further to cooling. This indicates a possible extension of the period in which clouds cool the surface and highlights the impact of surface albedo-cloud interactions and a required reassessment of the CRF in the Arctic.

Long-term measurements, such as those performed during the SHEBA campaign or currently within the Multidisciplinary drifting Observatory for the Study of Arctic Climate (MOSAiC) expedition (www.mosaic-expedition.org), with an appropriate instrumentation and radiative transfer modeling will be required to quantify these effects and their potential seasonal dependence by continuously estimating the cloud-free albedo in cloudy conditions. The proposed method to estimate the surface albedo in cloud-free conditions using the parameterization from Gardner and Sharp (2010) can be easily applied to common Arctic long-term observations above snow and ice surface types, especially if high quality LWP measurements are available.

Besides observations, global climate models estimate the cloud radiative feedback based on the impact of clouds on the surface REB, for which the surface albedo is fundamental. For specific surface types, often fixed values of shortwave surface

albedo are assigned and parameterized using surface temperature. However, these simplified parameterizations are not appropriate to accurately describe surface albedo-cloud interactions. The use of parameterizations accounting for these effects, such as that of Gardner and Sharp (2010), are necessary and highlight the need for coupled surface atmosphere models including representative surface microphysical properties. The shortwave net irradiances depend not alone on cloud transmissivity and surface albedo, moreover the interaction between both needs to be represented.

Further effort in coupled surface atmosphere radiative transfer modeling with a representation of common surface albedo types, like the ones from melt ponds in the Arctic, are required to track the seasonal cycle of shortwave CRF. Spectral albedo observations combined with the common broadband devices will help to account for the spectral features in surface albedo and trace changes in SSA. The proposed approach of reproducing the cloud-free albedo can not adequately reflect the diversity of spectral surface albedo types and issues related to the surface albedo-cloud interaction, especially in summer.

Considering the surface albedo-cloud interaction in global climate models and upcoming long-term observations such as MOSAiC will further improve our understanding of CRF and cloud radiative feedback in the Arctic environment and its role for Arctic amplification.

*Data availability.* The pyranometer and pyrgeometer broadband irradiance and KT-19 nadir brightness temperature from AWI aircraft Polar 5&6 during the May to June 2017 ACLOUD campaign are published on the PANGAEA database (Stapf et al., 2019a). The retrieved quantities of CRF, LWP equivalent and cloud-free albedo are available on the PANGAEA database (Stapf et al., 2019b). Air temperature, relative humidity and pressure in situ profiles from both aircraft are used from (Hartmann et al., 2019). Polar 5 Dropsondes: (Ehrlich et al., 2019a). Calibrated fisheye camera images: (Jäkel and Ehrlich, 2019; Jäkel et al., 2019). Radiosoundings from Polarstern (Schmithüsen, 2017) and Ny-Ålesund (Maturilli, 2017a, b).

## Appendix A: Transmissivity-based retrieval of LWP equivalent

The cloud transmissivity is defined by the ratio of measured $F_{\mathrm{sw,all}}^{\downarrow}$ and the simulated cloud-free $F_{\mathrm{sw,cf}}^{\downarrow}$ downward irradiance:

$$\mathcal{T} = \frac{F_{\mathrm{sw,all}}^{\downarrow}}{F_{\mathrm{sw,cf}}^{\downarrow}}. \tag{A1}$$

$\mathcal{T}$ can be converted into cloud optical thickness or LWP, however, it is important to account for the surface albedo dependences due to multiple scattering. The $\mathcal{T}$ for a cloud with the same microphysical properties over snow and ice is higher compared to over open ocean, where the majority of photons will be absorbed by the surface and are not available for new back-scattering events of the upward irradiance in the cloud towards the surface. Taking this dependence into account, the broadband $\mathcal{T}$ is used to derive the cloud optical thickness similar to approaches by Leontyeva and Stamnes (1993), Fitzpatrick et al. (2004), and Fitzpatrick and Warren (2005).

Lookup tables of $\mathcal{T}$ for a range of surface albedo between 0 and 1 and LWP between 0 and $320\,\mathrm{g\,m^{-2}}$ are simulated for the local solar zenith angle and thermodynamic profile and subsequently compared to the values derived from the observations

along the flight track. In the simulations, vertically homogeneous pure liquid water clouds are assumed to limit the complexity of the simulations. In the following, therefore, the LWP is referred to an equivalent LWP, because no ice water content is assumed. The cloud is located between 400 m and 600 m with a fixed $r_{\text{eff}}$ of 8 μm, typical for Arctic clouds in this season and region (Mioche et al., 2017). These rather crude assumptions result in uncertainties of the simulated irradiance, which were

quantified by Leontyeva and Stamnes (1993) and Fitzpatrick et al. (2004) as a function of surface albedo, SZA, $r_{\text{eff}}$ and cloud optical thickness.

Similar to the simulations of $F_{\text{sw,cf}}^{\downarrow}$ for heterogeneous surface albedo fields, an effective albedo, which influences the local scattering processes in cloudy conditions needs to be considered in the retrieval simulations of $\mathcal{T}$ (Pirazzini and Raisanen, 2008).

The diversity of potential 3D effects induced by surface and cloud heterogeneities in the MIZ omit a specific solution for the smoothing problem of the areal-averaged effective albedo and can only partially be depicted by radiative transfer modeling. To make the retrieval applicable to ACLOUD measurements and reduce the uncertainties induced by horizontal photon transport, a commonly observed cloud/surface scene, with a cloud base height of 200 m and leads with different sizes, are simulated using 3D radiative transfer (not shown here). The estimated kernel $k$ is based on a Cauchy distribution:

$$k(x, \mu, \gamma) = \left( \pi \cdot \gamma \cdot \left[ 1 + \left( \frac{x - \mu}{\gamma} \right)^2 \right] \right)^{-1}, \tag{A2}$$

with $\gamma$ of 400 m, the median $\mu$ and a scale $x$ of 10 km. The horizontal extent is, as expected, smaller compared to the cloud-free kernel introduced in Fig. 7b, due to the low cloud base height limiting the free photon path length. Applied to the 3D modelled (theoretically observed) albedo the simulated 1D irradiance adequately reproduces the results obtained from the 3D output, and thus, reduces for these cloud/surface scenes the uncertainties of the retrieved LWP considerably.

Nevertheless, multiple scattering, changes in cloud base height (Pirazzini and Raisanen, 2008) and 3D radiative effects due to inhomogeneous cloud/surface scenes, might induce large uncertainties in this retrieval. However, the observed $I_{\text{f}}$ statistics indicate that the majority of ACLOUD flights were conducted over a rather homogeneous surface, where the discussed issue is of minor importance. The sensitivity of the retrieval is in general higher over open water compared to over ice, since changes in $F_{\text{sw}}^{\downarrow}$ with increasing LWP are more pronounced. The relative uncertainty of this retrieval for homogeneous clouds and surfaces

can be expected to range between 15 % and 35 % over open ocean and sea ice, respectively.

The conversion from LWP to optical thickness ($\tau$), as required for the parameterization by Gardner and Sharp (2010), is applied by,

$$\tau = \frac{9}{5} \cdot \frac{\text{LWP}}{\varrho_{\text{w}} \cdot r_{\text{eff}}}, \tag{A3}$$

with the density of liquid water $\varrho_{\text{w}}$ and the simulated $r_{\text{eff}}$.

*Author contributions.* All authors contributed to the editing of the manuscript and to the discussion of the results. JS drafted the manuscript and initialized the study. JS processed the radiation data, merged the data sets and performed the radiative transfer simulations. EJ contributed to the radiative transfer simulations and their interpretation. MW, AE and CL designed the experimental basis of this study.

*Competing interests.* The authors declare that they have no conflict of interest.

5    *Acknowledgements.* We gratefully acknowledge the funding by the Deutsche Forschungsgemeinschaft (DFG, German Research Foundation) – Project Number 268020496 – TRR 172, within the Transregional Collaborative Research Center "ArctiC Amplification: Climate Relevant Atmospheric and SurfaCe Processes, and Feedback Mechanisms (AC)³". The authors are grateful to AWI for providing and operating the two aircraft during the ACLOUD campaign. We thank the crews of Polar 5 and Polar 6, the technicians of the aircraft for excellent technical and logistical support. The generous funding of the flight hours for ACLOUD by AWI is greatly appreciated. Observations in Fig. 6 were made

10    by the SHEBA Atmospheric Surface Flux Group, Ed Andreas, Chris Fairall, Peter Guest, and Ola Persson. Data provided by NCAR/EOL under the sponsorship of the National Science Foundation (https://data.eol.ucar.edu/).

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
