# Peer review of "Reassessment of shortwave surface cloud radiative forcing in the Arctic: Consideration of surface albedo – cloud interactions"

_Atmospheric Chemistry and Physics, 2019_

## Referee Comment (RC1) · Anonymous Referee #1 · 21 Aug 2019

Major revisions to this paper are needed prior to publication.

In this paper, cloud radiative forcing (CRF) and the surface radiation energy budget (REB) are calculated for measurements along the Arctic springtime marginal sea ice zone. The authors investigate and quantify the effects on CRF and REB of (A) using incorrect atmospheric thermodynamics (based on geographic mismatch between cloud and clear-sky measurements), (B) using the local surface albedo instead of an areal-averaged albedo, and (C) accounting for the impact of clouds on the albedo.

[Figure]

This paper includes novel data and is worthy of publication. The application of (A)-(C) above may also have novel elements.

Main criticisms of this paper:

1) Additional referencing of the literature is needed to put the paper into context and understand what is novel in this work.

2) An uncertainty analysis is needed.

3) Some of the writing is confusing and needs to be edited for clarity.

4) There are gaps in the descriptions of the measurements and methods that make the work difficult to understand or reproduce.

Details follow.

Title: The title should be reworded. Perhaps something like, "Cloud radiative forcing at the Arctic springtime marginal sea ice zone derived using low-level airborne observations."

Language (overall): The authors should go through the paper and make sure every paragraph makes a single, clear point. They should also go through each sentence and make sure that it is correct, comprehensible, and stated as simply as possible. I suggest asking a colleague to read the paper and then working with them on how to clarify anything that they do not understand. Here are just a few examples:

- Abstract lines 1 and 2: "warming or cooling effect . . . on the radiative energy budget." Clouds do not cool the energy budget. Please restate.

- First paragraph of the introduction: This paragraph is difficult to understand, has a lot of unnecessarily detail, and only provides general motivation. I think a few sentences can explain that clouds are important for the Arctic. What is really needed prior to the second paragraph is more specific motivation for this work, including why estimates of CRF and REB are important and how they are used and calculated in the literature

(discussed more below).

- Page 2 lines 18-22: these sentences are a distraction from the rest of the paragraph, which focuses on SZA and albedo.

- When the words heterogeneous or heterogeneity are used, the authors need to state what is varying – for heterogeneous sea ice is it type of ice? Or ice fraction? What is meant by heterogeneous albedo?

Abstract and Introduction: Is the most important result the application of the parametrization of Gardner and Sharp (2010) to measurements? If this is the case, this should be clear in the abstract and introduction. For example, the authors state in the introduction that, "Both processes have been parametrized, for example by Gardner and Sharp (2010) based on simulations, however, their impact on estimates of the CRF in the Arctic have not yet been evaluated." They should go on to state in the following paragraph that they apply these parametrizations in this work.

Introduction: Although the paper has a long reference list, missing are examples from the literature of calculations of CRF and radiative fluxes based on observations in the Arctic. Context of the literature is also needed to show what ideas or parametrizations are novel here. For example, if Eqs (7) and (8) are novel, please make that clear in the introduction as well as in Sect. 3, as well as how they relate to calculations of CRF in the literature. The authors should explain how CRF is used in such studies, what are the shortcomings, and how their work addresses these shortcomings. Some possibilities:

Intrieri, J. M., Fairall, C. W., Shupe, M. D., Persson, P. O. G., Andreas, E. L., Guest, P. S., & Moritz, R. E. (2002). An annual cycle of Arctic surface cloud forcing at SHEBA. Journal of Geophysical Research: Oceans, 107(C10), SHE-13.

Dong, X., Xi, B., Crosby, K., Long, C. N., Stone, R. S., & Shupe, M. D. (2010). A 10 year climatology of Arctic cloud fraction and radiative forcing at Barrow, Alaska. Journal of

Geophysical Research: Atmospheres, 115(D17).

Sedlar, J., Tjernström, M., Mauritsen, T., Shupe, M. D., Brooks, I. M., Persson, P. O. G., ... & Nicolaus, M. (2011). A transitioning Arctic surface energy budget: the impacts of solar zenith angle, surface albedo and cloud radiative forcing. Climate dynamics, 37(7-8), 1643-1660.

Hartmann, J. , Kottmeier, C. , Wamser, C. and Augstein, E. (2013). Aircraft Measured Atmospheric Momentum, Heat and Radiation Fluxes Over Arctic Sea Ice. In The Polar Oceans and Their Role in Shaping the Global Environment (eds O. M. Johannessen, R. D. Muench and J. E. Overland). doi:10.1029/GM085p0443.

Cox, C. J., Walden, V. P., Rowe, P. M., & Shupe, M. D. (2015). Humidity trends imply increased sensitivity to clouds in a warming Arctic. Nature communications, 6, 10117.

Södergren, A.H., McDonald, A.J. & Bodeker, G.E. Clim Dyn (2018) An energy balance model exploration of the impacts of interactions between surface albedo, cloud cover and water vapor on polar amplification, 51: 1639. https://doi.org/10.1007/s00382-017-3974-5

Measurements: More information is needed about the measurements. What wavelengths are used? (Also, please use "longwave" and "shortwave" instead of "terrestrial" and "solar." Presumably the aircraft had both up-looking and down-looking pyrgeometer and pyranometers? The text refers to the cloudy ABL, but I think data are only used where the cloud was above the aircraft? (E.g. all upwelling flux measurements were for clear skies). Please clarify this. How did you ensure that there was no cloud around or below the aircraft? Please also provide more detail about measurements of atmospheric thermodynamics, and a table showing the various measurements (with time and location) as context. Finally, Page 2, line 11 implies that this work uses all-sky minus clear-sky (other definitions of CRF use cloudy-sky minus clear sky). Were all downwelling flux measurements of cloudy sky?

Radiative transfer simulations: Sufficient information is needed here that the results are reproducible. What was the vertical resolution? How were the measurements (dropsonde, radiosonde, and surface) merged? What was used for concentrations of other trace gases (most notably for the longwave calculations, $CO_2$). It would be helpful to specify which flux and albedo terms were calculated with the various models (longwave, shortwave and 2D vs. 3D). What are the uncertainties for the radiative transfer calculations?

Section 3.2: A lot of work has been done on the longwave CRF that is relevant here. For example, Cox et al 2015 (listed above) examines temperature and humidity.

Uncertainty analysis: A variety of assumptions are made in this work and the calculations and measurements all have associated errors and uncertainties. An uncertainty analysis is needed.

Technical details:

- I_f needs to be defined before the first time it is used.

---

## Referee Comment (RC2) · Anonymous Referee #2 · 21 Aug 2019

Summary In this study the authors argue for corrections in the surface albedo they suggest are necessary for the calculation of the Cloud Radiative Effect in the Arctic, where the surface albedo is very variable on all time and space scales. It is an interesting study but it needs some rethinking before it is pub-lished. Therefore, I recommend that this manuscript is accepted after major revision.

General comments In summary, I think this is interesting work, but it needs a more solid foundation and a discisson about the why's and the how's, and less details on ACLOUD; maybe do a separate but more exten-sive paper on CREs during ACLOUD,

referencing this paper.

Maybe I'm nitpicking, but the terminology has been changed by the climate community, from Cloud Radiative Forcing, or CRF, to Cloud Radiative Effect, or CRE, quite a while ago. The title is also in my personal opinion too long and clunky; try something shorter. Maybe "Interactions be-tween surface albedo and clouds for Arctic cloud radiative effect estimation".

The content in the study balances along many borders and as a consequence it doesn't quite fulfill any of the topics it crosses well enough. It is unclear if this is a theoretical study that uses ACLOUD data just because its good and convenient or if it is a con-tribution to ACLOUD as such. I already hear the authors say "can't it be both?" and my response is it would be a better radiation interaction paper if ACLOUD was tuned down and a better ACLOUD paper if the radiation stuff was more background and the actual results where more detailed. Typically to be a good paper on both aspects it would have to be longer – which is not good. There is also a lot taken for granted on the readers; not everyone is a radiative transfer modeling expert. So choices must be made.

Besides the use of an old terminology (CRF instead of CRE) there needs to be a much more in-depth and philosophical background. Why are we interested in CRE (or CRF) and how does that impact how we do these calculations? The question "How does the clouds affect the surface energy budget" is not the same as the question "How would the surface energy budget look if the clouds were not present?". As an exam-ple, the authors argue both that changes in surface broadband albedo between cloudy and clear states must be considered, and that the details in thermodynamic are im-portant. However, we know that the thermodynamic profiles for clear and cloudy cases are very dif-ferent. The presence of clouds depends on the vertical profiles, but the clouds themselves also modi-fy the profiles by their presence. Yet the authors argue that we change the surface albedo to how would be without the clouds, but keep the thermodynamic profiles as they are; only remove the cloud water. I think that the an-

Interactive
comment

swers depend on what we want to use this metric for. Ideally the an-swer to my second question above would have us examine the conditions in clear and cloudy condi-tions separately, not modifying cloudy cases by removing the condensed water. However, the Arctic is a very cloudy place and there are not enough clear cases to make this possible. Therefore, I think this paper needs a much more detailed introduction and background to what we are trying to do and why.

There are three corrections that the authors argue are necessary: 1) Effects of proper thermody-namic profiles; 2) Effects of heterogeneous surfaces, and; 3) Effects of the clouds on the characteris-tics of the solar radiation.

1) This is pretty obvious; of course one must use the profile from the same location as the CRE is considered. In this paper this is discussed in the context of aircraft observa-tions covering an area, but the conclusion is also important for fixed-point observations. One question is, if a proper sounding at the location is not available, from how far away can it be used? This question of course has no an-swer other than "it depends." But here one could also raise the issue of cloudy profiles being differ-ent from the corre-sponding clear case; in other words, if we could magically remove the cloud water so that the clouds vanish, what would that do to the thermodynamic profiles? Or, are the profiles found in clear conditions systematically different from those in cloudy condi-tions? From both mod-eling and observations in subtropical stratocumulus regions we know that the moist PBL is deeper and warmer when clouds are present; for the Arctic we don't really know, but I would wager a bet that they are different!

2) That heterogeneous surfaces poses a problem for upwelling shortwave radiation is also pretty obvious. This is a main factor in the MIZ but also in the pack ice mainly due to melt ponds. This pa-per doesn't even mention the effect of melt ponds, presum-ably because there weren't any during ACLOUD. So is this an ACLOUD paper usuing advanced radiation methods or a radiation paper using ACLOUD observations?

Moreover, to this reviewer it is not obvious that the downwelling radiation is dependent

on sur-face albedo; in most NWP or climate models that I know this is not considered, but may be ignorant. Either way, for a reader like me, this needs to discussed in more detail. Commenting about "horizon-tal photon transport" is not sufficient. If this is a factor, how large is it? What is it caused by? Are there differences between say the MIZ, with alternating ice and open water, pack ice with melt ponds, pack ice with open leads, or pack ice with many substantial pressure ridges? Or all of the above?

3) Changes in the spectral composition from absorption in clouds is a real tangible ef-fect that one can discuss if it is necessary to compensate for or not; see my discussion above. Changes due to the different distribution between direct and diffuse radiation is trickier. Also the cloud albedo is sensi-tive to solar zenith angle.

Finally, the language is mostly OK, but occasionally I stumble on unnecessarily difficult wording, for example "exemplarily" in the context it is used is an existing word but even the dictionary indi-cates it isn't much used in modern English. There are also past/present inconsistencies; what is done and presented in this paper is sometimes described in past tense and sometimes in present. Either is fine with me; just be consistent. Finally, final: among the data made available here, only a subset is actually really made available; the rest is just referenced.

Some detailed comments Title is unnecessarily clunky – also here and throughout the paper, Cloud Radiative Forcing (CRF) should be replaced by Cloud Radiative Effect (CRE); even maybe surface CRE.

Page 1, line 23: Comma after amplification.

Page 2, line 1-2: If this is the prime question, one can not use the cloud profile minus the cloud as representative for clear conditions; one must do the clear and the cloudy cases completely separately.

Page 2, line 3: It is not at all clear that clouds are cooling the surface in the Arctic in summer, so I would drop "dominates". It depends on a lot of factors, some of which

this paper deal with. A clear case when this statement is correct, perhaps the only one observed case, is for SHEBA; the only annual observations that exist and BTW where the CRE was calculated without any of the corrections discussed here. Suffice it to say that I've seen summer conditions with a lot of snow and almost no melt ponds at very high latitudes where surface temperatures plummets when the clouds dissipate.

Page 2, line 11: The cases are "clear" or "all-sky", so I would swap places between "all-sky" and clear here. You can still define that as "cloudy" from her on, but it should be stated that the normal case is the existing clouds; not just when it is completely overcast.

Page 2, line 20: Here it is stipulated that surface albedo affects the incoming (down-welling) solar radiation for clear skies. To me that is not obvious, and even if it is obvious from multiple refelctions in clouds it is not obvious that it is important for clear-sky radiation. Do spend some more time on this please.

Page 2, line 14-15: "... observations of ... conditions and of atmospheric thermody-namic state."

Page 2, line 26: Is shape the right word here? Isn't it the magnitudes; not just the shape?

Page 2, line 28: Comma after "albedo".

Page 3, line 17-18: Here's an example of tense mismatch: "... were investigated in this paper" and "... aircraft is displayed ...". Later on same page the dataset "is" merged and on line 3-4 on the next page "... concentration was calculated ...".

Paragraph staring on Page 6, line 29:How do you handle the observed surface albedo when calcu-lating CRE (or surface ïĄĎF)? The text says that the clear-sky albedo is set to 80% and the zenith an-gle to 80°; presumably those were not the observed conditions?

Section 3.3: Spend some time explaining why albedo affects downward solar radiation.

Also, ex-plain the choice of albedo filter; is there any theoretical consideration here or was it "trial and error"? Paragraph starting at Page 9, line 14: It seems to me this is a test comparing calculated results to observed results from a clear day; correct? If so, maybe this should not be reported under this head-ing? And maybe the term CRF (or rather CRE) should not be used when there are no clouds and the CRE is expected to be zero?

Page 10, line 6: Explain SSA.

Page 10, line 9: I don't think you could find a case with 1 cm snow thickness in reality. That would be > 1cm at some location and no snow at other.

Paragraph starting at Page 10, last line: This should come before the calculation spe-cific. First ex-plain why and then how.

Page 11, line 3-4: Why is diffuse radiation coming in at a zenith angle of $\sim50°$? When the cloud is thick enough that where the sun is in the sky can no longer be determined, is there a zenith angle at all? I thought, but may be wrong, that diffuse mean precisely that the radiation was equally strong in all directions.

Page 11, line 18: Drop "the".

Page 12, line 17: ". . . with increasing LWP is not, or only poorly, parameterized. . ."

Page 12, line 18: Unclear past tense in "have been used". Previously, or did you do this work now. In the previous case, give reference; in the latter, present tense should be used.

Figure 8: The SHEBA albedo line includes melt ponds and eventually even a lead. The drop in albedo starting at the beginning of June is due to this; as there were no melt ponds I ACLOUD(?), your comparing apples and pears here. For the SZA calculations, did you take into account that SHEBA moved northward during the year? Another idea would be to redo the SHEBA Intrieri et al. CRE study with this new information. Maybe a bit more work than anticipated for now, but it would be interesting.
Page 13, line 2 & 3: Again, two examples of past-tense confusion. When was this done; for this paper or by an earlier investigator.

Page 13, line 10: Using both "indicate" and "might" in the same sentence almost obliterates the conclusion.

Page 14, line 4: Can't find any red line in Figure 8.

Page 17, line 5: "indispensable" is a strong word. Since it is impossible to know the cloud-free state with any accuracy, I would mellow the language here. If something indispensable is also im-possible, then why even try?

Page 17, line 11: If by "local" you mean in one single specified point, then I'm confused. The lo-cal albedo is what it is; it is different at a different locale when the sea ice is variable; I still get hung up on this concept. If you are referring to the effects on the cloud free downwelling radiation, that I wanted to have elaborated on, the at least write "local cloud-free albedo".

Page 18: lines 27-31: Only part of the data is available, a large chunk is only cited. Why? Appendix A: OK; but, why don't show that this works, using aircraft passages over Polarstern, where you have both transmissivity and LWP?

---

## Referee Comment (RC3) · Anonymous Referee #3 · 21 Aug 2019

The manuscript "Reassessment of the common concept to derive the surface cloud radiative forcing in the Arctic: Consideration of surface albedo – cloud interactions" by Stapf et al. describes the calculation of cloud radiative forcing (CRF) from measurements collected from aircraft over the MIZ in the eastern Arctic near Svalbard during the ACLOUD campaign in June 2017. The authors leverage the spatial nature of their measurements over the heterogeneous surface albedo that is characteristic of the MIZ to identify and correct biases in calculations of CRF under such conditions. The targeted biases are specifically those associated with cloud-surface albedo interactions

when the estimates of the shortwave clear-sky surrogates used in the CRF calculation are derived using radiative transfer models. The authors focus on a relevant problem that is suitable for ACP and this problem has received less direct attention from previous studies than analogous problems in the infrared. However, I respectfully disagree that this as a "reassessment" of CRF calculations, as the influence of surface albedo on CRF has been acknowledged previously and managed in various ways. The focus on albedo is warranted here because the observational platform and environmental conditions discussed make the present work particularly sensitive in that regard but I think the advancement promised by the title is overstated. I would be more comfortable with the paper being presented in either of the following ways: (a) As calculations of CRF in the MIZ during ACLOUD with the albedo work being an important, though incidental component. In this case more work or more detailed explanations of the longwave calculations are needed (see comments below). (b) If the authors wish to focus on the shortwave, they should drop the longwave data altogether and pitch the study as a proposed methodology for calculations of shortwave CRF in the MIZ where treatment of clear-sky shortwave fluxes requires special attention. In either case, the study needs to be more carefully contextualized and motivated by referencing previous work. More details and comments are provided below. I also suggest a thorough copy editing for grammar, typos, missing words or letters, etc., of which there are many to be found.

Major Comments:

(1) The introduction and study motivations need substantial improvement. Some portions of the introduction actually belong in the Methods section. More troubling is that the study promises to improve upon (indeed, to "reassess") surface-based observations of CRF without actually referencing a single example of previous work on this subject, which has developed for several decades in the Arctic. I suggest rewriting the introduction to more clearly contextualize the present work in the existing literature. I have included some (not exhaustive) useful references throughout this review.

[Figure]

(2) The title indicates a focus on shortwave processes. In my opinion is a fair representation of main the scope of the work, yet there are sections devoted entirely to the longwave and total CRF. There are complications in CRF calculations that are specific to the longwave (Allan et al. 2003) which are analogous to the issues affecting the shortwave; e.g., lapse rate and surface temperature responses to clearing skies (Long and Turner 2008) and systematic differences in water vapor between skies that are clear and those that are cloudy ("water vapor CRF", e.g., Dong et al. 2006). These issues are ignored and consequently the longwave and total CRF parts of the manuscript are somewhat confusing and do not serve a clear purpose. There is quite a lot packed into this study already. I think the study would be much clearer if only shortwave data were included, in keeping with the advertised focus of the work.

(3) Ehrlich et al. (2019b) (P3L30) is in review and the DOI provided is unreachable. Thus, I cannot evaluate the processing of the radiometric data, which is central to this study. Indeed, I don't even know what equipment was used. I wish to know more in particular because radiometric data from airborne platforms requires additional processing, though I am aware that the authors are familiar with some of these complexities (e.g., Ehrlich and Wendisch 2015). In addition to the instrument response corrections of the aforementioned work, how did you correct for tilt in the pyranometer? How did you correct the pyrgeometer data (measured at altitude) to represent the value that would be observed at the surface? (I think the answer is you did not [P7L9]). I have a similar question about the KT-19, which does not observe thermodynamic temperature of the surface, but rather a brightness temperature relative to the FOV and dependent on the path to the target. Is there any reason to similarly correct the shortwave data for altitude given that such details are the focus of the present work?

(4) Unless I misunderstand something, I believe there are errors in the presentation of the CRF equations. This is simple to correct if it is merely a typo in the subscripts, but if the equations were applied as stated the study's results could be impacted. Specifically, be careful how you use the terms "all sky" and "cloudy sky" because they are

not equivalent in CRF nomenclature. CRF may be defined as either (all – clear) or as CF(cloudy – clear) where CF is the cloud fractional occurrence and the other terms refer to net radiative fluxes for "all" (clear and cloudy sky conditions together), "cloudy" (only times when clouds are present) and "clear" (clear skies). Ramanathan et al. discuss both definitions. Admittedly, they are confusing on the nomenclature themselves in using the term "cloudy" early on in discussing their Eqs (1) and (2), but their meaning becomes clear when they introduce Eq. (4). Your Eq. (2) is therefore incorrectly stated; it shows the maximum CRF (e.g., Intrieri et al. 2002). The "cld" subscript needs to be "all" or the entire right side of the equation needs to be multiplied by the cloud fraction. For your purposes, and for the Arctic in general, I suggest the former. This comment applies to equations throughout the text.

(5) P7L9: I do not agree that this is a good assumption. About 70% of the downwelling longwave at the surface originates from atmosphere below the altitude of your aircraft (Ohmura 2001). Your assumption is plausibly (not certainly) valid if the atmosphere is isothermal between surface and the base of the cloud. While this condition could be met, in your case studies (Fig. 2), it is not. Your observations of flux at the altitude of the aircraft should be corrected to represent the surface.

(6) P8L11-P9L7: The approach you suggest to achieve a downwelling clear-sky shortwave is intriguing, but more information is needed for future studies to adopt your method. As written, it is not reproducible and there is no information on the sensitivities of the estimation; for example, I would expect that a filter of constant width assumes that leads are randomly distributed and roughly of the same size. I would also like to know more about the justification for your choice of a Laplace distribution as the most appropriate filter for this application.

Specific Comments

(1) Given that your upwelling shortwave is observed from an aircraft platform, the FOV covers and enormous area. I therefore do not understand why your albedo measure-

ments (e.g., Fig 3a) are not implicitly area-averaged, even if observed from a relatively low altitude (e.g., Podgorny et al. 2018).

(2) P2L18: Consider using "L" for longwave instead of "t" for terrestrial to avoid confusion with the terrestrial surface. "S" would then represent "shortwave" rather than "solar".

(3) P2L18: It is true that RT simulations are a common approach, but there are other approaches as well. Long and Ackerman (2000) present a method for estimating clearsky fluxes that implicitly accounts for the albedo dependencies on sky conditions. (Refer also to Dong et al. (2010) and Long (2005)). More in line with your study, Miller et al. (2015) parameterized clear-sky albedo for their RT simulations, though your situation is considerably more complex with regard to surface cover. Other studies have analyzed the dependencies. These studies do not necessarily detract from your work here and in some ways maybe motivate it, but either way really need to be referenced.

(4) P4L15 - P5L4: (1) I don't understand how (or when) you combined the dropsondes and NYA radiosoundings. (2) You do not say how you represent the atmosphere above the height of the soundings; this is necessary (even if estimated using a standard atmosphere) to a reasonable effective TOA (say, 60 km). (3) You do not say how you represent atmospheric gases that are radiatively active in the infrared, but were not measured by the sounding (not notably, $CO_2$, but also $O_3$, methane, etc.).

(5) P5L14-16: I do not agree that the upward longwave between clear and cloudy conditions is equal, but I doubt that this is what you actually mean to say. I think you mean you defined them to be so because (a) you do not account for the response of the atmospheric lapse rate (and surface skin temperature) to changes in sky cover and your calculations for the longwave are therefore "instantaneous" CRF (e.g., Miller et al. 2015), and (b) that you also neglect the influence of differences in the amount of longwave reflected from the surface between clear and cloudy skies. It is acceptable to make the first assumption (see Allan et al. 2003), but you should include the emissivity

term and then proceed with your CRF calculation. See my next comment.

(6) P4 Eq. (4): This variable is more frequently referred to as (longwave) "cloud radiative effect" (CRE) (e.g, McFarlane et al. 2013; Viudez-Mora 2015; Cox et al. 2015, 2016) and should be distinguished somehow from CRF. At the surface CRF and CRE are different, the former being the difference in the net fluxes and the latter being the difference in the incident fluxes. Confusion sometimes arises because the terms are frequently used interchangeably in satellite studies, being that the terms are equal against the backdrop of space.

(7) P5L28: Multiple scattering also depends on the albedo of the sky. How do you account for this?

(8) P6L30: Is 60deg SZA representative of the flight conditions?

(9) P9L20: You might also consider that your pyranometer is at best a 2% instrument and thus you might expect uncertainty of around 10 Wm-2 in the measurement. Thus, Figure 4 looks quite good. I am however curious about the source of the bimodality of the solar CRF in Figure 4. My first thought is that one of these peaks is associated with ice-covered areas and the other with open water, pointing to some residual bias in the method.

(10) Section 3.4: It would substantially increase the value of this section if you contextualized your simulated biases with your observations. For example, it would be interesting to see the biases from Figure 3c plotted over Figure 7 in the phase space of the figure panel that is most appropriate.

(11) P18L9: You have mentioned SHEBA a couple times, but have not referenced it (Uttal et al., 2002), nor have you defined the acronym.

References

Allan, R. P., & Ringer, M. A. (2003). Inconsistencies between satellite estimates of longwave cloud forcing and dynamical fields from reanalyses. Geophysical research

letters, 30(9).

Cox, C. J., Uttal, T., Long, C. N., Shupe, M. D., Stone, R. S., & Starkweather, S. (2016). The role of springtime Arctic clouds in determining autumn sea ice extent. Journal of Climate, 29(18), 6581-6596.

Cox, C. J., Walden, V. P., Rowe, P. M., & Shupe, M. D. (2015). Humidity trends imply increased sensitivity to clouds in a warming Arctic. Nature communications, 6, 10117.

Curry, J. A., Schramm, J. L., Rossow, W. B., & Randall, D. (1996). Overview of Arctic cloud and radiation characteristics. Journal of Climate, 9(8), 1731-1764.

Dong, X., Xi, B., Crosby, K., Long, C. N., Stone, R. S., & Shupe, M. D. (2010). A 10 year climatology of Arctic cloud fraction and radiative forcing at Barrow, Alaska. Journal of Geophysical Research: Atmospheres, 115(D17).

Dong, X., Xi, B., & Minnis, P. (2006). A climatology of midlatitude continental clouds from the ARM SGP central facility. Part II: Cloud fraction and surface radiative forcing. Journal of climate, 19(9), 1765-1783.

Ehrlich, A., & Wendisch, M. (2015). Reconstruction of high-resolution time series from slow-response broadband terrestrial irradiance measurements by deconvolution. Atmospheric Measurement Techniques, 8(9), 3671-3684.

Intrieri, J. M., Fairall, C. W., Shupe, M. D., Persson, P. O. G., Andreas, E. L., Guest, P. S., & Moritz, R. E. (2002). An annual cycle of Arctic surface cloud forcing at SHEBA. Journal of Geophysical Research: Oceans, 107(C10), SHE-13.

Long, C. N. (2005). On the estimation of clear-sky upwelling shortwave and longwave. Pacific Northwest National Lab., Richland, WA (US).

Long, C. N., & Ackerman, T. P. (2000). Identification of clear skies from broadband pyranometer measurements and calculation of downwelling shortwave cloud effects. Journal of Geophysical Research: Atmospheres, 105(D12), 15609-15626.

Long, C. N., & Turner, D. D. (2008). A method for continuous estimation of clear‐sky downwelling longwave radiative flux developed using ARM surface measurements. Journal of Geophysical Research: Atmospheres, 113(D18).

McFarlane, S. A., Long, C. N., & Flaherty, J. (2013). A climatology of surface cloud radiative effects at the ARM tropical western Pacific sites. Journal of Applied Meteorology and Climatology, 52(4), 996-1013.

Miller, N. B., Shupe, M. D., Cox, C. J., Walden, V. P., Turner, D. D., & Steffen, K. (2015). Cloud radiative forcing at Summit, Greenland. Journal of Climate, 28(15), 6267-6280.

Ohmura, A. (2001). Physical basis for the temperature-based melt-index method. Journal of applied Meteorology, 40(4), 753-761.

Podgorny, I., Lubin, D., & Perovich, D. K. (2018). Monte Carlo study of UAV-measurable albedo over Arctic sea ice. Journal of Atmospheric and Oceanic Technology, 35(1), 57-66.

Sedlar, J. (2018). Spring Arctic Atmospheric Preconditioning: Do Not Rule Out Shortwave Radiation Just Yet. Journal of Climate, 31(11), 4225-4240.

Sedlar, J., Tjernström, M., Mauritsen, T., Shupe, M. D., Brooks, I. M., Persson, P. O. G., ... & Nicolaus, M. (2011). A transitioning Arctic surface energy budget: the impacts of solar zenith angle, surface albedo and cloud radiative forcing. Climate dynamics, 37(7-8), 1643-1660.

Shupe, M. D., & Intrieri, J. M. (2004). Cloud radiative forcing of the Arctic surface: The influence of cloud properties, surface albedo, and solar zenith angle. Journal of Climate, 17(3), 616-628.

Stone, R. S. (1997). Variations in western Arctic temperatures in response to cloud radiative and synoptic‐scale influences. Journal of Geophysical Research: Atmospheres, 102(D18), 21769-21776.

Uttal, T., Curry, J. A., McPhee, M. G., Perovich, D. K., Moritz, R. E., Maslanik, J. A., ... & Heiberg, A. (2002). Surface heat budget of the Arctic Ocean. Bulletin of the American Meteorological Society, 83(2), 255-276.

Viúdez‐Mora, A., Costa‐Surós, M., Calbó, J., & González, J. A. (2015). Modeling atmospheric longwave radiation at the surface during overcast skies: The role of cloud base height. Journal of Geophysical Research: Atmospheres, 120(1), 199-214.
* * *

---

## Author Comment (AC1) · 19 Nov 2019

**Referee comments** are highlighted in **bold**, *changes* in the manuscript in *italic*.
* * *
**Main criticisms of this paper:**
**1) Additional referencing of the literature is needed to put the paper into context and understand what is novel in this work.**
**2) An uncertainty analysis is needed.**
**3) Some of the writing is confusing and needs to be edited for clarity.**
**4) There are gaps in the descriptions of the measurements and methods that make the work difficult to understand or reproduce.**
These points are discussed in the detailed comments below.

**Title: The title should be reworded. Perhaps something like, "Cloud radiative forcing at the Arctic springtime marginal sea ice zone derived using low-level airborne observations."**
The title proposed by the reviewer would implicate that we analyze/characterize the CRF during the ACLOUD campaign and analyze e.g. its dependence on surface and cloud properties. However this is not the primary intension of the manuscript and will be done in more details in an upcoming paper. Instead, the manuscript aims to discuss and refine the methods used to derive the CRF. In particular, we show the importance of the clear sky albedo estimate for the estimate of CRF, for which in fact we need observations to show the impact. However, we acknowledge that the focus of the work was not made clear. We changed the title to:

*"Reassessment of shortwave surface cloud radiative forcing in the Arctic: Consideration of surface albedo-cloud interactions"*

**Language (overall): The authors should go through the paper and make sure every paragraph makes a single, clear point. They should also go through each sentence and make sure that it is correct, comprehensible, and stated as simply as possible. I suggest asking a colleague to read the paper and then working with them on how to clarify anything that they do not understand.**
We apologize that our writing was confusing and restated/edited a couple of sections. Please see the marked-up manuscript for details.

**Abstract lines 1 and 2: "warming or cooling effect . . . on the radiative energy budget."Clouds do not cool the energy budget. Please restate.**
We have changed the first two sentences in the abstract accordingly to:

*"The concept of cloud radiative forcing (CRF) is commonly applied to quantify the impact of clouds on the surface radiative energy budget (REB). In the Arctic, radiative interactions between microphysical and macrophysical properties of clouds and the surface modify the warming or cooling effect of clouds, complicating the estimate of CRF obtained from observations or models."*

**First paragraph of the introduction: This paragraph is difficult to understand, has a lot of unnecessarily detail, and only provides general motivation. I think a few sentences can explain that clouds are important for the Arctic. What is really needed prior to the second paragraph is more specific motivation for this work, including why estimates of CRF and REB are important and how they are used and calculated in the literature (discussed more below).**

We agree, that the focus of the introduction did not fit perfectly the motivation of the study. Therefore, we changed the introduction to be more specific on the general estimate of CRF. In addition, we added a literature overview in the new section 3.1 "Common approaches" describing the state of the art methods to calculate the CRF.

Furthermore we restated the sentence about the cloud feedback (more specifically cloud radiative feedback) to (p.2 l.2):

*"One prominent example is the cloud radiative feedback, which includes the effects of an increasing cloud amount in the Arctic, balancing between potential increase of both longwave downward radiation (positive) and cloud top reflectivity (negative)."*

**Page 2 lines 18-22: these sentences are a distraction from the rest of the paragraph, which focuses on SZA and albedo.**

We completely changed this section. (See marked-up manuscript)

**When the words heterogeneous or heterogeneity are used, the authors need to state what is varying – for heterogeneous sea ice is it type of ice? Or ice fraction? What is meant by heterogeneous albedo?**

The reviewer is right, we should have been more specific. We intended to say "fluctuations of albedo in space scale", or "heterogeneous albedo fields". This heterogeneity is caused by sea ice concentration, patches of snow on bare sea ice etc. We reworded the sections, where it was unclear or misleading and used the two extended phrases "fluctuations of albedo", or "albedo fields".

**Abstract and Introduction: Is the most important result the application of the parametrization of Gardner and Sharp (2010) to measurements? If this is the case, this should be clear in the abstract and introduction. For example, the authors state in the introduction that, "Both processes have been parametrized, for example by Gardner and Sharp (2010) based on simulations, however, their impact on estimates of the CRF in the Arctic have not yet been evaluated." They should go on to state in the following paragraph that they apply these parametrizations in this work.**

The reviewer is right that we should have emphasized the importance of the Gardner and Sharp (2010) parameterization more clearly, which enables to reproduce the clear sky albedo from the cloudy observations. We added in the abstract:

*"A method to consider this surface albedo effect by continuously retrieving the cloud-free surface albedo from observations under cloudy conditions is proposed, using an available snow and ice albedo parameterization."*

As well we modified the last paragraph in the introduction to make the outcome/structure of the study more clear:

*"In section 4 a method is introduced to retrieve the shortwave surface albedo in the hypothetical cloud-free atmosphere from measurements under cloudy conditions, by using an available snow and ice albedo*

*parameterization from Gardner and Sharp (2010) and a shortwave transmissivity-based retrieval of cloud liquid water path (Appendix A)."*

**Introduction: Although the paper has a long reference list, missing are examples from the literature of calculations of CRF and radiative fluxes based on observations in the Arctic. Context of the literature is also needed to show what ideas or parametrizations are novel here. For example, if Eqs (7) and (8) are novel, please make that clear in the introduction as well as in Sect. 3, as well as how they relate to calculations of CRF in the literature. The authors should explain how CRF is used in such studies, what are the shortcomings, and how their work addresses these shortcomings.**

We extended the introduction by a section discussing the general conclusions from available studies (p.2 l.14).

*"Long-term ground-based observations of CRF in the Arctic (Walsh and Chapman, 1998; Shupe and Intrieri, 2004; Dong et al., 2010; Miller et al., 2015) showed that in the longwave wavelength range clouds tend to warm the surface. The magnitude of the warming is influenced by macrophysical and microphysical cloud properties (e.g., Shupe and Intrieri, 2004) and by regional characteristics (Miller et al., 2015) and climate change (Cox et al., 2015). In the solar spectral range, clouds rather cool, whereby the strength and timing over the year is determined, besides cloud microphysical properties, by the solar zenith angle (SZA) and the seasonal cycle of surface albedo (e.g., Intrieri et al., 2002; Dong et al., 2010; Miller et al., 2015). However, the required cloud-free reference ($F_{net;cf}$) poses a serious problem to all observations in the cloudy Arctic (Shupe et al., 2011), as the unknown thermodynamic and surface albedo conditions in cloud-free environments is modified by the presence of clouds itself."*

In addition, we added a more detailed literature overview in the new section 3.1 "Common approaches", where a discussion of the available approaches is given (please see the new manuscript). This section aims to identify shortcomings of the common approaches such as the representations of cloud-free albedo.

**Measurements: More information is needed about the measurements. What wavelengths are used? Presumably the aircraft had both up-looking and down-looking pyrgeometer and pyranometers?**
In section 2.2 we referred to two papers Wendisch et al. (2019) and Ehrlich et al. (2019b). These papers give a detailed overview of the campaign (Wendisch et al., 2019) and instrumentation (Ehrlich et al., 2019b), where all required information is provided. They extensively describe the campaign, the flights, the instrumentation, wing-by-wing comparison and necessary processing. However, to make it easier for the reader we added some general specifications of the broadband radiatiometer in the revised manuscript (p.4 l.3).

*"In this paper, shortwave and longwave, upward and downward broadband irradiance have been analyzed from measurements with a frequency of 20 Hz obtained from two sets of Pyranometer (0.2-3.6 µm) and Pyrgeometer (4.5-42 µm). From 5 these irradiance data the net irradiance and surface albedo have been derived."*

**(Also, please use "longwave" and "shortwave" instead of "terrestrial" and "solar."**

We have changed the wording, equations, subscripts and labels in the figures.

**The text refers to the cloudy ABL, but I think data are only used where the cloud was above the aircraft? (E.g. all upwelling flux measurements were for clear skies). Please clarify this.**

We apologize if the definition of CRF in the introduction with "cloudy" was misleading and we changed it to "all-sky". However, as stated in Ramanathan et al. (1989), "cloudy" is not equal to overcast conditions. The low level section are not filtered for overcast conditions. All data, from overcast, broken cloud fields, and clear-sky conditions are included, as can be seen in the CRF distributions shown in Fig. 10b,c (clear-sky values around 0).

As described in section 3.2 the upward longwave fluxes cancel in the CRF equation assuming the instantaneous approach. The upward shortwave fluxes or the surface albedo are obtain in all-sky conditions and are corrected by the method described in section 4 to represent the cloud-free albedo.

**How did you ensure that there was no cloud around or below the aircraft?**

The average flight altitude for the low-level section was 80m, and often even below 60m. Therefore, the low-level section were always below the lowest cloud base. There have been cases with precipitation, but precipitation does influence the radiative fluxes only little. To confirm the cloud-free conditions below the aircraft , we used cameras installed in the aircraft, participated in the flights or checked the Nevzorov Probe (LWC,TWC) data (if available) for cloud particles in flight altitude. (Dataset available: https://doi.org/10.1594/PANGAEA.906658 )

**Please also provide more detail about measurements of atmospheric thermodynamics, and a table showing the various measurements (with time and location) as context.**

During the ACLOUD campaign we obtained hundreds of in situ profiles covering the 6 weeks with 2 aircraft and dropsondes in a region north-west of Svalbard (see Fig. 1), which were merged with radiosoundings from Ny-Alesund and from the (partly moving) research vessel Polarstern during the PASCAL campaign, described in section 2.2 and 2.3. Therefore, it is hard to give a comprise overview of these data in a single table. In general, the airborne in situ profiles are distributed similar to the flight pattern shown in Fig. 1 (descending/ascending before/after each low-level section from/to lower/higher altitude). We added (p.4 l. 10):

*"The local atmospheric thermodynamic state, including air temperature and relative humidity was determined by dropsondes (Ehrlich et al.,2019) and aircraft in situ observations (Hartmann et al.,_2019) during ascents and descents in the vicinity of the low-level flight sections."*

All details (e.g. cruise track of Polarstern) can be accessed from the cited papers Wendisch et al. (2019) and Ehrlich et al. (2019). The datasets are all available on the PANGAEA database (see data availability). Time series of vertical profiles from radiosoundings during ACLOUD/PASCAL are shown in Knudsen et al. (2018).

**Finally, Page 2, line 11 implies that this work uses all-sky minus clear-sky (other definitions of CRF use cloudy-sky minus clear sky). Were all downwelling flux measurements of cloudy sky?**

See reply three comments above. We used all scenes during the flight sections including overcast, broken cloud fields and clear sky.

**Radiative transfer simulations: Sufficient information is needed here that the results are reproducible. What was the vertical resolution? How were the measurements (dropsonde, radiosonde, and surface) merged? What was used for concentrations of other trace gases (most notably for the longwave calculations, CO2). It would be helpful to specify which flux and albedo terms were calculated with the various models (longwave, shortwave and 2D vs. 3D). What are the uncertainties for the radiative**

**transfer calculations?**

We apologize that we did not fully described these technical details, which are in fact important for the reproducibility. We extended the section 2.3 to fulfill all points:

*"The radiative transfer simulations for the cloud-free conditions were performed with the libRadtran package (Emde et al., 2016) using the one-dimensional, plane-parallel discrete ordinate radiative transfer solver DISORT (Stamnes et al., 1988) and the molecular absorption parameterization from Kato et al. (1999) for the shortwave spectral range (0.28–4 μm), and from Gasteiger et al. (2014) for the longwave wavelengths range (4–100 μm)."*

*"Hence, profiles from in situ measurements of temperature and relative humidity on board of both aircraft and, if available, dropsonde measurements from the Polar 5 aircraft were used to replace the radiosounding layers by the local atmospheric profiles."*

*"The atmospheric levels below flight altitude were linearly interpolated to the surface temperature observed by the KT-19 assuming an emissivity of unity. The assumption of the black-body emissivity is justified by the high spectral emissivity for nadir observations in this wavelength range (Hori et al., 2006)."*

*"The sub-Arctic summer profile (Anderson et al., 1986) was used to complete the profiles including gas concentrations up to 120 km altitude."*

*"The high vertical resolution of the in situ observations was reduced for the radiative transfer simulations to 30m below 1000m and stepwise increases to 5km at 120km altitude. The surface albedo is obtained from upward and downward looking pyranometers and a method described in section 4."*

In the last paragraph of section 2.3 we clarify which radiative transfer model is used for which purpose (p.5 l18):

*"Spectral surface albedo values for the sensitivity study in section 3.5 were simulated using the spectral Two-streAm Radiative TransfEr in Snow model (TARTES) (Libois et al., 2013). 3D radiative transfer simulations for the albedo smoothing kernels applied in section 3.4 and the appendix A were performed with the open-source Monte Carlo Atmospheric Radiative Transfer Simulator (MCARaTS) (Iwabuchi, 2006; Iwabuchi and Kobayashi, 2008)."*

Regarding the uncertainties of radiative transfer simulations we referred to Randels et al. (2013). We added in section 3.4.1 (p.12 l.14-16):
*"In addition to measurement uncertainties of the used broadband radiometer (<3 %, (Ehrlich et al., 2019b)), the radiative transfer modelling can induce a bias (<2 %) in the shortwave wavelength ranges (Randles et al., 2013)."*

**Section 3.2: A lot of work has been done on the longwave CRF that is relevant here. For example, Cox et al 2015 (listed above) examines temperature and humidity.**

We are sorry, if the focus of this section was not made clear. This study should show, how important it is to track thermodynamic profile changes during large scale processes like warm air intrusions or cold air outbreaks, because they significantly influence the local estimate of CRF and how these processes impact

the local energy budget. We did not intend to generally explain how temperature and humidity changes downward irradiance.

We changed some sentences in this section and added specific citations to make the focus of the section more clear:

(p. 8 l. 30-31)

*"As was shown by Tjernström et al. (2015, 2019) such events might significantly impact the local energy budget along the trajectory."*

(p.10 l. 4-6)

*"Especially for airmass transformation like warm air intrusions and cold air outbreaks in the Arctic (Pithan et al., 2018), this is a relevant issue, and requires a precise representation of airmass transformations by models or local in situ observations along the trajectory."*

**Uncertainty analysis: A variety of assumptions are made in this work and the calculations and measurements all have associated errors and uncertainties. An uncertainty analysis is needed.**

In the following we try to give a realistic estimate of uncertainties based on the presented workflow, which consisted of uncertainties in the observed broadband radiation and surface albedo, the simulated radiation, the LWP equivalent retrieval and the retrieved cloud-free albedo and cloud-free net fluxes. A detailed discussion of this uncertainty analysis would blow up the manuscript significantly and distract from the main conclusions. We, therefore, did not put all the calculations in the revised manuscript but added the uncertainty estimate where it was needed.

[Figure]

Underlined: Uncertainties in the LWP equivalent retrieval:

To keep the uncertainties in a realistic range, we applied an deviation between the observed and simulated downward solar irradiance of 1.2 %, based on the results from the cloud-free flux closure study in section 3.4 (Figure 4 and deviations in the text). The albedo uncertainties result in 2.4 %. In Figure 1 (this document) the estimated absolute and relative uncertainties are given over ice and open ocean as a function of LWP and for a solar zenith angle of 60°

*Figure 1 Absolute and relative retrieval uncertainties of equivalent LWP as a function of cloud LWP and for two surface albedos above open ocean (0.07) and sea ice/snow (0.8).*

(representative for ACLOUD). The theoretical observations and lookup tables for the "observed" conditions are combined in worst case scenario.

We added in the appendix (p. 23 l. 16-17):

*"The relative uncertainty range of this retrieval for homogeneous clouds and surface can be expected between 15 % and 35 % over open ocean and sea ice respectively."*

Underlined: Uncertainties in the retrieval of cloud-free albedo:

In Fig. 2 (this document) synthetic distributions of LWP and Albedo are applied to the lookup tables generated using the parameterization from Gardner and Sharp (2010). The major contribution to the

uncertainties of the retrieved cloud-free albedo stems from the observed broadband albedo itself. Uncertainties from the LWP retrieval contribute only minor to errors in the cloud-free albedo. Given the uncertainty of the observed albedo of 2.4 % and the error in simulated shortwave F_down of (2 %), we conclude that the relative uncertainty of solar net fluxes (F_down_cf – alpha_cf * F_down_cf) is below 20 % above ice and lower above the open ocean and added in section 4.1.1 (p. 18 l. 32 – p.19 l.2):

*"The uncertainties in the estimate alpha_cf and the shortwave F_net,cf depend mainly on the observed alpha_all, as was investigated by applying synthetic albedo and LWP distributions to the lookup tables. Due to the non-linear increase of alpha_all with LWP the potential error induced by uncertainties in the retrieved LWP is larger for lower LWP and additionally depends on the prevailing surface types. The overall uncertainty in the cloud-free shortwave net fluxes using the retrieved alpha_cf can be expected below 20% above high surface albedos, decreasing with decreasing surface albedo."*

[Figure]

*Figure 2  Upper Panels: Input for the cloud-free albedo retrieval. Lower Panels: Output of cloud-free albedo (red distribution). Albedo input in black as reference. Left panels: Only LWP uncertainty applied. Middle panels: Only albedo uncertainties applied. Right panel: LWP and Albedo uncertainties combined.*

**Technical details:**
**- I_f needs to be defined before the first time it is used.**
We corrected it for the first appearance in section 2.1.

Additional changes:
We changed a typo on page 16 l.27-18 (manuscript in discussion):
The average SZA during the low level flights over sea ice under cloudy conditions was 61° not 59°.

*"These values hold for the ACLOUD observations with an average LWP during cloudy conditions over sea ice of 58 gm-2 and a SZA of 61°."*

We corrected a wrong statement on p.16 l. 22-24 (manuscript in discussion):
*"The non-linearity in the functional dependence of surface albedo and LWP spreads the frequency distribution, while the mode for cloud-free conditions is not affected. "*

In section 4.1 we reduced the paragraph covering the ice fraction dependent scaling of the Gardner and Sharp (2010) parameterization, because only homogeneous sea ice distributions are discussed/showed with respect to the surface albedo-cloud interaction and the required introduction of the equation, which would distract from the section (p. 19 l.5-7):
*"However, making use of the cosine weighted sea ice fraction I_f and its linear relation to the albedo, changes due to the surface albedo-cloud interaction can be scaled to the prevailing I_f by assuming diffuse radiative transfer (Lambertian albedo) (not shown in this study)."*

---

## Author Comment (AC2) · 19 Nov 2019

**Referee comments** are highlighted in **bold**, *changes* in the manuscript in *italic*.
* * *
**General comments In summary, I think this is interesting work, but it needs a more solid foundation and a discisson about the why's and the how's, and less details on ACLOUD; maybe do a separate but more exten-sive paper on CREs during ACLOUD, referencing this paper.**
See third comment.

**Maybe I'm nitpicking, but the terminology has been changed by the climate community, from Cloud Radiative Forcing, or CRF, to Cloud Radiative Effect, or CRE, quite a while ago.**

As was noted by reviewer 3, there are inconsistences in the definition of CRF and CRE. These parameters represent different definitions, as described in Cox et al. (2015). We derive the instantaneous CRF and, therefore, kept CRF in the revised manuscript. To make the definition of CRF more precise, we added this sentence in section 3.2 (p.7l.10):

*"As was stated by Cox et al. (2015) the CRF definition refers to net irradiances, while the cloud radiative effect (CRE) characterizes only changes in the downward irradiance."*

**The title is also in my personal opinion too long and clunky; try something shorter. Maybe "Interactions be-tween surface albedo and clouds for Arctic cloud radiative effect estimation".**

We changed the title to:

*"Reassessment of shortwave surface cloud radiative forcing in the Arctic: Consideration of surface albedo-cloud interactions"*

**The content in the study balances along many borders and as a consequence it doesn't quite fulfill any of the topics it crosses well enough. It is unclear if this is a theoretical study that uses ACLOUD data just because its good and convenient or if it is a contribution to ACLOUD as such. I already hear the authors say "can't it be both?" and my response is it would be a better radiation interaction paper if ACLOUD was tuned down and a better ACLOUD paper if the radiation stuff was more background and the actual results where more detailed. Typically to be a good paper on both aspects it would have to be longer – which is not good. There is also a lot taken for granted on the readers; not everyone is a radiative transfer modeling expert. So choices must be made.**

We tried to reduce the ACLOUD topic as much as possible, but the study needs to be based on observations to show the impact of the investigated physical processes in real conditions. A study completely based on theoretically constructed scenarios would raise the question how relevant these results are in reality. The

results shown are not meant to specifically characterize the clouds observed during ACLOUD. ACLOUD data is more a test base to apply the new approach and contributes in the end with (see abstract):

*"Applying ACLOUD data it is shown that the estimated average shortwave cooling effect by clouds almost doubles over snow and ice covered surfaces (-62Wm-2 instead of -32Wm-2), if surface albedo-cloud interactions are considered."*

However, these observations require an introduction, especially because it is a new dataset with challenges discussed in this paper. We totally agree that unfortunately these sections partly distract from the main aspect of the paper. On the other hand, observations like the one from ACLOUD enables to quantify the dependence of surface albedo with cloud LWP (Fig. 9) from observations, a process that represents the major motivation for this study. Regarding **"can't it be both?"**, we have the opinion it must be both. By using synthetic radiative transfer simulation the manuscript shows, why surface albedo-cloud interactions matter for the estimate of CRF. The solution of this problem depends on the application on real data when the cloud-free albedo needs to be estimated from cloudy sky observations, a problem reported by different studies in literature. Only with help of the ACLOUD data, we can show that the proposed approach does lead to an improvement of the CRF estimates.

Furthermore, we keep the ACLOUD specific interpretation of the cloud properties as short as possible. We do not interpret the obtained CRF in detail, simply the relevant sign of the distributions and magnitude of changes related to surface albedo-cloud interactions (main aspect of this manuscript) are highlighted. The interpretation of CRF during ACLOUD will be treated in an upcoming paper.

**Besides the use of an old terminology (CRF instead of CRE) there needs to be a much more in-depth and philosophical background.**

See reply to comment 2.

**Why are we interested in CRE (or CRF) and how does that impact how we do these calculations? The question "How does the clouds affect the surface energy budget" is not the same as the question "How would the surface energy budget look if the clouds were not present?". As an example, the authors argue both that changes in surface broadband albedo between cloudy and clear states must be considered, and that the details in thermodynamic are important. However, we know that the thermodynamic profiles for clear and cloudy cases are very dif-ferent. The presence of clouds depends on the vertical profiles, but the clouds themselves also modi-fy the profiles by their presence. Yet the authors argue that we change the surface albedo to how would be without the clouds, but keep the thermodynamic profiles as they are; only remove the cloud water. I think that the an- swers depend on what we want to use this metric for. Ideally the an-swer to my second question above would have us examine the conditions in clear and cloudy condi-tions separately, not modifying cloudy cases by removing the condensed water. However, the Arctic is a very cloudy place and there are not enough clear cases to make this possible. Therefore, I think this paper needs a much more detailed introduction and background to what we are trying to do and why.**

We agree, that the motivation and definition of the CRF is not well given in the manuscript. Therefore, in the revised manuscript, a new section is added. In the new section 3.1 the available approaches applied in literature to derive CRF are discussed. We adjusted the introduction to explain what is changing between

the cloudy and cloud-free atmospheric state in the Arctic and how it affects the estimate of CRF by the approaches listed in Section 3.1.

For the longwave CRF we are totally aware of changing thermodynamics between the cloudy and clear state. However, we aim to quantify an instantaneous effect, switching cloud on/off without changing the atmospheric profile. This is justified by putting the main focus on radiative effects caused by the shortwave CRF (see also changes in the title). To highlight which approach we use, we added "instantaneous" CRF at couple of sections, what should make clear, that we neglect the impact of changes of temperature and humidity.

**1) Effects of proper thermody-namic profiles:**
**1) This is pretty obvious; of course one must use the profile from the same location as the CRE is considered. In this paper this is discussed in the context of aircraft observations covering an area, but the conclusion is also important for fixed-point observations. One question is, if a proper sounding at the location is not available, from how far away can it be used? This question of course has no an-swer other than "it depends." But here one could also raise the issue of cloudy profiles being differ-ent from the corresponding clear case; in other words, if we could magically remove the cloud water so that the clouds vanish, what would that do to the thermodynamic profiles? Or, are the profiles found in clear conditions systematically different from those in cloudy conditions? From both mod-eling and observations in subtropical stratocumulus regions we know that the moist PBL is deeper and warmer when clouds are present; for the Arctic we don't really know, but I would wager a bet that they are different!**
We did not intend to discuss in general how the atmospheric thermodynamic profiles influence the estimate of CRF and that a local profiles are required, which is of course obvious. In this section the focus is mainly on the impact of air mass transformation, such as warm air intrusions or cold air outbreaks, which changes the thermodynamic state within relatively small horizontal scales. By citing the literature from Tjernström et al. (2015,2019) and Pithan et al. (2018) in this section our message should get more clear now. Regarding the thermodynamic states, see also new section 3.1.
**"…from how far away can it be used?"**
Yes, it depends on the scenario and where exactly a different air mass is located. We, therefore, do not try to give a general answer to this question and try to estimate the effect for our observations where we use the in situ profiles directly before and after each low level leg to replace the layers of the remote radiosoundings from Polarstern or Ny-Alesund.

**"…, if we could magically remove the cloud water so that the clouds vanish, what would that do to the thermodynamic profiles?"**
See comment above. We aim to quantify an instantaneous effect neglecting changes of the thermodynamic state and their impact on the terrestrial CRF. To clarify this we added the new section 3.1.

**2) Effects of heterogeneous surfaces**
**2) That heterogeneous surfaces poses a problem for upwelling shortwave radiation is also pretty obvious. This is a main factor in the MIZ but also in the pack ice mainly due to melt ponds.**
In this section we focus on the impact of surface heterogeneity on shortwave downward radiation by horizontal photon transport, not the obvious upwelling shortwave radiation, which is roughly a linear function of cosine weighted sea ice concentration.
**This pa-per doesn't even mention the effect of melt ponds, presumably because there weren't any during ACLOUD.**
We mention melt ponds and discuss their effects in several sections in the paper, although we had only a low percentage during the ACLOUD campaign as was shown by Jäkel et al. (2019).  Of course we are aware

of their radiative effects in the summer melting season, which are part of the hypothesis Fig. 8 and discussed in section 3.5/3.51 as well as the conclusion.

**So is this an ACLOUD paper usuing advanced radiation methods or a radiation paper using ACLOUD observations?**
See our reply to the general comment above.

**Moreover, to this reviewer it is not obvious that the downwelling radiation is dependent on sur-face albedo; in most NWP or climate models that I know this is not considered, but may be ignorant.**
The downward radiation is affected by the surface albedo due to multiple scattering effects. With increasing surface albedo, the upward irradiance will increase. This increased upward irradiance is then scattered back by aerosol particles and atmospheric gases in cloud-free conditions (or clouds during cloudy conditions) back to the surface. This multiple scattering contributes to an increase of downward irradiance over highly reflective surface types like snow, compared to absorbing surfaces like water (see also reply next comment). This effect of course is considered in NWP models as it is simulated by any radiative transfer scheme.
What is not considered in NWP models is:
- Horizontal photon transport due to multiple scattering between neighboring grid cells (3D radiative transfer required, or the areal average albedo) (see section 3.4)
- A change of the surface albedo due to different illumination conditions (cloudy vs. cloud-free) and cloud optical thickness (see section 3.5/3.51). This effect is a major subject of the manuscript.

**Either way, for a reader like me, this needs to discussed in more detail. Commenting about "horizon-tal photon transport" is not sufficient.**
In the revised manuscript, a short explanation of the horizontal photon transport in case of inhomogeneous surface conditions is given (p.8 l.2).

*"For highly reflective surface types like snow the upward irradiance is significantly higher compared to over mostly absorbing surfaces like ocean water. A part of this upward irradiance is scattered back towards the surface (often referred to as multiple scattering), and thus, contributes to the downward irradiance. Consequently the multiple scattering between surface and atmosphere causes an increase of downward irradiance over snow and ice compared to open ocean. Photons reflected from a bright surfaces like an ice flow might scatter back to the surface increasing the downward radiation over dark areas like surrounding ocean water. For airborne observations in the MIZ, characterized by strong variability in surface albedo due to the variable sea ice cover, as well as ground based measurements in heterogeneous terrain, this, often referred to as, horizontal photon transport due to multiple scattering from the surrounding area to the actual point of observation is not negligible for the estimate of F_down_sw (Ricchiazzi and Gautier, 1998; Kreuter et al., 2014)."*

In addition we cite five papers distributed in the manuscript (Ricchiazzi and Gautier, 1998; Kreuter et al., 2014; Weihs et al., 2001; Wendisch et al., 2004; Pirazzini and Raisanen, 2008), which specifically discuss this topic.

**If this is a factor, how large is it?**
The magnitude of the horizontal photon transport is quantified for the case study in Fig. 3. To make this more obvious, we added/reworded this section by (p.10 l. 22):
*"However, due to horizontal photon transport from surrounding ice fields in reality the changes in F_sw,cf are less pronounced. The quantitative impact of multiple scattering on F_sw,cf is indicated by the gray shaded area in Fig. 3b with a maximum contribution of almost 40Wm-2 (relative to open ocean)."*

**What is it caused by? Are there differences between say the MIZ, with alternating ice and open water, pack ice with melt ponds, pack ice with open leads, or pack ice with many substantial pressure ridges? Or all of the above?**

The 3D radiative effects due to horizontal photon transport in the MIZ are complex and depend on the given scenario (surface albedo map) as indicated by the reviewer.

The presented approach (smoothing the surface albedo using an appropriate filter shape, simulated using simplified scenes (see replies to the other reviewers)) can only give a rough estimate of the conditions during ACLOUD, but on the other hand, it is still simple enough to be applied to observations. However, to address this issue, we added a short discussion in section 3.4 (p.12 l.2):

*"This enables a more reliable estimate of the CRF in the heterogeneous MIZ and over the specific surface types, taking into account that the complexity of surface albedo fields in the MIZ can only be insufficiently represented by this simplified approach to estimate the areal averaged albedo."*

**3) Effects of the clouds on the characteris-tics of the solar radiation.**
**3) Changes in the spectral composition from absorption in clouds is a real tangible effect that one can discuss if it is necessary to compensate for or not; see my discussion above.**

The change of the spectral composition is not only due to absorption. The largest difference is the almost wavelength independent scattering by clouds (Mie regime) compared to the preferred scattering of short (blue) wavelength of atmospheric gases (Rayleigh regime). Clouds are white, the cloud-free sky is blue. In section 3.5/3.51 of the manuscript, we demonstrate why these spectral differences are of relevance and how a compensation affect the estimated CRF.

**Changes due to the different distribution between direct and diffuse radiation is trickier. Also the cloud albedo is sensi-tive to solar zenith angle.**

All these effects are considered and analyzed in the radiative transfer based study in section 3.5/3.51.

**Finally, the language is mostly OK, but occasionally I stumble on unnecessarily difficult wording, for example "exemplarily" in the context it is used is an existing word but even the dictionary indi-cates it isn't much used in modern English.**

Thanks for this remark, we reworded the sections.

**There are also past/present inconsistencies; what is done and presented in this paper is sometimes described in past tense and sometimes in present. Either is fine with me; just be consistent.**

We apologize for these inconsistences and corrected it.

**Finally, final: among the data made available here, only a subset is actually really made available; the rest is just referenced.**

During the submission of the manuscript, the publication process of the data in PANGAEA was not finalized and parts of the data might have been inaccessible. All basic data from the broadband radiometers, dropsondes, radiosondes, aircraft temperature and humidity measurements, and the camera images are available (Please see data availability). The publication of derived quantities such as LWP, cloud-free albedo and CRF is currently in progress. We wanted to wait for the reviews before publishing the data. The references will be added in the manuscript before publication. The just referenced data are dataset with a doi made fully available on the PANGEA database.

**Title is unnecessarily clunky**

We changed the title, see reply third comment.

**also here and throughout the paper, Cloud Radiative Forcing (CRF) should be replaced by Cloud Radiative Effect (CRE); even maybe surface CRE.**
See first reply in this document.

**Page 1, line 23: Comma after amplification.**
Corrected.

**Page 2, line 1-2: If this is the prime question, one can not use the cloud profile minus the cloud as representative for clear conditions; one must do the clear and the cloudy cases completely separately.**

Thanks for this comment, which indicates, that our text might lead to a mix-up of the cloud feedback in the Arctic and the cloud radiative forcing. The former is obtained from climate model studies, the latter is a measure of how cloud influence the energy budget at a certain location and time. The manuscript only deals with the cloud radiative forcing. We changed the sentence to more clearly separate feedback and CRF (p.2 l.2).

*"One prominent example is the cloud radiative feedback, which includes the effects of an increasing cloud amount in the Arctic, balancing between potential increase of both longwave downward radiation (positive) and cloud top reflectivity (negative)."*

**Page 2, line 3: It is not at all clear that clouds are cooling the surface in the Arctic in summer, so I would drop "dominates". It depends on a lot of factors, some of which this paper deal with. A clear case when this statement is correct, perhaps the only one observed case, is for SHEBA; the only annual observations that exist and BTW where the CRE was calculated without any of the corrections discussed here. Suffice it to say that I've seen summer conditions with a lot of snow and almost no melt ponds at very high latitudes where surface temperatures plummets when the clouds dissipate.**
The reviewer is totally right. See the changes in the previous comment. We furthermore added a more specific discussion of the CRE of Arctic clouds (in the introduction) (p.2 l.14):

*"Long-term ground-based observations of CRF in the Arctic (Walsh and Chapman, 1998; Shupe and Intrieri, 2004; Dong et al., 2010; Miller et al., 2015) showed that in the longwave wavelength range clouds tend to warm the surface. The magnitude of the warming is influenced by macrophysical and microphysical cloud properties (e.g., Shupe and Intrieri, 2004) and by regional characteristics (Miller et al., 2015) and climate change (Cox et al., 2015). In the solar spectral range, clouds rather cool, whereby the strength and timing over the year is determined, besides cloud microphysical properties, by the solar zenith angle (SZA) and the seasonal cycle of surface albedo (e.g., Intrieri et al., 2002; Dong et al., 2010; Miller et al., 2015)."*

**Page 2, line 11: The cases are "clear" or "all-sky", so I would swap places between "all-sky" and clear here. You can still define that as "cloudy" from her on, but it should be stated that the normal case is the existing clouds; not just when it is completely overcast.**
Ramanathan et al. (1989) defines cloudy as not necessarily overcast, but to clarify the definition throughout the whole manuscript we changed it to "all-sky".

**Page 2, line 20: Here it is stipulated that surface albedo affects the incoming (downwelling) solar radiation for clear skies. To me that is not obvious, and even if it is obvious from multiple refelctions in**

**clouds it is not obvious that it is important for clear-sky radiation. Do spend some more time on this please.**

We discuss here multiple scattering in cloud-free conditions, not multiple scattering related to clouds. This part was moved to section 3.2, where we state now (p. 8 l.1):

*"The downward shortwave irradiance at the surface in cloud-free conditions (F_dw_sw_cf ) is modulated by the atmospheric profile parameters, but also by the surface albedo. For highly reflective surface types like snow the upward irradiance is significantly higher compared to over mostly absorbing surfaces like ocean water. A part of this upward irradiance is scattered back towards the surface (often referred to as multiple scattering), and thus, contributes to the downward irradiance. Consequently the multiple scattering between surface and atmosphere causes an increase of downward irradiance over snow and ice compared to open ocean. Photons reflected from a bright surfaces like an ice flow might scatter back to the surface increasing the downward radiation over dark areas like surrounding ocean water. For airborne observations in the MIZ, characterized by strong variability in surface albedo due to the variable sea ice cover, as well as ground based measurements in heterogeneous terrain, this, often referred to as, horizontal photon transport due to multiple scattering from the surrounding area to the actual point of observation is not negligible for the estimate of F_dw_sw  (Ricchiazzi and Gautier, 1998; Kreuter et al., 2014)."*

**Page 2, line 14-15: ". . . observations of . . . conditions and of atmospheric thermodynamic state."**
This sentence was removed from the introduction.

**Page 2, line 26: Is shape the right word here? Isn't it the magnitudes; not just the shape?**
Yes the reviewer is right, we intended to say spectral albedo type not shape. This sentence was remove from the introduction and the topics is covered now in section 3.5.1.

**Page 2, line 28: Comma after "albedo".**
This sentence was remove from the introduction.

**Page 3, line 17-18: Here's an example of tense mismatch: ". . . were investigated in this paper" and ". . . aircraft is displayed . . .". Later on same page the dataset "is" merged and on line 3-4 on the next page ". . . concentration was calculated . . .".**
We checked the whole manuscript for tense mismatch. Thanks for this eye-opener.

**Paragraph staring on Page 6, line 29:How do you handle the observed surface albedo when calcu-lating CRE (or surface ïA¸Dˇ F)? The text says that the clear-sky albedo is set to 80% and the zenith an-gle to 80_; presumably those were not the observed conditions?**
Only for this sensitivity study in section 3.3 the albedo and SZA was fixed. For all simulations used to derive the CRF based on observations (section 4), the cloud-free albedo estimated from the measured are applied. To clarify that it is only applied for this sensitivity study, and why we did this, we added (p.9 l.7):
*"The surface albedo and SZA is fixed for this sensitivity study to 0.8 and 60° respectively, similar to the observed conditions over sea ice during that flight, in order to avoid any effects induced by changing SZA or surface albedo."*

**Section 3.3: Spend some time explaining why albedo affects downward solar radiation.**
See reply on comment on Page 2, line 20.
**Also, ex-plain the choice of albedo filter; is there any theoretical consideration here or was it "trial and error"?**

There is no analytic theoretical basis for the filter. However, we based our choice on 3D radiative transfer simulations of the downward irradiance in case of a typical lead, which are presented here in Fig. 1 (this document). The simulations indicate, that the shape of a La-Place distribution (with the shape parameter as defined now on in the section) is required/suitable to obtain the observed weighting of albedo information in the near-field of the aircraft. Yes it is "trial and error".

[Figure]

*Figure 1 Broadband shortwave 3D radiative transfer simulations in clear-sky conditions of a 1 km lead embedded in homogeneous sea as shown in the lower bound of the upper panel. Upper panel: Vertical distribution of albedo. Comparison between surface albedo, albedo as observed by an aircraft in 80 m flight altitude and the smoothed albedo using the filter embedded in the lower panel. Lower panel: Comparison between 3D downward irradiance perpendicular to the lead as it would be observed in 80 m altitude (solid), 1D simulations using the observed albedo as input and the final product 1D simulations using the smoothed albedo. SZA is set to 60°.*

Although the match (red dashed to solid black) is not perfect, by comparing it with the approach using simply the observed albedo (dashed black) it becomes clear that a certain smoothing is required. We tried different filter shapes and found that the La Place filter gives the appropriate weighting of the near and far-field albedo for this specific scene also for different lead sizes. It should be stated as well, that this smoothing depends on the atmospheric profile (here simply subarctic summer standard profile), albedo distribution etc. and can only give a rough estimate of the observed conditions during ACLOUD.

**Paragraph starting at Page 9, line 14: It seems to me this is a test comparing calculated results to observed results from a clear day; correct? If so, maybe this should not be reported under this heading? And maybe the term CRF (or rather CRE) should not be used when there are no clouds and the CRE is expected to be zero?**
Yes, this is correct, here we look at cloud-free conditions. Ideally, the CRF should be zero. However, we still think, that we need to use CRF in this section as well. The idea of the histogram is to give an uncertainty estimate for the CRF and not for, e.g., the downward irradiance. This approach is similar to Shupe and Intrieri (2004). The calculations use the same method as described in 3.3 to estimate the CRE. Differences

from zero indicate the uncertainties related to non-cloud effects.  In the revised manuscript, we separate this comparison by using a subsection 3.4.1:
*"Uncertainty estimate in cloud-free conditions"*
In addition, we give the average and standard deviation of downward irradiance between simulations and observations in the text.

**Page 10, line 6: Explain SSA.**
SAA is a measure of the snow grain size. To make this link more clear, we added *"a measure of snow grain size"* to the introduction of SSA and added a citation of Gardner and Sharp (2010), where it is nicely introduced (p.13 l.4).
*"Different snow packs with a density of 300 kgm-2 are specified with various values of snow geometric thickness and specific surface area (SSA, a measure of snow grain size) (Gardner and Sharp, 2010), and located above a layer representing bare sea ice with a wavelength constant broadband albedo of 0.5."*

**Page 10, line 9: I don't think you could find a case with 1 cm snow thickness in reality. That would be > 1cm at some location and no snow at other.**
The purpose of these three snow packs is to represent a typical spectral albedo not necessarily "one of the real" snow packs in the Arctic. It is more relevant to show the spectral features related with the onset of melting. Of course, on small spatial scales, the snow depth can vary and be zero. However, the albedo used in the simulations should represent the spatial average of a representative area where snow covered and bare ice are mixed. As the TARTES model is not made for slush or melt-ponds, the snow thickness is a scaling factor for the shorter wavelengths to roughly represent snow or white ice in the summertime Arctic.
An example of 2-4cm of snow above sea ice can be found in Fig. 4 in Zatko and Warren (2015), where nicely the impact on the shorter wavelength is shown in reality.

Zatko, M., & Warren, S. (2015). East Antarctic sea ice in spring: Spectral albedo of snow, nilas, frost flowers and slush, and light-absorbing impurities in snow. Annals of Glaciology, 56(69), 53-64. doi:10.3189/2015AoG69A574

**Paragraph starting at Page 10, last line: This should come before the calculation specific. First ex-plain why and then how.**
In the revised manuscript, these effects are already introduced in the introduction (p.2 l.31):

*"Besides temperature and humidity changes, clouds modify the illumination and reflection of the surface. For highly reflecting snow surfaces, radiative transfer simulations show that two processes are crucial: (i) A cloud-induced weighting of the transmitted downward irradiance to smaller wavelengths, causing an increase of shortwave surface albedo, and (ii) a shift from mainly direct to rather diffuse irradiance in cloudy conditions, which decreases the shortwave albedo (Warren, 1982). Observations have shown that, in general, there is a tendency that the surface albedo is larger in cloudy, compared to cloud-free conditions (e.g., Grenfell and Perovich, 2008), and was demonstrated for a seasonal cycle by Walsh and Chapman (1998) for highly reflecting surface types."*

**Page 11, line 3-4: Why is diffuse radiation coming in at a zenith angle of _50_? When the cloud is thick enough that where the sun is in the sky can no longer be determined, is there a zenith angle at all? I thought, but may be wrong, that diffuse mean precisely that the radiation was equally strong in all directions.**

It is correct, that the radiation field below clouds is diffuse and the Sun is not visible. The 50° represent an "effective" solar zenith angle for which the surface albedo of pure direct illumination is similar to diffuse illumination. The angular weighted incoming diffuse irradiance has an "effective" (average) incoming angle of 50°, for details we refer to Warren (1982) or Gardner and Sharp (2010).

In the manuscript we write (p.13 l.19):
"… clouds decrease the averaged incoming (effective) angle of the mainly diffuse irradiance to approximately 50° above snow (Warren, 1982)."

**Page 11, line 18: Drop "the".**
Corrected.

**Page 12, line 17: ". . . with increasing LWP is not, or only poorly, parameterized. . ."**
Corrected.

**Page 12, line 18: Unclear past tense in "have been used". Previously, or did you do this work now. In the previous case, give reference; in the latter, present tense should be used.**
Corrected.

**Figure 8: The SHEBA albedo line includes melt ponds and eventually even a lead. The drop in albedo starting at the beginning of June is due to this; as there were no melt ponds I ACLOUD(?), your comparing apples and pears here. ; as there were no melt ponds I ACLOUD(?)**

As ACLOUD is limited in time, the aim of this section is to transfer the results estimated for the albedo-cloud interaction to the seasonal cycle in the Arctic. The SHEBA data, even when influenced by melt ponds, is the only comparable data set available. Therefore, the comparison does not aim to have a perfect match between ACLOUD and SHEBA. As was shown by Intrieri et al. 2002 with the onset of melt pond formation and the related strong drop in the albedo, the total CRF shifts to a cooling effect. ⇔ For ACLOUD we had higher albedo values, but we still find the transition to a total cooling in the end of the campaign caused by the surface albedo-cloud interaction (which is not represented in the study from Intrieri et al. (2002)). That is why we state in the last sentence of section 4.2 that the transition to the cooling might start earlier in the year, simply by accounting for surface albedo cloud interaction and represents a reassessment of shortwave CRF in the Arctic.
To make this clearer, we extended this sentence in the conclusion (p.21 l.17):
*"Hence, the observed albedo trend during the campaign (Fig. 8) induces a transition in CRF from a warming to a cooling already for snow covered surface types, and thus, earlier in the season as reported during SHEBA."*

**For the SZA calculations, did you take into account that SHEBA moved northward during the year?**
As stated in the figure description: "Computed daily averaged SZA for 80° N in dashed black." it is fixed and only serves as a reference in which range of Figure 7 the underestimation/overestimation might take place. As shown in Fig. 1, 80° N is representative for ACLOUD and for the last months of SHEBA.

**Another idea would be to redo the SHEBA Intrieri et al. CRE study with this new information. Maybe a bit more work than anticipated for now, but it would be interesting.**
That's true but out of the scope of this study. The interpretation of the SHEBA CRF with the new knowledge is exactly why we made this hypothetical sketch. However, we do not have yet the radiative transfer model to represent the melt pond properties and, therefore, the SHEBA CRF was not revised. Additionally, we wrote in the conclusion that further effort is required to fully understand the seasonal cycle of solar CRF

in the Arctic by application of a similar approach to long-term observations like SHEBA or the upcoming MOSAIC. In general this approach using the parameterization from Gardner and Sharp (2010) is easy to apply to common ground based stations in the Arctic with cloud microphysical remote sensing instrumentation providing high quality LWP values, however a snow and ice dominated surface is required (no slush, melt ponds,… etc.). We added to the conclusion (p.21 l. 26):

*"The proposed method to estimate the surface albedo in cloud-free conditions using the parameterization from Gardner and Sharp (2010) can be easily applied to common Arctic long-term observations above snow and ice surface types, especially if high quality LWP measurements are available."*

**Page 13, line 2 & 3: Again, two examples of past-tense confusion. When was this done; for this paper or by an earlier investigator.**

Corrected.

**Page 13, line 10: Using both "indicate" and "might" in the same sentence almost obliterates the conclusion.**

We removed "might" from the sentence.

**Page 14, line 4: Can't find any red line in Figure 8.**

We changed it to: *"red scatter points"* in the figure description and in the given line.

**Page 17, line 5: "indispensable" is a strong word. Since it is impossible to know the cloud-free state with any accuracy, I would mellow the language here. If something indispensable is also im-possible, then why even try?**

We reworded the sentence (p.20 l. 11):

*"To estimate the warming or cooling effect of clouds on the surface REB in the Arctic from observations or models, a precise characterization of the cloud-free state is required."*

**Page 17, line 11: If by "local" you mean in one single specified point, then I'm confused. The lo-cal albedo is what it is; it is different at a different locale when the sea ice is variable; I still get hung up on this concept. If you are referring to the effects on the cloud free downwelling radiation, that I wanted to have elaborated on, the at least write "local cloud-free albedo".**

The reviewer is right, the word "local" is confusing in this sentence and was removed.

**Page 18: lines 27-31: Only part of the data is available, a large chunk is only cited. Why? Appendix A: OK; but, why don't show that this works, using aircraft passages over Polarstern, where you have both transmissivity and LWP?**

All primarily measured data are made available on the PANGEA database. DOI-links, where the specific dataset can be downloaded are given in the references. The publication of derived quantities such as LWP, cloud-free albedo and CRF is currently in progress. We wanted to wait for the reviews before publishing the data. The references will be added in the manuscript before publication.

We would have very much liked to validate the method using PASCAL data. Unfortunately this was not possible for different reasons. For the PASCAL campaign unfortunately no broadband albedo measurements are available from the ship. Without the information of broadband albedo, transmissivity cannot be interpreted or linked to cloud microphysics, due to multiple scattering.

In order to validate this retrieval we compared different observations during the ACLOUD/PASCAL campaign with our estimate of LWP. In Fig. 2 (this document) a comparison of LWP for the 2 June 2017

flight is shown. In situ observations of the Nevzorov probe (dataset on PANGAEA: https://doi.pangaea.de/10.1594/PANGAEA.906658 ) during ascents and descents before and after low-level section (orange scatter in Fig. 2) are used to derive the LWC profiles, which have been vertically integrated to obtain the LWP. During that flight only low-level clouds have been present. In addition to our transmissivity based retrieval (blue scatter) the MODIS overpass (0945 UTC) is shown in red scatter points collocated to the aircraft low-level flight sections. For the flight section close to Polarstern (last flight section), the HATPRO microwave retrieval of LWP on Polarstern is shown for the times where the aircraft was close by (dataset on PANGAEA: https://doi.pangaea.de/10.1594/PANGAEA.899898 or "Cloudnet" LWC data https://doi.pangaea.de/10.1594/PANGAEA.900106 ).

In general, the two first flight section have been over open ocean close to the MIZ. The third section is over the MIZ, while the last long section is in the vicinity of Polarstern, where a "staircase pattern" was flow.

In general, the transmissivity based retrieval shows a good agreement with the MODIS observations as well as the cloud microphysical in situ observations on the same aircraft. Unfortunately, the microwave radiometer retrieval shows significantly lower LWP values, which cannot be explain easily. From the good agreement with in situ and satellite observations we conclude, that our LWP retrieval fulfills the required accuracy to estimate the surface albedo in cloud-free conditions.

[Figure]

*Figure 2 Time series of different LWP retrieval during the 2 June 2017 ACLOUD flight. The blue scatter represent the transmissivity based retrieval presented in this study, the scatter points collocated MODIS cloud water path retrieval. In orange scatter points the vertically integrated in situ profiles from the Nevzorov probe (measuring LWC) on Polar 6 is shown during descents and ascents before and after the low-level flight sections through the clouds. In black scatter the "Cloudnet" retrieval of LWP from the Polarstern research vessel for the time period where the aircraft was close by.*

---

## Author Comment (AC3) · 19 Nov 2019

Referee comments are highlighted in **bold**, *changes* in the manuscript in *italic*.
* * *
**The manuscript "Reassessment of the common concept to derive the surface cloud radiative forcing in the Arctic: Consideration of surface albedo – cloud interactions" by Stapf et al. describes the calculation of cloud radiative forcing (CRF) from measurements collected from aircraft over the MIZ in the eastern Arctic near Svalbard during the ACLOUD campaign in June 2017. The authors leverage the spatial nature of their measurements over the heterogeneous surface albedo that is characteristic of the MIZ to identify and correct biases in calculations of CRF under such conditions. The targeted biases are specifically those associated with cloud-surface albedo interactions when the estimates of the shortwave clear-sky surrogates used in the CRF calculation are derived using radiative transfer models. The authors focus on a relevant problem that is suitable for ACP and this problem has received less direct attention from previous studies than analogous problems in the infrared. However, I respectfully disagree that this as a "reassessment" of CRF calculations, as the influence of surface albedo on CRF has been acknowledged previously and managed in various ways.**
Please find the discussion after the first major comment.

**The focus on albedo is warranted here because the observational platform and environmental conditions discussed make the present work particularly sensitive in that regard but I think the advancement promised by the title is overstated. I would be more comfortable with the paper being presented in either of the following ways:**
**(a) As calculations of CRF in the MIZ during ACLOUD with the albedo work being an important, though incidental component. In this case more work or more detailed explanations of the longwave calculations are needed (see comments below).**
**(b) If the authors wish to focus on the shortwave, they should drop the longwave data altogether and pitch the study as a proposed methodology for calculations of shortwave CRF in the MIZ where treatment of clear-sky shortwave fluxes requires special attention. In either case, the study needs to be more carefully contextualized and motivated by referencing previous work.**
**I also suggest a thorough copy editing for grammar, typos, missing words or letters, etc., of which there are many to be found.**

**Major Comments:**
**(1) The introduction and study motivations need substantial improvement. Some portions of the introduction actually belong in the Methods section. More troubling is that the study promises to improve upon (indeed, to "reassess") surface-based observations of CRF without actually referencing a single example of previous work on this subject, which has developed for several decades in the Arctic. I suggest rewriting the introduction to more clearly contextualize the present work in the existing literature. I have included some (not exhaustive) useful references throughout this review.**

The reviewer is right, we should have made more clear, that we focus on the shortwave estimate of CRF. Therefore, we changed the title to:

*"Reassessment of shortwave surface cloud radiative forcing in the Arctic: Consideration of surface albedo-cloud interactions"*

However, after including a literature overview of the common approaches to derive the CRF from the last two decades (section 3.1), an introduction focusing on the estimate of CRF and the conclusion part from section 3.5, we come to the conclusion, that the general knowledge of the seasonal cycle of shortwave CRF in the Arctic is based on simplified assumptions.

Neither available observational, nor model studies represent the discussed process (surface albedo-cloud interaction, see section 3.5) sufficiently. Climate models will not even rudimentarily represent this effect. However, these kind of models are used to estimate the cloud radiative feedback. Therefore, we want to point out the importance of reconsidering the described processes with respect to the estimate of the solar CRF. We added also in the conclusion (p.21 l.34):

*"The shortwave net irradiances depend not alone on cloud transmissivity and surface albedo, moreover the interaction between both needs to be represented. "*

To draw the attention also to the modelling community.

Another also quite important aspect is the homogenization of CRF estimates from different available studies/datasets. For the shortwave CRF, we provide an approach for snow surface types, which can be easily applied to long-term and high quality ground-based observations. As an example, the three important studies from Shupe and Intrieri (2004) during SHEBA, Dong et al. (2010) for Barrow/Alaska and Miller et al. (2015) for Greenland used different approaches for their CRF estimates.

We discuss the different approaches in the new section 3.1 and why a comparison of the CRF values of specific studies might be error-prone and misleading.

In section 3.5 we give reasons to estimate the shortwave CRF using the cloud-free albedo, but we also want to make clear, why our approach should give a better estimate of shortwave CRF.

There are two approaches available deriving the shortwave CRF specifically with a cloud-free albedo estimate.

The first one is from Miller et al. (2015), where the cloud-free observations are linearly fitted as a function of SZA. In Gardner and Sharp (2010) (Fig. 10b) it is shown that the albedo is a non-linear function of SZA, which is further affected by the snow grain size. In Fig. 2 from Miller et al. (2015) this fit is shown and shows significant deviations from the applied fit, potentially induced by snow grain size on the Greenland ice sheet. Neglecting these fluctuations can easily cause deviations in the shortwave net flux, and thus, also in the shortwave CRF for the specific prevailing conditions. For example assuming a downward irradiance of 550 Wm-2 for an SZA of 60° and applying a cloud-free albedo data point of 0.79 and 0.87 in the study from Miller et al. will cause deviations of up to 44 Wm-2 in the CRF. These albedo induced fluctuations in CRF are concealed in the obtained time series and might be related to precipitation events, warm or cold periods, or the seasonal cycle.

For the cloud-free albedo or upward shortwave estimate from the climatological approach in Dong et al. (2010) it is stated:

*"The clear-sky SW-up flux is estimated using the technique described in the study by Long [2005], where the clear-sky solar zenith angle dependence of the surface albedo is taken into account, and the clear-sky SW-up flux is estimated by the clear-sky albedo and SW-down flux."*

However, if we look at Long (2005) it is noted that only the observed albedo (cloudy) can reproduce significant changes in the surface albedo (like precipitation events) during longer cloudy periods (nicely shown in Fig. 1 in Long (2005)).

Although an albedo change like the presented one in the Southern Great Plains will not occur in the Arctic snow grain size can quickly change the surface albedo. That demonstrates that during longer cloudy periods, which are common in the Arctic, the cloud-free surface albedo estimate by this approach will induce significant uncertainties.

That is why we state in section 3.1 (p.6 l.27):

*"An application of the climatological approach is primarily limited by the high cloud fraction commonly observed in the Arctic (Shupe et al., 2011). It causes large uncertainties in the estimated cloud-free irradiance, as reported by Intrieri et al. (2002), preventing an application to long-term observations with reported high cloud fractions (e.g., Sedlar et al., 2011)."*

Also in the following sentence:

*"Although the climatological approach will produce a more realistic estimate of CRF (especially longwave) with reduced uncertainties and representation of humidity changes (Dong et al., 2006), it remains unclear how representative a monthly average of cloud-free irradiance with a monthly averaged cloud fractions often well above 90% can be."*

For the longwave range, we state now in the introduction (p.2 l.28):

*"As demonstrated by Walsh and Chapman (1998), the surface temperature change accompanied by the transitions from cloudy to clear skies is not an instantaneous effect; it rather occurs in the range of hours to days and potentially only advanced boundary layer models might be able to predict the transition between the two states after a given time."*

While the terrestrial instantaneous CRF studies from Shupe and Intrieri (2004), Sedlar et al. (2011) and Miller et al. (2015) can be nicely compared for saturation effects with increasing LWP, the estimate from Dong et al. (2010) appears unique as compiled by Miller et al. (2015). However, we know that the climatological approach with a colder surface temperature in cloud-free conditions should cause a weaker warming effect in the longwave CRF compared to the instantaneous approach. On the other hand, the cloud-free surface temperature response to changes in the cloud cover is a function of time as demonstrate by Walsh and Chapman (1998) and other surface fluxes. This circumstance raises the question, which time span after a dissipation of clouds is representative for longwave cloud-free fluxes estimated by the climatological approach, 5 minutes or 1 day? In the shortwave instead, we see an instantaneous response of surface albedo and shortwave net fluxes.

Another issue is added by the study from de Boer et al. (2011).

de Boer, G., Collins, W. D., Menon, S., and Long, C. N.: Using surface remote sensors to derive radiative characteristics of Mixed-Phase Clouds: an example from M-PACE, Atmos. Chem. Phys., 11, 11937-11949, https://doi.org/10.5194/acp-11-11937-2011, 2011.

There the temperature inversion caused by cloud top cooling is removed for the radiative transfer simulations by linearly interpolating between the surface and atmosphere above the cloud induced inversion, while keeping the surface temperature the same, and thus, mixing both approaches.

In the end, we have a lot of different estimates of CRF in the Arctic, and we conclude in section 3.1 (p. 6 l. 34):

*"By comparing the available studies, using different approaches to estimate the CRF, it becomes evident that the variety of strategies and the handling of physical processes involved in the CRF in the Arctic limits the comparability of the individual studies and our understanding of CRF in the Arctic."*

**(2) The title indicates a focus on shortwave processes. In my opinion is a fair representation of main the scope of the work, yet there are sections devoted entirely to the longwave and total CRF. There are complications in CRF calculations that are specific to the longwave (Allan et al. 2003) which are analogous to the issues affecting the shortwave; e.g., lapse rate and surface temperature responses to clearing skies (Long and Turner 2008) and systematic differences in water vapor between skies that are clear and those that are cloudy ("water vapor CRF", e.g., Dong et al. 2006). These issues are ignored and consequently the longwave and total CRF parts of the manuscript are somewhat confusing and do not serve a clear purpose. There is quite a lot packed into this study already. I think the study would be much clearer if only shortwave data were included, in keeping with the advertised focus of the work.**
Obviously, the original title was misleading. We adjusted it to make the overall focus more clear.
*"Reassessment of shortwave surface cloud radiative forcing in the Arctic: Consideration of surface albedo – cloud interactions"*

"…**there are sections devoted entirely to the longwave and total CRF…**":
In section 3.3 we also show the shortwave CRF. To quantify the total CRF in the Arctic and answer the question if a cloud is warming or cooling the surface, the longwave contributions needs to be included in this study, even when only the instantaneous effect is considered here.
We revised the manuscript in a couple of sections to make clear, that we simply derive the instantaneous longwave CRF, so the discussion about the longwave effects are not relevant at this point (besides the conclusion).
From section 4.2 and the conclusion we see a shift from a mainly warming effect of clouds during ACLOUD to a cooling one in the end of the campaign simply by accounting for surface albedo- cloud interactions. Furthermore we added in the conclusion (p.21 l.19):
*"In addition, the instantaneous longwave CRF approach might additionally induce an overestimate of the warming effect potentially shifting the total CRF further to cooling."*

**(3) Ehrlich et al. (2019b) (P3L30) is in review and the DOI provided is unreachable.**
We apologize for that, but in our submitted manuscript the DOI works, unfortunately after the processing of ACP (discussion version) it does not work anymore. However it is an open access journal (ESSD) and can also be found without https://doi.
**Thus, I cannot evaluate the processing of the radiometric data, which is central to this study. Indeed, I don't even know what equipment was used.**
In section 3.2 we referenced two papers, which fully describe the campaign, the used instrumentation, the processing, also in Wendisch et al. (2019).
**I wish to know more in particular because radiometric data from airborne platforms requires additional processing, though I am aware that the authors are familiar with some of these complexities (e.g., Ehrlich and Wendisch 2015). In addition to the instrument response corrections of the aforementioned work, how did you correct for tilt in the pyranometer?**
The inertia correction is applied by the approach from Ehrlich and Wendisch (2015). In cloud-free conditions we applied the approach from Bannehr and Schwiesow (1993) and Boers et al. (1998) as described in Ehrlich et al. (2019b) to correct for the tilt/attitude of the sensor. In cloudy conditions an attitude correction cannot be applied as the irradiance is mainly diffuse, also the upward solar fluxes (assumed mainly diffuse) were not corrected out of this reason. Further details are given in Ehrlich et al. (2019b).

**How did you correct the pyrgeometer data (measured at altitude) to represent the value that would be observed at the surface? (I think the answer is you did not [P7L9]).**

Yes we did not correct the CRF to represent values at the surface, because the impact is of minor importance.

In the next sentence we made clear that we are not analyzing single irradiance quantities, which are definitely influenced by the flight altitude. In the sentence the reviewer refers to, we argue that the vertical gradient of longwave irradiance remains the same, if we remove the cloud in radiative transfer simulation from the atmosphere (instantaneous approach). We clarified this sentence (p.10 l.8):

*"Due to the fact that the vertical gradient $dF\_lw/dz$ and $dF\_lw/dz$ below clouds remains almost the same with or without a cloud in the radiative transfer simulations (for atmospheric profiles as observed during ACLOUD), the observed CRF in flight altitude can be related to surface CRF values causing uncertainties below +-5Wm-2."*

To show that the average flight altitude of 80 m has only a minor impact on the CRF estimate, we simulated two observed thermodynamic states during the ACLOUD campaign and implemented simplified vertical homogenous clouds. In Fig. 1 (in this document) it can be clearly seen that the vertical gradient between cloudy and cloud-free single flux directions changes only slightly. Consequently, also the vertical profile of CRF (right panels) changes only slightly (see values for flight altitude of 80m and the surface embedded). We have the opinion that *"…can be related to the surface CRF…"* is correct regarding the other potential uncertainties of radiative transfer modelling and observations. But we added an uncertainty of +- 5 Wm-2 (conservative estimate) to account for this effect. The surface based inversion was a clear-sky profile observed by a dropsonde over water, of course the assumed cloud is kind of sketchy, but only serves as a test case for a stable atmosphere.

**I have a similar question about the KT-19, which does not observe thermodynamic temperature of the surface, but rather a brightness temperature relative to the FOV and dependent on the path to the target. Is there any reason to similarly correct the shortwave data for altitude given that such details are the focus of the present work?**

The flight altitude (average 80 m) does hardly affect the KT19 observations. We are aware of the study from Haggerty et al. (2003), which is cited in Ehrlich et al. (2019b). The applied corrections for flight altitude are necessary due to higher flight altitudes compared to the low-level observations during ACLOUD. Regarding the assumed surface emissivity for the KT19 wavelength range we use the results from Hori et al. (2006), which indicate an emissivity in this wavelength range and for the nadir viewing angle close to unity. As the values are only used for a linear interpolation of the temperature profile (required for the radiative transfer simulations) from in average 80 m to the surface, an influence on the simulated downward irradiance can be excluded. We added in the specific sentence (p.5 l.9):

*"The atmospheric levels below flight altitude were linearly interpolated to the surface temperature observed by the KT-19 assuming an emissivity of unity. The assumption of the black-body emissivity is justified by the high spectral emissivity for nadir observations in this wavelength range (Hori et al., 2006)."*

The impact of flight altitude on the estimate of shortwave CRF can also be seen in the specific panels of Fig. 1 (in this document) and is minor important.

[Figure]

*Figure 1 Simulated thermodynamic profiles during ACLOUD with a rarely found surface based inversion and a typical profile over sea ice with a lifted inversion at 400m. Cloud extent and assumed homogeneous LWC is shown in the grey box together with the temperature profile (left panel). First row: Shortwave fluxes and CRF. Second row: Longwave fluxes and forcing. Third row: Shortwave fluxes. Forth row: Longwave fluxes. Fluxes are always given for cloudy and cloud-free (removed cloud) case. The surface emissivity is set to 0.99, the surface albedo 0.8 and the SZA 60°.*

**(4) Unless I misunderstand something, I believe there are errors in the presentation of the CRF equations. This is simple to correct if it is merely a typo in the subscripts, but if the equations were applied as stated the study's results could be impacted. Specifically, be careful how you use the terms "all sky" and "cloudy sky" because they are not equivalent in CRF nomenclature. CRF may be defined as either (all − clear) or as CF(cloudy − clear) where CF is the cloud fractional occurrence and the other terms refer to net radiative fluxes for "all" (clear and cloudy sky conditions together), "cloudy" (only times when clouds are present) and "clear" (clear skies). Ramanathan et al. discuss both definitions. Admittedly, they are confusing on the nomenclature themselves in using the term "cloudy" early on in discussing their Eqs (1) and (2), but their meaning becomes clear when they introduce Eq. (4). Your Eq. (2) is therefore incorrectly stated; it shows the maximum CRF (e.g., Intrieri et al. 2002). The "cld" subscript needs to be "all" or the entire right side of the equation needs to be multiplied by the cloud fraction. For your purposes, and for the Arctic in general, I suggest the former. This commentapplies to equations throughout the text.**

The reviewer is right that definition was kind of confusion, although cloudy is not equal to overcast (Ramanathan et al.). To avoid any issues with the cloud fraction definition, we replaced "cld" with "all" in the whole manuscript.

**(5) P7L9: I do not agree that this is a good assumption. About 70% of the downwelling longwave at the surface originates from atmosphere below the altitude of your aircraft (Ohmura 2001). Your assumption is plausibly (not certainly) valid if the atmosphere is isothermal between surface and the base of the cloud. While this condition could be met, in your case studies (Fig. 2), it is not. Your observations of flux at the altitude of the aircraft should be corrected to represent the surface.**

From our personal point of view the statement "**70% of the downwelling longwave at the surface originates from atmosphere below the altitude of your aircraft (Ohmura 2001)**" is kind of misleading and gives the impression that everything above does not matter, which is definitely not true. Please have a look on Fig. 1 (in this document). Even for single terrestrial flux directions the average flight altitude will cause a deviation well below 10 Wm-2 (quite conservative estimate again), in the net fluxes even less, because of the same vertical gradient.

The atmosphere during ACLOUD was in the most cases not isothermal between surface and cloud base (during the low-level sections a least), which can be seen in Fig. 14 from Wendisch et al. (2019), where the distribution of the observed longwave net fluxes in the cloudy state shows always negative numbers indicating a colder "effective cloud base temperature" compared to the surface. The profile from the surface based inversion was during clear-sky conditions over open ocean and did not "affect" or observation during that day.

Nevertheless, the estimate of CRF is hardly influenced by the flight altitude. Please see also the answer to comment 3 together with Fig. 1 (in this document).

**(6) P8L11-P9L7: The approach you suggest to achieve a downwelling clear-sky shortwave is intriguing, but more information is needed for future studies to adopt your method. As written, it is not reproducible and there is no information on the sensitivities of the estimation; for example, I would expect that a filter of constant width assumes that leads are randomly distributed and roughly of the same size. I would also like to know more about the justification for your choice of a Laplace distribution as the most appropriate filter for this application.**

We added in section 3.4 the equation and settings of the estimated smoothing kernel to make it more reproducible. In addition we added (p.12 l.2):
*"This enables a more reliable estimate of the CRF in the heterogeneous MIZ and over the specific surface types, taking into account that the complexity of surface albedo fields in the MIZ can only be insufficiently represented by this simplified approach to estimate the areal averaged albedo."*

Please see the next comment for further information about the simulations/estimate of filters.

**Specific Comments**
**(1) Given that your upwelling shortwave is observed from an aircraft platform, the FOV covers and enormous area. I therefore do not understand why your albedo measure-ments (e.g., Fig 3a) are not implicitly area-averaged, even if observed from a relatively low altitude (e.g., Podgorny et al. 2018).**

Thanks for bringing up this point. Yes, the albedo observed in the average flight altitude of 80 m is "already smoothed". We did a mistake in estimating the smoothing kernel in the 3D simulation for the surface (0 m) and unfortunately neglected this altitude induced effect. So we revised the whole procedure and estimate the kernels for a representative flight altitude of 80 m.
In Fig. 2 (in this document) one of the clear-sky simulations is shown for a lead with 1 km width embedded in homogeneous sea ice similar to the study from Podgorny et al. (2018). In the upper panel the albedo is color coded as a function of scale and altitude. Also the comparison between surface albedo (0 m) and flight altitude (80 m) is shown, where the altitude induced smoothing effect can be seen, which is rather weak. The upward irradiance is cosine weighted so the alleged FOV is not really representative for the upward fluxes. In an average flight altitude of 80 m, 80 % of the signal is represented within a radius of 102 m below the aircraft. Even when we did not resolve the ground level, the smoothing effect is still in a small range, but of course the estimated filter width and shape changed accordingly (embedded in the lower panel).

[Figure]

*Figure 2 Broadband shortwave 3D radiative transfer simulations in clear-sky conditions of a 1 km lead embedded in homogeneous sea as shown in the lower bound of the upper panel. Upper panel: Vertical distribution of albedo. Comparison between surface albedo, albedo as observed by an aircraft in 80 m flight altitude and the smoothed albedo using the filter embedded in the lower panel. Lower panel: Comparison*

Also for the transmissivity based LWP retrieval, we recalculated the kernel and give the settings in the appendix to represent average flight altitude, as shown in Fig. 3 (this document).

[Figure]

*Figure 3 Broadband shortwave 3D radiative transfer simulations in homogeneous cloudy conditions with a constant LWP of 50 gm-2 and a 300 m lead embedded in homogeneous sea, as shown in the lower bound of the upper panel. Upper panel: Vertical distribution of shortwave downward irradiance in the color code and comparison between surface albedo, albedo as observed by an aircraft in 80 m flight altitude and the smoothed albedo using the filter embedded in the middle panel. Middle panel: Comparison between 3D downward irradiance perpendicular to the lead as it would be observed in 80 m altitude (solid), 1D simulations using the observed albedo as input and the final product 1D simulations using the smoothed albedo. Lower panel: Impact of smoothed albedo on the LWP retrieval for the homogeneous 50 gm-2 cloud. SZA is set to 60°.*

As we state in the appendix the variety of potential surface heterogeneity prevents a specific solution. These results should work out for ACLOUD, where we exactly observed these simulated surface / cloud scenes, which brings us back to "**I would also like to know more about the justification for your choice of a Laplace distribution as the most appropriate filter for this application.**"
It simply fits to our observations and simulated cases and enables us to make it applicable to our study. A general solution like the one from Pirazzini and Raisanen (2008) requires unfortunately surface albedo maps, which we do not have.

The changes due to the new smoothing kernels caused small changes in Fig. 3, 4, 9 and 10, and the related statistics (text section 4.2 and 3.4.1). Due to the shorter scale of the clear-sky kernel in Fig. 3 changes in the areal averaged albedo occurred accordingly, which do not change the general picture. Changes in the kernel for the LWP retrieval caused only small changes in strongly fluctuating albedo sections, but did not affect the obtained statistics or retrieval of cloud-free albedo.

However, we found that occasionally surface albedo values exceeded the range of Gardner and Sharp (2010) parameterization and filtered the specific values out and added (p. 18 l.23):

*"Rarely occurring surface albedo values above/below the range of the parameterization from Gardner and Sharp (2010) have been filtered out."*

Hence, the average values of CRF and Albedo in Fig. 10 and the distribution in Fig 10a and 9 changed slightly accordingly, see also changes in the values given in the text, abstract (p.1 l.12):

*"Applying ACLOUD data it is shown that the estimated average shortwave cooling effect by clouds almost doubles over snow and ice covered surfaces ( -62Wm-2 instead of  -32Wm-2), if surface albedo-cloud interactions are considered."*

And conclusion (p.21 l.14):

*"For the ACLOUD campaign, characterized by snow on sea ice in the beginning melting season, the averaged shortwave CRF estimate over homogeneous sea ice of -32Wm-2 (cooling) almost doubles to -62Wm-2, when surface albedo-cloud interactions are taken into account."*

**(2) P2L18: Consider using "L" for longwave instead of "t" for terrestrial to avoid confusion with the terrestrial surface. "S" would then represent "shortwave" rather than "solar".**
We adapted the subscripts and wording in the whole manuscript.

**(3) P2L18: It is true that RT simulations are a common approach, but there are other approaches as well. Long and Ackerman (2000) present a method for estimating clearsky fluxes that implicitly accounts for the albedo dependencies on sky conditions. (Refer also to Dong et al. (2010) and Long (2005)). More in line with your study, Miller et al. (2015) parameterized clear-sky albedo for their RT simulations, though your situation is considerably more complex with regard to surface cover. Other studies have analyzed the dependencies. These studies do not necessarily detract from your work here and in some ways maybe motivate it, but either way really need to be referenced.**
We included the different approaches in the new literature overview in section 3.1. (See also reply to the first comment in this document)

**(4) P4L15 - P5L4: (1) I don't understand how (or when) you combined the dropsondes and NYA radiosoundings.**
Before or after most of the low-level section the aircraft descended or ascended from/to higher altitudes, and thus, in situ profiles of thermodynamic state were observed. In addition dropsonde data from P5 could be used. For each low-level section we need a representative local thermodynamic profile for the calculations of F_dw (impact was shown for example in Fig. 2 in the manuscript). Therefore, we replaced the layers from the radiosoundings (either from Ny-Alesund or Polarstern (the temporal and spatial closer one)) with the local profiles to obtain a merged representative profile.

**(2) You do not say how you represent the atmosphere above the height of the soundings; this is necessary (even if estimated using a standard atmosphere) to a reasonable effective TOA (say, 60 km). (3) You do not say how you represent atmospheric gases that are radiatively active in the infrared, but were not measured by the sounding (not notably, CO2, but also O3, methane, etc.).**
Thanks for this remark. We agree with the reviewer, that these are important information to make the study/ RT simulations reproducible. We included all information in section 2.3.

**(5) P5L14-16: I do not agree that the upward longwave between clear and cloudy conditions is equal, but I doubt that this is what you actually mean to say. I think you mean you defined them to be so because (a) you do not account for the response of the atmospheric lapse rate (and surface skin temperature) to changes in sky cover and your calculations for the longwave are therefore "instantaneous" CRF (e.g., Miller et al. 2015),**

We apologize for this unclear definition. After the literature overview and the definitions section this should be clear now.

**and (b) that you also neglect the influence of differences in the amount of longwave reflected from the surface between clear and cloudy skies. It is acceptable to make the first assumption (see Allan et al. 2003), but you should include the emissivity term and then proceed with your CRF calculation. See my next comment.**

We slightly adjusted the longwave CRF definition (Eq. 3 - 6) and account now also for the reflected downward longwave irradiance. (See changes in the definition section). Accordingly also the average longwave CRF value in Fig. 10b,c and section 4.2 changed slightly. See also next reply.

**(6) P4 Eq. (4): This variable is more frequently referred to as (longwave) "cloud radiative effect" (CRE) (e.g, McFarlane et al. 2013; Viudez-Mora 2015; Cox et al. 2015, 2016) and should be distinguished somehow from CRF. At the surface CRF and CRE are different, the former being the difference in the net fluxes and the latter being the difference in the incident fluxes. Confusion sometimes arises because the terms are frequently used interchangeably in satellite studies, being that the terms are equal against the backdrop of space.**

Thanks a lot for this remark. We totally agree that a standardized nomenclature (and definition) is required in the literature and a difference should be made between CRF and CRE. We added (p. 7 l.10):
*"As was stated by Cox et al. (2015) the CRF definition refers to net fluxes, while the cloud radiative effect (CRE) characterizes only changes in the downward irradiance."*

As we stated and show now in the definitions section (p.7 l.11), the upward terms cancel for the instantaneous CRF estimate (e.g. Shupe and Intrieri, 2004; Miller et al., 2015) accounting for the reflected residual.
But the reviewer is right, the reflection term needs to be represented, why we changed the formulation (Eq. 3 - 6). Assuming an emissivity of 0.99 (Warren, 1982) the reflected residual is 1 % of the CRE, and thus, will cause a difference in the derived long CRF below 1 Wm-2. (Average values in the Fig. 10, section 4.2 changed slightly)

**(7) P5L28: Multiple scattering also depends on the albedo of the sky. How do you account for this?**
The albedo of the sky (scattering processes of the atmosphere) is implemented by the radiative transfer simulation of the local atmospheric profile. We continuously update/run the simulation along the flight track using the closest thermodynamic profile and the local estimated areal averaged albedo.
For the smoothing kernel of the areal average albedo estimated by the idealized 3D radiative transfer study we had to fix the atmosphere (represented by the subarctic summer standard profile), which will influence only the smoothing filter, and thus, will induce only a minor impact on the resulting 1D online simulations using the smoothed albedo.

**(8) P6L30: Is 60deg SZA representative of the flight conditions?**
These are rounded values but representative for the conditions over sea ice during that flight. To clarify that we only used this fixed values for this sensitivity study to avoid an impact of changing SZA and surface albedos during the specific profiles and only focus on the thermodynamic impact in this section 3.3 we added (p.9 l.7):

*"The surface albedo and SZA is fixed for this sensitivity study to 0.8 and 60 respectively, similar to the observed conditions over sea ice during that flight, in order to avoid any effects induced by changing SZA or surface albedo."*

**(9) P9L20: You might also consider that your pyranometer is at best a 2% instrument and thus you might expect uncertainty of around 10 Wm-2 in the measurement. Thus, Figure 4 looks quite good. I am however curious about the source of the bimodality of the solar CRF in Figure 4. My first thought is that one of these peaks is associated with ice-covered areas and the other with open water, pointing to some residual bias in the method.**

To rule that out, we show in Fig. 4 (in this document) the 2D histogram of this shortwave CRF distribution of Fig. 4 (in the manuscript) as a function of sea ice concentration. The occurrence of leads do not induce a bias in the derived downward irradiance as the bimodal distribution can be seen also in the 2D distribution. Only the data of the Polar 5 aircraft produce this bimodality. We found a slight correlation to aircraft heading for this aircraft, what might indicate an issue related to the observations not the radiative transfer simulations. However, we have double checked the processing and could not find any issues regarding the offsets for the correction applied by the approach from Bannehr and Schwiesow (1993) and Boers et al. (1998) as described in Ehrlich et al. 2019b.

In Ehrlich et al. 2019b also a comparison between the two aircraft is shown indicating a good agreement. Unfortunately, we cannot resolve this bimodality during this flight, it could also be related to slightly different local conditions not captured by the in situ profiles or aerosol/haze layers as a huge area was covered. But we should be aware that these differences are well below the measurement uncertainty (<3 %) of these instruments and the albedo smoothing method is not the cause of this deviation.

[Figure]

*Figure 4 2D histogram of shortwave CRF derived during the clear-sky flight (as shown in the manuscript Fig. 4) as a function of observed sea ice fraction (I_f).*

**(10) Section 3.4: It would substantially increase the value of this section if you contextualized your simulated biases with your observations. For example, it would be interesting to see the biases from Figure 3c plotted over Figure 7 in the phase space of the figure panel that is most appropriate.**

We do not fully understand the point of the reviewer here. The bias from figure 3c is attributed to the changing downward shortwave irradiance due to multiple scattering, while Figure 7 shows a completely different process and a quantitative estimate of surface albedo-cloud interactions.

In Fig. 10 we show the impact of the surface albedo-cloud interaction (Fig. 7) on the ACLOUD observations by comparing the different approaches.

**(11) P18L9: You have mentioned SHEBA a couple times, but have not referenced it (Uttal et al., 2002), nor have you defined the acronym.**

We give now an appropriate citation and the introduced the acronym.

---

## Author Response (AR2)

**Reply to Report #1**

We would like to thank the reviewer for her or his time and the beneficial comments, which will help to improve the manuscript. Please find the replies to the referee comments below. The page and line numbers given by the referee relate to the manuscript in discussion, the numbers in our reply to the revised manuscript.

**Referee comments** are highlighted in **bold**, *changes* in the manuscript in *italic*.
* * *
**The authors have completed a thorough revision, though I do have some remaining suggestions after consideration of which I think the manuscript will be ready for publication.**

**(1) I am pleased to see the inclusion of a new literature review on CRF in section 3.1. Though nitpicking, the authors may consider working in the perspective of Allen et al. 2003 and references therein into this section too, which is relevant framing for their overall discussion, in particular for the interpretation of LW CRF described in that section and page 7 (L17-19).**

We totally agree with reviewer, but we were hesitant to include it, due to the different perspective (TOA and subsampling approach), which in fact represents a third methodology. However, it supports the longwave CRF discussion and we added now a paragraph in section 3.1:

*"Similar issues with the cloud-free reference net irradiances are reported from satellite-based approaches (Allan and Ringer, 2003), where a subsampling of cloud-free regions is used for the estimate of CRF similar to the climatological approach. The satellite-observed cloud-free conditions are in general more stable and drier compared to the cloudy regimes assumed to be cloud-free, which affects the obtained longwave CRF values and results in inconsistencies when compared to climate model longwave CRF estimates (Allan and Ringer, 2003), where the cloud-free irradiances are calculated by neglecting clouds in the radiation scheme."*

**(2) I still have philosophical differences with the authors that prevent me from seeing how this study is a "reassessment" of a fundamental concept. The authors are correct that there are a variety of methods in use and that these methods are not necessarily comparable, but this is the nature of CRF, which is a hypothetical quantity. Here, the authors devote attention to a more sophisticated treatment of the albedo and I think reached some interesting conclusions. If they insist that this is a reassessment then I suppose I can live with it as it is largely stylistic point.**

The reviewer is right, the concept of CRF using cloud-free fluxes is established, already by its definition. Indeed, we wanted to point out that a more "sophisticated treatment of the albedo" is recommended, as we try to show in the sections 4 & 6 by quantifying the potential impact, which suggests that a reassessment of the shortwave CRF (not the concept itself) is necessary (regarding the hypothesis in Fig. 6 and the related differences in Fig. 5). We had therefore removed the "the common concept to derive the" out of the title to bring the focus on the implementation. Also with respect to CRF in climate models and the estimate of cloud radiative feedbacks in the Arctic by partly simple albedo parameterizations e.g. in CMIP5 models; we would appreciate it if we could make this "largely stylistic point", also to draw the attention of the modeling community to this topic.

**(3) In their response to my previous comments, the authors provide a sensitivity study to show the errors in CRF based on the height of the instruments. However, in the main text, there continues to be some handwaving on the treatment of the data collected at altitude, especially the LW. If the authors will not implement atmospheric corrections, I suggest that this analysis be included as supplementary material and referenced in the appropriate section.**

In order to explicitly demonstrate the effect of the flight altitude, we have compiled a supplement *"Impact of flight altitude on the surface radiative energy budget and cloud radiative forcing derived from low-level flights during the ACLOUD campaign"*.

Regarding an atmospheric correction:

In our point of view, an atmospheric correction is not essential for the main conclusions of the study. The analysis in the supplement clearly shows: it is reasonable to know that there is a slight bias, which was estimated in the manuscript in a quite conservative way (±5 Wm-2); but the relative contribution to the total uncertainty caused by this issue is negligible compared to other potential sources of uncertainty in the derivation of CRF. But we follow the remark of the reviewer regarding the supplement, it is an important issue of this measurement approach.

We restated the whole paragraph (now in section 5.3) and refer to the supplement:

*"A special aspect of this dataset concerns the measurement strategy itself, whereby the irradiances are observed in the aircraft flight altitude (during low-level sections in average 80 m) and in various atmospheric thermodynamic profiles. Radiative transfer simulations of all available profiles during the campaign indicate that the derived total CRF in flight altitude may have an offset less than ±2 Wm-2 compared to surface-related/surface-based observation, depending on the prevailing thermodynamic profiles. Details are given in the supplement provided to this study."*

**(4) In the revision the authors argue that this is a study on SW processes. This is clearly stated now in the title and abstract where the focus is entirely on the SW and the LW is absent. However, within the text there is still consideration of the LW, which becomes important only at the end where the authors argue that their treatment of the SW causes the net CRF to go from strongly positive (LW dominant) to "near neutral" (perhaps "weakly positive" would be better). This is an important statement in the manuscript and it does not appear in the abstract, but is an interesting result that gives motivation to present LW calculations. I suggest including this finding in the abstract and perhaps working the concept of the balance of warming and cooling processes as a hypothesis in the introduction that warrants testing the sensitivity of SW CRF methods on net CRF results. This would set up the scope nicely for the reader.**

The reviewer is right, we should have included this motivating aspect of total CRF. We added in the abstract:
*"As a result, the observed total (shortwave plus longwave) CRF shifted from a clear warming effect to an almost neutral one."*

An in the last paragraph of the introduction:

*"This allows us, in combination with the derived longwave CRF, to analyze the general concept of a warming or cooling effect of clouds on the sea ice during the campaign and illustrate the impact of the surface albedo-cloud interaction for the total balancing effect of clouds in the springtime MIZ (section 6)."*

**(5) P21L19-20: Is this so? Nominally, if the boundary-layer lapse rate and surface temperature response under clearing skies is accounted for in the clear-sky term, the value of the clear-sky term should become more positive. Given that your LWD is measured at altitude, it is a bit difficult to say how things would change if the response were included.**

If a cloud layer dissipates, the strongest contribution of upward emission (in a cloudy atmosphere the cloud top cooling) will be displaced to the surface, which will consequently cool. The surface will respond quicker to the new environment compared to the atmosphere, and thus, the upward irradiance (surface temperature) will quickly/strongly decrease, while the atmosphere under the assumption of any external forcing (strong large scale subsidence, advection of new air masses) will only slightly change. In case of large-scale subsidence the inversion will strengthen, and thus, the atmosphere will become warmer which will increase the downward irradiance. Thus, the net longwave irradiance will be less negative (less outgoing) in the real cloud-free atmosphere due to a colder (significantly less emitting) surface and an unchanged or slightly warmer atmosphere (same, slightly more emitting). More critical is the aspect of an in general drier cloud-free atmosphere, which can cause a compensating effect on downward irradiance (less emitting atmosphere).

[Figure]

*Figure 1 SHEBA seasonally averaged radiosoundings separated by cloud fraction (CF) in observations during cloudy (CF > 95 %, solid) and cloud-free (CF <5%, dashed) conditions and seasons; winter (light blue), spring (blue), summer (red) and autumn (black). Average surface temperatures during the soundings for cloud-free (open) and cloudy (filled) conditions are shown in scattered circles.*

In order to underline this statement we analyzed SHEBA atmospheric soundings:

https://data.eol.ucar.edu/dataset/13.818

In Fig. 1 the seasonally averaged soundings are separated by cloudy (cloud fraction > 95 %) and cloud-free (cloud fraction < 5 %) conditions and are shown together with the temporal collocated observed averaged surface temperatures. Cloud fraction and surface temperature are obtained from:

https://data.eol.ucar.edu/dataset/13.114

Soundings without valid surface pressure have been neglected. For the average profile the net, downward and upward irradiances in assumed cloud-free conditions are simulated. The results are shown in Tab. 1.

The difference between the irradiances for cloudy profiles simulated as cloud-free and the ones for the real cloud-free atmosphere represents the average bias between the climatological and instantaneous radiative transfer based approach. In Tab. 1 the cloud-free net irradiances in the cloudy atmosphere from autumn, spring and winter are more negative (outgoing). This is likely caused by the less stable atmosphere and relatively warmer surface temperatures. Thus, the longwave CRF by the instantaneous radiative transfer based approach using the cloudy soundings for the longwave simulations of the cloud-free reference will be overestimated (stronger warming effect). In late spring and summer it will potentially

depend more on the prevailing conditions and the value of CRF itself and a weaker overestimate has to be expected. Please note that the SON might be influenced by bad statistics for clear conditions associated with the high cloud fraction in this season.

The flight altitude (during ACLOUD average 80 m) would have had a minor impact on the results (see Tab. 1).

*Table 1 Simulated cloud-free longwave irradiances at the surface and in 80 m (flight altitude of an aircraft) in Wm-2 for the seasonally averaged SHEBA profiles shown in Fig. 1.*

| | DJF | | MAM | | JJA | | SON | |
|---|---|---|---|---|---|---|---|---|
| Atmosphere: | Clear | **Cloudy** | Clear | **Cloudy** | Clear | **Cloudy** | Clear | **Cloudy** |
| F_net | -43.6 | **-54.7** | -65.7 | **-71.8** | -72.0 | **-72.6** | -38.7 | **-68.1** |
| F_dw | 130.0 | 145.5 | 180.1 | 178.5 | 236.0 | 238.0 | 151.6 | 193.0 |
| F_up | 173.6 | 200.2 | 245.8 | 250.3 | 307.9 | 310.6 | 190.3 | 261.1 |
| F_net (80 m) | -44.5 | **-55.5** | -66.9 | **-72.3** | -72.6 | **-73.8** | -40.4 | **-70.3** |
| F_dw (80 m) | 132.2 | 145.8 | 181.1 | 177.6 | 237.8 | 236.2 | 156.3 | 192.1 |
| F_up (80 m) | 176.7 | 201.3 | 247.9 | 250.0 | 310.5 | 310.0 | 196.7 | 262.4 |

We restated:

 *"This is potentially due to the colder surface temperatures and frequent presence of surface-based temperature inversions in cloud-free conditions, causing a less negative longwave net irradiance compared to the cloud-free simulated cloudy atmosphere, which is in general less stable and exhibits a warmer surface temperature."*
and:

 *"In late spring and summer, the surface temperature difference between cloud-free and cloudy state is smaller (Walsh and Chapman, 1998), and smaller differences between the CRF estimates of the two approaches should be expected. Therefore, the bias in the longwave CRF estimate between the climatological and radiative transfer based approach will be controlled by the prevailing conditions and the CRF itself."*

However, we would prefer not to include this analysis in the already extensive manuscript, as it distracts from its focus (shortwave) and poses a new problem in making the paper more comprehensive. We had already stated,

*"From autumn to spring, the longwave CRF derived from the radiative transfer based approach **should** tend to produce a more positive (warming) CRF."*

to underline the hypothetical statement.

**(6) I am not convinced that the brightness temperature measured by the KT15 at flight altitude is representative of a surface skin temperature without applying atmospheric and emissivity corrections, the latter being even more critical over the heterogeneous surface and the former being necessary**

**despite the narrow-band (though I know of know proven methods). I suggest at minimum drawing more attention to this assumption in the text: While the Hori article does show nadir emissivity near unity at the center of the instrument band, the range is more like ~0.96-0.995 across even the sensitivity of this narrow-band instrument and even this assumes a consistent surface type.**

An important aspect concerning our study is that this emissivity assumption for the surface temperature estimate using the KT19 has almost no effect on the results presented in this study. Only for the lowest ~80m of the atmosphere the temperature is linearly interpolated between the surface brightness temperature and air temperature observed in ~80 m. The downward irradiance in ~80 m, will not be influenced by a potential biased surface temperature below that altitude.

The brightness temperature information from the KT19 consists of three components:

1. Emission from the surface with the physical surface temperature and the specific surface emissivity
2. The impact of absorption and emission of the atmosphere between the aircraft and the surface
3. The contribution of downward longwave radiation reflected at the surface in this wavelength range

[Figure]

*Figure 2 Simulate vertical profile of difference between nadir brightness temperature (radiance integrated from 9.6 to 11.5 μm, assuming constant spectral response of the KT19 instrument) and surface temperature (emissivity 1, only atmospheric impact) for all atmospheric profiles obtained during ACLOUD/PASCAL.*

We have to point out again that our flight altitude in average was 80 m (e.g. see image in the supplement); for observations above sea ice often below that value. In Fig. 2 (this document) the atmospheric profiles obtained during ACLOUD/PASCAL which have been used for the simulations of downward irradiance are now simulated to show the vertical profile of "KT19-like" nadir brightness temperatures. A surface emissivity of 1.0 is used in order to show only the impact of atmospheric absorption between the surface and the aircraft (component 2). The small values indicate that an atmospheric correction for the absorption term between aircraft and surface is not necessary (can be neglected) in this wavelength range and the low flight altitudes. By neglecting this component, we obtain a simple formulation for the KT19 brightness temperature $T_B$ consisting of the two remaining terms:

$$\sigma T_B^4 = \epsilon_s \sigma T_s^4 + (1 - \epsilon_s)\sigma T_{atm}^4.$$

We assume in the following a surface emissivity $\epsilon_s$ of 0.99 like in the study from Haggerty et al. (2003).

*Haggerty, J. A., J. A. Maslanik, and J. A. Curry, Heterogeneity of sea ice surface temperature at SHEBA from aircraft measurements, J. Geophys. Res., 108(C10), 8052, doi:10.1029/2000JC000560, 2003.*

We solve for the surface temperature $T_s$:

$$T_s = \sqrt[4]{\frac{T_B^4 - (1 - \epsilon_s)T_{atm}^4}{\epsilon_s}}.$$

The brightness temperature of the atmosphere $T_{atm}$ depends on the atmospheric profile, the conditions (cloudy or cloud-free) and in case of clouds on the cloud base height and optical thickness. Applying and continuous atmospheric correction would be a challenge itself.

To solve this issue appropriately to the circumstances, we have simulated in Fig. 3 all the profiles with the surface emissivity of 0.99 for cloud-free conditions (Fig. 3a) and cloudy conditions (LWP of 50 gm-2 between 200 and 400 m) (Fig. 3b).

[Figure]

*Figure 3 Same as Fig. 2, however, a surface emissivity of 0.99 was assumed. (a) Cloud-free simulated. (b) Cloudy simulated with a cloud between 200 and 400 m (LWP of 50 gm-2).*

For clear-sky conditions a relatively constant averaged bias of approximately 0.5 K is indicated. Thus, in cloud-free conditions the surface physical temperature is in average 0.5 K warmer than the brightness temperature by the KT19. The majority of the flights have been conducted in cloudy conditions where a weak bias in average below 0.1 K is indicated (Fig. 3b). The impact of flight altitude can be neglected.

We restated in section 3.2:

"The temperature profile below the lowest flight altitude was linearly interpolated to the surface _brightness_ temperature observed by the KT19. _For cloudy conditions radiative transfer simulations (assuming a surface emissivity of 0.99, Hori et al., 2006) of the observed atmospheric profiles indicate that neglecting an atmospheric correction for cloudy conditions is justified and the brightness temperature of the KT19 can be related to the surface temperature causing uncertainties below ±0.2 K. During cloud-free conditions a correction of 0.5 K have been added to the brightness temperature (average value obtained from radiative transfer simulations) to compensate for atmospheric and surface emissivity effects. The impact of flight altitude (average 80 m) can be neglected and potential surface temperature uncertainties hardly affect the downward irradiance simulated in flight altitude."_

**(7) P5L26: "location and time" instead of "location"; P21L7: "enables" to "enables us to"**

Corrected.

Additional changes:

- We included the citation of the now published dataset with the retrieved quantities of LWP equivalent, CRF and cloud-free albedo required to reproduce the results of this study (in "Data availability") (https://doi.org/10.1594/PANGAEA.909289 ).

- We updated the citation of the SHEBA data used in Fig. 6 to the more formal/general NCAR/UCAR EOL data page https://data.eol.ucar.edu/dataset/13.818 , not the "acknowledgement way" as described on https://atmos.uw.edu/~roode/SHEBA.html

**Reply to Report #2**

We would like to thank the reviewer for her or his time and the beneficial comments, which will help to improve the manuscript. Please find the replies to the referee comments below. The page and line numbers given by the referee relate to the manuscript in discussion, the numbers in our reply to the revised manuscript.

**Referee comments** are highlighted in **bold**, *changes* in the manuscript in *italic*.
* * *
**Overall: The manuscript is significantly improved by the authors' edits and it is worth publishing. However, it still requires major revision. Although the writing is generally understandable, the entire manuscript should be reviewed and edited for grammar and clarity. Overall, the main issue with the manuscript is the organization, which makes it extremely difficult to follow. It should be possible to easily find the results of the paper and discussion of those results, which is currently very difficult since they are intermixed with introductory material and methods.**

**Please rewrite the statement of what this work does on page 3, lines 8-10, to be more general and relate better to what is actually done in the paper. The current lines suggest that this work specifically focusses on the two processes that change the surface albedo in cloudy conditions (1) cloud-induced weighting of the transmitted downward irradiance to smaller wavelengths and 2) a shift from direct to diffuse irradiance in cloudy conditions). However, this seems to be too specific based on the results presented. A better statement would be that this work evaluates the effects of surface albedo-cloud interactions on CRF, including the impact of areal vs local albedo and exploration of the dependence of surface albedo on cloud LWP for different surface types, and during ACLOUD; and that this work presents the effect of surface albedo-cloud interactions on CRF during ACLOUD. This statement should be parallel with what is in the abstract and conclusions.**

The reviewer is right. This last paragraph of the introduction did not clarify the main aspects of this paper (and structure) and we have accordingly restructured it. Please see the track change file.

**Sections 3 and 4: These sections intermix introductory material, methods, and results. To my read, Section 3.1 is introductory material, Section 3.2 is definitions, Section 3.3 is an uncertainty estimate for a particular source of error, and Section 3.4 presents an interesting result that is perhaps novel? Section 3.4.1 seems to give an overall estimation of uncertainty and it is unclear to me why it is a subsection of 3.4. Section 3.5 refers to the processes (1) and (2) above and, as such, would seem to be a main result of the paper, so why is it buried here? Why are all these subsections grouped together in one section? Section 4.1 seems to be methods again. Section 4.1.1 is entitled "Application to the observations," but various things have already been applied to various observations, so a more descriptive title is needed; perhaps "Dependence of surface albedo on cloud LWP for ACLOUD measurements." The Section 4.2 title, "Correction of CRF," is also insufficiently descriptive and should be changed.**

**Please put introductory material (Section 3.1) into the introduction, or its own section. Please consider having a section on methods, that groups all methods together (e.g. 2.2, 3.2, 4.1), and a section on uncertainty (which would include Sections 3.3 and 3.4.1.) The results seem to be contained in Sections 3.4, 3.5, 4.1.1 and 4.2, so these should be grouped together. Furthermore, section titles should make**

**clear that 3.5 represents a model study related to the two processes showing how broadband albedo changes with LWP for different surface types (related to process 2) and attenuation of incident irradiance by clouds (related to process 1), whereas 4.1.1 shows how broadband albedo changes with LWP for the ACLOUD measurements (related to process 2).**

We have invested an extensive amount of time to restructure the manuscript to meet the requirements and try out various arrangements, but it was quite delicate due to the linkage between methods and results. We tried to give the section more descriptive titles.

The new structure reads:

**1      Introduction**
**2      The concept of cloud radiative forcing (CRF)**
2.1     Review of approaches to derive the CRF
2.1.1    Estimating net irradiance in cloud-free conditions
2.1.2    Handling of surface albedo
2.2     Definitions and process handling
2.2.1    Longwave CRF
2.2.2    Shortwave CRF
**3      Observations and modeling**
3.1     Airborne measurements and instrumentation
3.2     Radiative transfer simulations
3.3     Necessity of a local thermodynamic profile for the estimate of CRF
**4      Modeling the surface albedo-cloud interaction**
4.1     Impact of clouds on surface albedo
4.2     Impact on shortwave CRF
4.3     Seasonal cycle of shortwave CRF
**5      Refining the derivation of shortwave CRF**
5.1     Considering surface albedo heterogeneities and horizontal photon transport
5.2     Retrieval of cloud-free albedo from cloud-sky observations
5.3     Uncertainties
**6      Impact of surface albedo-cloud interaction on the CRF during ACLOUD**
**7      Summary and conclusions**
**Appendix A: Transmissivity-based retrieval of LWP equivalent**

We tried to follow the reviewer`s advice by putting the literature overview right after the introduction. We avoided an inclusion in the introduction due the resulting length. It was also quite important to us to show our approach in the "Definitions section" right after the literature overview.

We "dug" the modeling study on surface albedo cloud interaction out and gave it an own section to emphasize the importance for this study (**"…main result of the paper, so why is it buried here?"**) and not include it in a results section, where it might be buried again ("**The results seem to be contained in Sections 3.4, 3.5, 4.1.1 and 4.2, so these should be grouped together.**").

Regarding the "Methods" section we decided to move "Necessity of a local thermodynamic profile…" to "Radiative transfer simulations" as it fits perfectly to the already discussed handling of the temperature profiles described in this section. However we think this subsection **("Section 3.3")** is not "**an uncertainty**

**estimate for a particular source of error"**, moreover it represents an important aspect of this measurement approach and the studied processes in the MIZ.

Left over are 5.1, 5.2 and a section combining now all uncertainties of the presented retrievals and issues regarding this dataset (section 5.3). These three subsections were grouped into kind of a "Methods" section "Refining the derivation of CRF", although the content already "**presents an … result".**

Same issue with the section "Retrieval of cloud-free albedo from cloud-sky observations", which contains already an application to ACLOUD data for one flight.

However, putting the section "Retrieval of cloud-free albedo…" already in a "methods" section, while the reasons ("modeling the surface albedo-cloud interaction") are given later in the results section, would have been hard to motivate ("**Please consider having a section on methods, that groups all methods together (e.g. 2.2, 3.2, 4.1)**", **The results seem to be contained in Sections 3.4, 3.5, 4.1.1 and 4.2, so these should be grouped together.**).

We have really struggled with the rearrangement and hopefully achieved a clearer outline. Please see the track change file/revised manuscript.

**Specific comments:**
**Please put a horizontal space (extra line break) between paragraphs or begin new paragraphs with an indent so the reader can tell when a paragraph ends.**

Corrected.

**Page 6 last line – Page 7 lines 1-2: Please delete these sentences. I do not see the point of them, as the current work would also fall into the category of being limited in comparability to previous results due to a different means of computing CRF. Furthermore, the current work relies on aircraft flights, which are not available to many of the previous studies and therefore could not be applied to them.**

We agree, but on the other hand this is a pretty important aspect we want to point out in this section. We changed the paragraph to:

*"The potential systematic differences in shortwave CRF estimates associated with the respective assumed surface albedo motivate to provide a quantitative measure of these differences as well as an improved solution for the albedo reference to enable a more harmonized understanding of CRF."*

However, we do not understand why our results and methods are not applicable to other studies? In fact, on the contrary, our approach would even benefit from high quality ground-based remote sensing (LWP retrievals). The cloud-free albedo retrieval should be easily applicable for common long-term observations (like Intrieri et al., 2002; Shupe and Intrieri, 2004; Dong et al., 2010; Sedlar et al., 2011; Miller et al.,2015) or the current MOSAiC campaign. The parameters required for the Gardner and Sharp parameterization: broadband surface albedo, LWP, SZA are available and even the impurity content of snow could be estimated from snow samples.

If needed for inhomogeneous surface conditions, the areal averaged albedo can be obtained from helicopter measurements or kind of "line albedo" measurements (like during SHEBA, Perovich et al. (2002)) for example, and would be helpful also for coastal sites with a strong reported gradient in albedo (impact discuss in Ricchiazzi and Gautier (1998) and Kreuter et al. (2014)).

As we show now in the supplement (reply to Report #1) our airborne observations are no measurements from space and we can reasonably relate the results to the surface REB.

**"…as the current work would also fall into the category of being limited in comparability to previous results..":** Of course, because we present a method with which comparability based on one distinct reference value (cloud-free albedo) would be given, not influenced by the cloud properties itself or fits or extrapolation of conditions a few days/weeks ago. Without comparability; what is the use of another dataset that gives another vacuous number of 36.9 Wm-2 total CRF…

**Page 7 lines 4-6: Please delete these lines or change them to "Below, we derive the … CRF. Our approach is unique in that we use a continuous … cloudy conditions." Please add to the sentence any aspect of using the areal-averaged albedo that is unique to this work. E.g, if it's true you could say, " Below, we derive the … CRF. Our approach is unique in that we use a continuous … cloudy conditions, and in that we account for horizontal photon transport through using the areal-averaged albedo to compute downward shortwave fluxes, as discussed below."**

We restated this section accordingly:

*"Below, we derive the radiative transfer based instantaneous CRF. Our approach is unique in that we use a continuous estimate of the cloud-free surface albedo of snow and ice obtained from observations in cloudy conditions, and in that we account for horizontal photon transport through using an areal-averaged albedo to compute the downward shortwave irradiances, as discussed below."*

**Pages 7-8**
**In these equations, the two alphas in Eqn. 7 are later replaced with two non-equal terms, which is not mathematically correct. You can avoid this and simplify by adding the subscripts "all" and "cf" to the relevant alphas in Eqn (8). With this change, Eqn (10) could then replace Eqn (9), which would no longer be needed. The text from lines 19-22 could then be moved to just before Eqn (8) to provide the necessary context, after revising it to read "The surface albedo changes for … Warren (1982). Thus, the measured surface albedo has been separated … alpha_cf."**

We have rearranged the section accordingly to eliminate equation 9 (old manuscript) and shorten the section. Please see the track change version.

**Page 11, line 12: reference needed after "(increasing grain size)."**

We added in P. 13 L.12 (old manuscript) the citations from Warren (1982) and Gardner and Sharp (2010) and slightly changed the sentence:

*"The impact of snow properties on the spectral surface albedo is characterized by the fact that with decreasing SSA (increasing effective grain size) the absorption at longer wavelengths increases (Warren, 1982; Gardner and Sharp, 2010)."*

**Lines 16-19: references are needed.**

We added a reference:

"Two processes influencing the broadband snow albedo are related to the transition from cloud-free to cloudy atmospheric conditions. In an overcast atmosphere with clouds of sufficient optical thickness,

mainly diffuse radiation illuminates the surface as compared to cloud-free conditions, when the direct shortwave radiation dominates *(Gardner and Sharp, 2010)*."

**Also, this text through page 14 line 2 seems to be an expansion of information that was given in the introduction and should therefore be moved to the introduction. Retain here only enough information needed to introduce and discuss the results (e.g. "Fig. 5 shows the effect of …").**

In the introduction we introduce the general concept by explaining the two processes relevant for the main conclusion of the paper. In our point of view, such details would distract in the introduction. However, where it is absolutely necessary to understand the processes shown in Fig. 3 and 4 (new manuscript) and following conclusions in the next sub-sections, the reader would have to look up 10 pages earlier. We think it would be best to have this information where it is needed (in the following lines) and where the effects are illustratively shown (main reason for this section 4.1).

**Section 4.2 seems like an important section but the results shown in Fig. 10 are not described sufficiently. The caption/text and the legend do not agree. What is "Corrected delta F?" This phrase occurs nowhere else in the paper.**

We have described/discussed the features more extensively (providing more statistical background) and adjusted the inconsistent labels in Fig. 10 and avoided "corrected". We apologize for this inconsistences. Please see the track change file.

**The conclusion (page 20) slowly transitions from a description of the literature, which is not always referenced (lines 12-14, line 17) to results from the current work. Is line 18-20 from the literature or current work? Please reference all statements that are already known and start a new paragraph for the findings of this work, which should begin with a sentence using language such as "we find" or "our results show" to make this clear.**

We have slightly restructured the conclusion (now "Summary and conclusions") and used bullet points to separate the single topological sections. We included citations for the general motivational sentences at the beginning of each section in order to separate our work more clearly from the literature.

**lines 12-14:** We added:

*"In the transition region between open ocean and closed sea ice (the MIZ), the thermodynamic state of the atmosphere changes on horizontal scales of a few kilometers (Lampert et al., 2012), which influences the cloud-free reference state and the resulting simulated radiative field."*

**line 17:** *"Variability in sea ice concentration is closely linked with fluctuations in surface albedo."* We think this statement is so general and obvious that we do not see a reason for a citation.

**line 18-20:** We added:
*"The derivation of downward shortwave irradiances under cloud-free conditions in heterogeneous surface albedo conditions requires an estimate of the effective areal average surface albedo, determining the multiple scattering on large spatial scales (e.g., Kreuter et al., 2014)."*

Additional changes:
- Due to the editing for reasons of clarity there have been slight changes in the legend of Fig. 3 and 10. In Fig. 6 the icons have been enlarged.

[revised manuscript text omitted]

$$-F_{lw,all}^\uparrow + F_{lw,cf}^\uparrow = -\epsilon_s \cdot \sigma \cdot T_s^4 - (1-\epsilon_s) \cdot F_{lw,all}^\downarrow + \epsilon_s \cdot \sigma \cdot T_s^4 + (1-\epsilon_s) \cdot F_{lw,cf}^\downarrow, \tag{4}$$

the upward term :

$$-F_{lw,all}^\uparrow + F_{lw,cf}^\uparrow = -\epsilon_s \cdot \sigma \cdot T_s^4 - (1-\epsilon_s) \cdot F_{lw,all}^\downarrow + \epsilon_s \cdot \sigma \cdot T_s^4 + (1-\epsilon_s) \cdot F_{lw,cf}^\downarrow, \tag{5}$$

reduces to:

$$-F_{lw,all}^\uparrow + F_{lw,cf}^\uparrow = (1-\epsilon_s) \cdot (F_{lw,cf}^\downarrow - F_{lw,all}^\downarrow). \tag{5}$$

This approach assumes a the same constant surface temperature in the cloudy and cloud-free state, and thus, represents the commonly defined instantaneous longwave CRF similar to  Shupe and Intrieri (2004), Sedlar et al. (2011), and Miller et al. (2015) and should be considered in the interpretation of the CRF values as discussed in the previous section. The essential input for radiative transfer simulations in the longwave wavelength range is the atmospheric temperature profile, the absorber gas profile and aerosol. Hence, the longwave instantaneous CRF  becomes independent of the upward irradiance and reduces to:

$$\Delta F_{lw} = F_{lw,all}^\downarrow - F_{lw,cf}^\downarrow + (1-\epsilon_s) \cdot (F_{lw,cf}^\downarrow - F_{lw,all}^\downarrow). \tag{6}$$

**2.2.2 Shortwave CRF**

The shortwave component of the CRF is given by:

$$\Delta F_{sw} = \left( F_{sw,all}^\downarrow - F_{sw,all}^\uparrow \right) - \left( F_{sw,cf}^\downarrow - F_{sw,cf}^\uparrow \right). \tag{7}$$

The surface albedo $\alpha$  is defined as the ratio of $F_{sw}^\uparrow$ and $F_{sw}^\downarrow$  . To account for surface albedo changes due to different illumination conditions (cloudy, cloud-free) and cloud optical thickness (Warren, 1982), the surface albedo is decomposed into an albedo observed in cloudy conditions ($\alpha_{all}$) and an albedo, which continuously represents the cloud-free state ($\alpha_{cf}$). Thus, the instantaneous shortwave CRF definition : reads:

$$\Delta F_{sw} = \left( F_{sw,all}^\downarrow - \alpha_{all} \cdot F_{sw,all}^\downarrow \right) - \left( F_{sw,cf}^\downarrow - \alpha_{cf} \cdot F_{
[revised manuscript text omitted]

---

## Author Response (AR3)

**Reply to Editor Comment**

We would like to thank the editor for the beneficial comments and the useful and extensive list of literature, which helps to relate this manuscript to the available literature and to acknowledge the significant work done in the past within this field.

Please find the replies to the comments below. **Editor comments** are highlighted in **bold**.

**1) First, it is important to correct the error in the definition of cloud radiative forcing (CRF), as originally defined by Ramanathan et al (1989). The Authors are using "all" to represent cloudy-sky conditions, whereas Ramanathan et al's definition states that "all" does truly represent all sky conditions (cloudy plus clear). The reason why Ramanathan et al defined CRF in this manner was so that their definition would not require the determination of the cloud fraction to compute CRF; users must merely estimate the SW and LW clear-sky fluxes and subtract these from measurements under all conditions. This definition does not impact the research or conclusions in this manuscript, but the use of "all" to represent "cloudy" conditions must be corrected. The alternative is to present values of the cloud radiative effect (CRE), which is defined to be cloudy minus clear. If the Authors choose to report values of CRF rather than CRE, then they must re-compute their CRF values using (all minus clear) instead of (cloudy minus clear). Depending on what the Authors choose, they must also carefully edit the text (and figures; e.g. Figure 6!) to change all instances of "all" to "cloudy", or all CRF values to CRE values.**

Thanks for pointing this out again. There seems to be a misunderstanding, which we had hoped to have clarified in the reply to the editors comments before entering the discussion phase (Initial Submission: Authors' response file Upload (22 Jul 2019)).

We derive the cloud radiative forcing (CRF) using the net irradiances indicated "all" minus "cf" (cloud-free), see Eq. (2). "all" refers to the cloudy atmosphere (consisting of cloud droplets plus aerosol particles (which we neglected) plus the gaseous components). This assures that in the cloudless atmosphere the CRF vanishes, as originally defined by Ramanathan. Thus, the CRF is derived for each data point of the local measurements of irradiances regardless of the cloud fraction, which justifies using the term "all-sky". The usefulness of this approach, which strictly follows the definition by Ramanathan, becomes obvious from Fig. 10 where the CRF reveals values around 0 Wm-2 in case of cloud-free conditions.

We apply the same nomenclature and CRF derivation as Miller et al. (2015) (their section 3, Eq. 2).

In Cox et al. (2015)
"At the surface, CRE differs from cloud radiative forcing (CRF) in that CRE is for the downwelling infrared component only, rather than the net flux, but they are similar because the upward component of CRF is small."

the CRE is defined as the difference between cloudy minus clear downward irradiance. We do not derive the CRE. We apply the instantaneous CRF definition (Eq. 3 - 6) using net irradiances instead of downward irradiances.

Regarding the label of the conceptual Fig. 6 where we use "alpha_cld" for the albedo in cloudy conditions (cloudy sky albedo is commonly used in literature). We are convinced that the use of "cloudy" in this sketch is more convenient/illustrative and can be used as an equivalent for "all-sky" after Ramanathan et al. (1989):

"The term "cloudy" should not be confused with the term "overcast". A cloudy domain is overcast only when it is completely covered with clouds. … Thus, clouds should cause any difference between the net radiative heating averaged over the domain and the heating averaged only over the clear-sky regions in that domain."

We hope to have eventually clarified this misunderstanding.

**2) This manuscript has evolved significantly as a result of this review process. In round 1, multiple reviewers expressed concern that there was a lack of previous literature cited in this manuscript. The Authors then provided additional citations and references on Arctic cloud radiative forcing. But as the manuscript evolved to focus on shortwave radiation, I believe that there are many relevant papers related shortwave radiation that the Authors should seriously consider citing. This is especially true in section 4 and for Figures 3, 4, 5 and 6. In particular, there is a wealth of literature on the interaction of shortwave radiation with Arctic and Antarctic sea ice, pioneered by Tom Grenfell, Don Perovich, Richard Brandt, Steve Warren and others. Below is a list of references that you should consider citing and referencing. In addition, I would encourage the Authors to add a paragraph at the beginning of section 4 that places their current manuscript in context of the body of work from others in this field, including the authors listed above.**

We have included the papers in a couple of sections and added a paragraph in section 4 where the available literature is introduced. Thank you for these useful suggestions. Please see the revised manuscript/track change version.

[revised manuscript text omitted]